# Mosaic gastruloids reveal a temporal restriction for developmental cell competition

Joshua D. Frenster [1] ✉, Stephen Babin[1], Pablo Casani-Galdon[1], Joel B. Josende-Garcia[1], Pau Pascual-Mas[1], Gaëlle Robertson[1], Shlomit Edri[2], Alexandra E. Wehmeyer [3], Sebastian J. Arnold[3], Jordi Garcia Ojalvo [1] & Alfonso Martinez Arias [1,4] ✉

Selective elimination of suboptimal cells is critical for the developmental integrity of early mammalian embryogenesis. Cell competition is a non-autonomous quality control in which 'winner' cells outcompete viable but suboptimal 'loser' cells based on fitness differences. Here we investigate cell competition dynamics using mosaic mouse gastruloids, a 3D embryonic stem cell-based model of gastrulation. Introducing just two *Trp53*-deficient supercompetitor cells suffices to impair growth in neighbouring wild-type cells through mitochondrial apoptosis. Competition is tightly restricted to a developmental transition stage between primed pluripotency and early gastrulation and involves gene regulatory networks of pluripotency exit. Heterochronic gastruloids from developmental stage-shifted cells, EpiGastruloids, and dynamic p53-degrons reveal that both winners and losers must reside within this permissive stage, during which acute relative p53 protein levels determine competitive outcomes. These findings advance our understanding of cell fitness evaluation and establish gastruloids as a powerful 3D model for investigating developmental stage-specific cell competition in mammalian embryogenesis.

During mammalian development, a single totipotent zygote gives rise to the pluripotent cells of the blastocyst, which go on to form the epiblast and engage in the process of gastrulation. Gastrulation is a conserved developmental process in all animals that establishes the body plan[1]. In mammals, gastrulation concurs with rapid cell proliferation, which increases the number of epiblast cells ~100-fold within 36 h (refs. 2,3).

In the mouse embryo, gastrulation starts at about E6.0 and is accompanied by a wave of apoptosis that eliminates up to 35% of the cells in the epiblast[4–6]. While some of this reflects autonomous death of severely compromised cells, it is complemented by a non-autonomous

mechanism that ensures that among the viable cells only the fittest prevail[7]. This has been suggested to reflect a process of cell competition in which cells compare their relative fitness. 'Fitter' cells are designated 'winners' and survive, while relatively less fit neighbours become 'losers' and are eliminated[8–10].

Cell competition was first described in *Drosophila* mosaics[11] where healthy wild-type (WT) cells undergo apoptosis when exposed to 'supercompetitor' cells overexpressing dMyc[12]. In mammalian development, loser cells can be defined by non-lethal defects such as low cMyc expression[7,13–15], impaired Hippo[16] or BMP signalling[6], autophagy deficiencies[13], mitochondrial defects[17], aneuploidy[13] or even species

[1]Department of Medicine and Life Sciences, University Pompeu Fabra, Barcelona, Spain. [2]Faculty of Biomedical Engineering, Technion – Israel Institute of Technology, Haifa, Israel. [3]Institute of Experimental and Clinical Pharmacology and Toxicology, Faculty of Medicine, University of Freiburg, Freiburg, Germany. [4]ICREA, Passeig de Lluís Companys, 23, L'Eixample, Barcelona, Spain. ✉e-mail: joshua.frenster@gmail.com; alfonso.martineza@upf.edu

identity during interspecies chimerism[18]. Loser cell fate can lead to p53 protein stabilization causing apoptosis[6,15,19]. Due to its central role in fitness perception, loss of the tumour suppressor p53 results in the emergence of supercompetitor cells, which stand at the apex of cell competition and cause apoptosis in healthy neighbouring WT cells[20,21]. Relative differences in pluripotency have also been proposed to factor into cell competition[14], as cells of more naive pluripotency outcompete less pluripotent cells after blastocyst co-injection[22].

Here, we use mouse gastruloids, a three-dimensional (3D) stem cell-based in vitro model of perigastrulation[23–25], to study cell competition during early mammalian development. This system mimics events in the early embryo and allows detailed testing of parameters that have been associated with this phenomenon. We find that cell competition is restricted to a narrow developmental window associated with the transition from formative pluripotency to gastrulation, during which acute p53 protein levels determine competitive outcomes. Exploiting the robustness and modularity of the gastruloids, we find that introducing a small number (~1% of initial population) of p53-knockout (p53KO) supercompetitor cells into WT gastruloids suffices to induce mitochondrial apoptosis in neighbouring cells. Our results define the physiological and temporal parameters of cell competition in embryos and establish gastruloids as a valuable system for studying this key developmental process.

## Results

### 2D cell competition occurs only at confluence during differentiation

To investigate cell competition in mammalian embryogenesis, we generated fluorescently labelled mouse embryonic stem (mES) cell clones that allow us to monitor interactions between winner and loser populations. An H2B–mCherry-labelled WT clone functions as the primary 'cell under investigation' (shown in green, '*WT–mCherry*') and an H2B–emiRFP670-labelled WT sister clone as the competitive neighbourhood environment (magenta, '*WT–emiRFP*') (Extended Data Fig. 1a). Building on previous observations[6,20], we created a system of unequal competitive strength by mutating *Trp53* (p53KO, full loss of function) in the H2B–emiRFP670 clone, resulting in a supercompetitor subclone (Fig. 1a,b and Extended Data Fig. 1a–d) (magenta, '*p53KO–emiRFP*').

We first tested cell competition in two-dimensional (2D) coculture settings by mixing WT–mCherry with WT–emiRFP or p53KO–emiRFP cells. In pluripotent copassaging conditions, we observed a slow drift in favour of both emiRFP clones independent of their *Trp53* status (Extended Data Fig. 1e,f). Similar observations were made when naive pluripotent cocultures were grown to full confluence without passaging (Extended Data Fig. 1g–i). When allowing cells to exit naive pluripotency towards a primed state (EpiSC-like; N2B27 + AA + FGF) without passaging, the presence of p53KO cells impaired the growth of cocultured WT cells, resulting in their decline over time (Extended Data Fig. 1j), which confirmed previous studies on 2D supercompetition[6,20]. However, timing of this depended on initial seeding density. Competition became apparent only once the cells reached confluence, at which point daily medium changes became insufficient to sustain the large number of cells per well (Extended Data Fig. 1j–l). When the same 2D cocultures exiting naive pluripotency were passaged as EpiSC-like clumps every 72 h to avoid confluence, they did not compete despite growing as mixed mosaic colonies (Extended Data Fig. 1m,n). In contact-free transwell cocultures, WT–mCherry cells experienced no growth impairment from cocultured p53KO cells independent of their pluripotency status (Extended Data Fig. 1o).

These observations suggest that, in adherent 2D conditions, p53KO-mediated cell competition occurs only when non-naive cells reach full confluence, which hints towards resource or space limitation rather than the developmental competition mechanisms found in the embryo. We thus moved on to investigate cell competition in

gastruloids, a 3D model of mammalian gastrulation that more closely resembles embryonic development[23–25].

### p53KO cells act as supercompetitors in mosaic gastruloids

Gastruloids are formed by aggregating defined numbers of ES cells, which, over 5 days and with a Wnt signalling pulse (CHIR99021 at 48–72 h), develop into elongated and polarized structures that represent parts of the embryo at embryonic day (E)8.5[23–25] (Fig. 1c and Extended Data Fig. 2a). Mixing 150 + 150 WT–mCherry and WT–emiRFP cells (WT + WT) during aggregation resulted in mosaic gastruloids composed of both clones with slight overrepresentation of WT–mCherry cells after 120 h (Fig. 1d). However, mixing 150 WT–mCherry with 25 p53KO–emiRFP cells (WT + p53KO), resulted in mosaic gastruloids mostly composed of p53KO cells (Fig. 1e). Using fluorescence-activated cell sorting (FACS) to titrate cell stoichiometries, we observed that as few as two p53KO–emiRFP starting cells per gastruloid are sufficient to measurably impair the growth of 150 co-aggregated WT cells (Fig. 1f,g and Supplementary Fig. 1). These data indicate that p53KO cells behave as supercompetitors in 3D gastruloids and that very few p53KO cells suffice to impair the growth or survival of neighbouring WT cells.

p53KO cells, whether in pure or mixed gastruloids, promoted multi-axis formation and led to larger overall aggregates. Furthermore, while WT + WT mosaic gastruloids grew to the average total size of their two monoclonal forms, WT + p53KO mosaic gastruloids grew to the same size as pure p53KO gastruloids despite eliminating most WT–mCherry cells and starting with only half the number of p53KO cells, suggesting compensatory growth during competitive interactions (Extended Data Fig. 2b–h).

*Trp53* and its homologues have been reported to be important for mesendoderm formation in mouse embryos[26]. Nonetheless, phenotypically normal triple-knockout embryos have been described at E11[27], and p53-deficient extraembryonic cells can contribute to embryonic gut formation[28]. We thus set out to test whether germ layer specification was influenced by p53 deletion in our mosaic competitive conditions. While WT + WT gastruloids at 120 h form organized regions of all three germ layers, WT + p53KO gastruloids contain smaller disorganized centres of *FoxA2* expression, aligning with impaired endoderm formation (Fig. 1h and Extended Data Fig. 2i,j). By contrast, patterned *Tbx6* and *Tbxt* (*Brachyury*) expression (mesoderm) was not impaired in WT + p53KO gastruloids. Bulk transcriptomic analysis of FACS-isolated cells from mosaic 120 h gastruloids confirmed that p53KO cells were biased towards paraxial mesoderm and caudal epiblast formation, while showing impairments in haemato-endothelial, primordial germ cell and, to a lesser extent, endodermal development (Fig. 1i and Extended Data Fig. 2k–n), and clustered towards more posterior E8.5 embryonic compartments than their WT counterparts (Extended Data Fig. 3a). At 120 h, WT–mCherry cells displayed only very slight differences in their lineage marker expression when together with p53KO cells instead of WT neighbours, suggesting a gentle non-autonomous influence on lineage choice but no lineage-specific elimination during competition (Fig. 1i). Although almost no differentially expressed genes were detected in WT–mCherry cells from mosaic WT + WT or WT + p53KO gastruloids at 48–72 h after aggregation (Extended Data Fig. 3b), gene set enrichment analysis (GSEA) revealed a downregulation of growth- and proliferation-related pathways, followed by a later upregulation of the p53 pathway, indicating a transcriptional stress response to their p53KO neighbours (Extended Data Fig. 3c,d).

### Cell competition results in stagnation of loser cell numbers after 48 h

Gastruloids are spatially organized 3D structures. However, using confocal microscopy and 3D segmentation in WT + WT and WT + p53KO mosaics, we observed that cells remained evenly distributed without clustering, independently of their origin (Fig. 2a–e). Development in

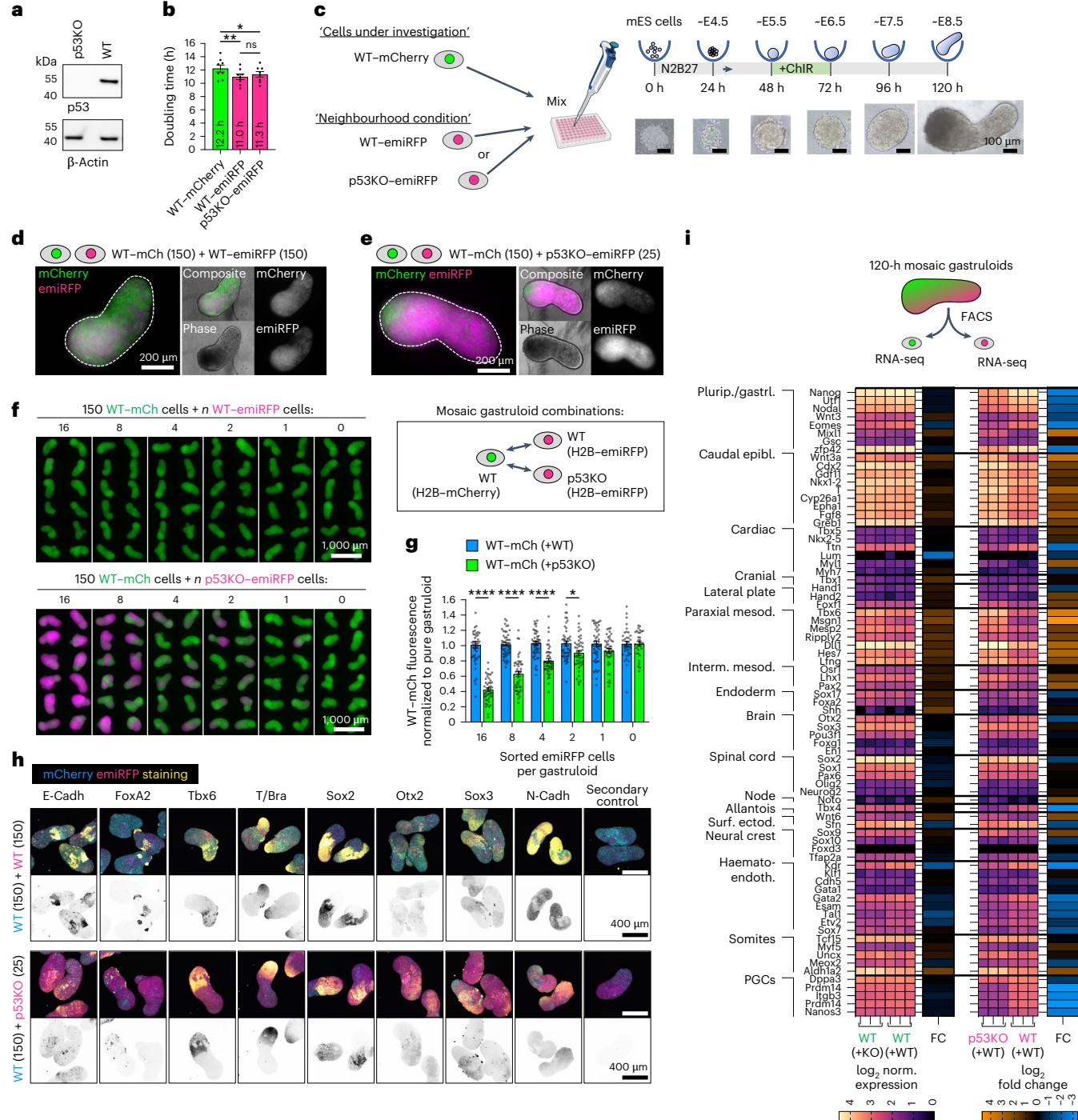

**Fig. 1 | p53KO cells act as supercompetitors in mosaic 3D gastruloids.**
**a**, Western blot of p53KO−emiRFP and WT−emiRFP cells stained against p53 and β-actin (Extended Data Fig. 1d). **b**, Doubling times of clonal mES cell lines. $n = 8$ indep. experiments. Top to bottom: $P_{adj} = 1 \times 10^{-2}$ and $P_{adj} = 3.5 \times 10^{-3}$. **c**, Schematic protocol of mosaic gastruloid formation with representative brightfield images. Equivalent timepoints of mouse development annotated at the top. **d**,**e**, Representative widefield fluorescence micrographs of mosaic gastruloids from 150 WT−mCherry + 150 WT−emiRFP cells (**d**) or 150 WT−mCherry + 25 p53KO−emiRFP cells (**e**) at 120 h. **f**, Cell number titration of 0−16 WT−emiRFP (top) or p53KO−emiRFP (bottom) cells seeded in mosaic gastruloids with 150 WT−mCherry cells each. Representative widefield fluorescence micrograph montage of 120 h gastruloids depicted as overlay of mCherry and emiRFP channels (Supplementary Fig. 1). **g**, Quantified mCherry fluorescent intensity per gastruloid normalized to pure mCherry gastruloids ('0') as a proxy of WT−mCherry cell growth. $n = 48$ gastruloids per condition from 3 independent experiments. (left to right) $P_{adj} < 1 \times 10^{-4}$, $P_{adj} < 1 \times 10^{-4}$, $P_{adj} < 1 \times 10^{-4}$, $P_{adj} = 2.9 \times 10^{-2}$.

**h**, Fate marker staining of mosaic WT + WT and WT+p53KO gastruloids at 120 h depicted as maximum intensity projections. Inverted grey-scale marker staining below the composites. **i**, Heatmap of developmental lineage marker expression from bulk mRNA sequencing of sorted populations from mosaic 120 h gastruloids. Plurip./gastrl., pluripotency/gastrulation; epibl., epiblast; paraxial mesod., paraxial mesoderm; interm. mesod., intermediate mesoderm; surf. ectod., surface ectoderm; hemato-endoth., hemato-endothelial; PGCs, primordial germ cells. Heatmaps depict $\log_2$-normalized transcript levels and $\log_2$ fold change of WT−mCherry cells from WT + WT versus WT+p53KO gastruloids, and of p53KO−emiRFP versus WT−emiRFP cells from mosaic gastruloids. $n = 3$ independent experiments. All data presented as mean ± s.e.m. Statistical significance was determined using row-matched one-way ANOVA with Tukey's post hoc test (**b**) or two-way ANOVA with Šidák's post hoc test (**g**) and is depicted as \*$P < 0.05$, \*\*$P < 0.01$, \*\*\*$P < 0.001$, \*\*\*\*$P < 0.0001$; n.s., not significant. Scale bars, 100 μm (**c**), 200 μm (**d** and **e**) and 400 μm (**h**).

gastruloids follows the same temporal pattern as in embryos[25], thereby offering a model to interrogate the temporal dynamics between competing clones. Although epifluorescence microscopy and quantitative single-gastruloid flow cytometry confirmed that population proportions in WT + WT mosaic gastruloids (50% + 50%) remain largely stable, the proportions in WT + p53KO mosaic gastruloids (86% + 14%) change rapidly between 48 h and 96 h, with p53KO cells becoming the dominant population (Fig. 2f–i and Extended Data Fig. 4a,b). Flow cytometry revealed that at 72–96 h around 70% of H2B–emiRFP cells had phagocytosed fluorescent material from neighbouring H2B–mCherry cells (Fig. 2j,k). However, p53KO status did not affect this percentage, suggesting that phagocytosis serves as a debris clearance mechanism rather than an active competition mechanism (Fig. 2l). Absolute fluorescent cell counts per gastruloid revealed that WT–mCherry cell numbers stagnate after 48 h when in mosaic gastruloids with p53KO cells, but not with other WT cells (Fig. 2m,n and Extended Data Fig. 4c–f). Together, these results suggest that the same WT–mCherry population that codevelops with WT cells without hindrance, becomes impaired in growth and/or viability when exposed to p53KO supercompetitor cells. Mouse gastruloids at 48 h, 72 h and 96 h correspond to the embryonic epiblast around E5.5, E6.5 and E7.5 (±0.5), respectively. This suggests that the timing of cell competition in gastruloids corresponds to the perigastrulation apoptosis peak in mouse embryos[4].

## Cell competition is mediated by mitochondrial apoptosis

After observing that WT–mCherry cells exhibit near-exponential growth in WT + WT mosaic gastruloids, but stagnate after 48 h in the presence of p53KO cells, we investigated whether this was caused by altered proliferation, or apoptosis. Surprisingly, WT–mCherry cells displayed only a small non-significant reduction in the percentage of mitotic phospho-Histone 3 (pH3)-positive cells (Fig. 3a,b and Extended Data Fig. 5a,b) and did not have any significant alterations to their cell cycle profile as measured by 5-ethynyl-2′-deoxyuridine (EdU)/Hoechst staining, independent of whether they were in mosaics with other WT or p53KO cells (Fig. 3c,d and Extended Data Fig. 3c–e). p53KO cells contained comparable percentages of pH3-positive cells but spent less time in G1/0 than their WT counterparts (Extended Data Fig. 3b,c). When investigating apoptosis in 72-h mosaic gastruloids, we found elevated percentages of Annexin-V-positive, as well as cleaved caspase 3-positive apoptotic WT–mCherry cells specifically when together with p53KO neighbours (Fig. 3e,f and Extended Data Fig. 3f–h). In addition, WT–mCherry cells with p53KO neighbours displayed elevated p53 protein levels, as determined by both image analysis and flow cytometry (Fig. 3g–k and Extended Data Fig. 5h–l). Besides elevated mean p53 levels, we detected increased fractions of cells with extreme p53 levels (>4.5× interquartile range) (Fig. 3j). Together, these results suggest that p53KO supercompetitors do not markedly impair their neighbours' proliferation but instead cause elevated p53 protein levels and apoptosis.

To test directly whether inhibition of mitochondria-mediated apoptosis rescued cell competition in gastruloids as had been described in 2D settings[20], we generated a WT–mCherry subclone with doxycycline-inducible expression of Bcl2, a dominant inhibitor downstream of p53, Puma and Noxa[29]. Upon doxycycline administration and Bcl2 expression, this WT–mCherry–Bcl2 clone remained cleaved caspase 3 negative and unaffected by p53KO neighbour cells (Fig. 3l–n,s and Extended Data Fig. 6a–f). Interestingly, the p53KO cell numbers increased less in the absence of WT cell apoptosis, supporting the notion of compensatory growth during competition. p53KO cells are intrinsically resistant to stress-induced apoptosis. It was thus paramount to test whether apoptosis resistance in one population was sufficient to recapitulate the competition phenotype. We therefore generated a doxycycline-inducible WT–emiRFP–Bcl2 clone. Surprisingly, WT–mCherry cells in mosaic WT + WT–Bcl2 gastruloids were entirely unaffected by the apoptosis resistance of neighbouring WT–emiRFP cells (Fig. 3o–q). Even altering seeding numbers and Bcl2 status, spanning 400–17,000 emiRFP neighbours per mosaic gastruloid at 120 h, did not affect the WT–mCherry cells (Fig. 3t,u). This indicates that WT–mCherry cells are unresponsive to the quantity and apoptosis resistance of their neighbours, and that p53KO-mediated cell competition pressure reflects a separate property of p53KO cells upstream of, or independent from, their own apoptosis resistance.

Using inducible Bcl2 expression of WT–mCherry–Bcl2 cells in WT + p53KO mosaics (Extended Data Fig. 6g), we demonstrated that continuous expression of Bcl2, or pulse induction of Bcl2 expression after 48 h of gastruloid formation, resulted in identical growth curves (Fig. 3r). This indicates that the transition stage from naive to primed pluripotency (0–48 h), is irrelevant to cell competition. Induction of Bcl2 after 72 h rescued cell competition partially, but induction of Bcl2 after 96 h did not, further supporting that cell competition is restricted to the developmental stages modelled in 48–96-h gastruloids, corresponding to the E5.5–E7.5 mouse epiblast.

Apoptosis can be initiated by intrinsic stress responses or extrinsic cell death signals, and involvement of both mechanisms in cell competition has been reported in different settings[7,30]. In the extrinsic apoptosis pathway, Fas receptor (Fas) expression peaks at 72 h but is absent in p53KO cells, and Fas ligand (Fasl) expression peaks at 96 h, being highest in p53KO cells (Extended Data Fig. 6h). However, neither knockout of Fasl in p53KO cells, nor knockout of Fas in WT–mCherry cells, the combination of both, or knockout of the critical downstream adaptor Fas associated death domain (Fadd)[31] prevented cell competition (Fig. 3v,w and Extended Data Fig. 6i,j). These results strongly support that cell competition in gastruloids is mediated by intrinsic (mitochondrial) apoptosis. Together, these results suggest that p53KO cells only marginally affect the proliferation of neighbouring WT cells, but instead induce elevated p53 protein levels followed by mitochondrial apoptosis, the inhibition of which abolishes cell competition.

**Fig. 2 | Cell competition stalls expansion of loser cells after 48 h without spatial segregation. a,b,** Representative confocal cross-sections of mosaic WT–mCherry + WT–emiRFP (**a**) and WT–mCherry + p53KO–emiRFP (**b**) gastruloids at 24–120 h after aggregation. **c**, Example rendering of 3D segmented nuclei in 48-h gastruloid. (Supplementary Fig. 3) **d**, A 3D neighbourhood analysis of segmented mosaic gastruloids. Normalized percentage of mCherry (green) or emiRFP (magenta) neighbours among the nearest 20 nuclei centred around each WT–mCherry, WT–emiRFP or p53KO–emiRFP nucleus. Mean of all nuclei from n = 4 (WT + WT_48h) or n = 8 (others) gastruloids per condition. **e**, Centre-to-edge population distribution in 3D segmented mosaic gastruloids. Normalized number of mCherry or emiRFP nuclei within an expanding shell from centre (left) to edge (right). Mean of n = 4 (WT + WT_48h) or n = 8 (others) gastruloids per condition with s.e.m. ribbons. **f,g,** Phase contrast and widefield fluorescent micrographs of representative mosaic WT + WT (**f**) and WT + p53KO (**g**) gastruloids in 24-h intervals. **h,i,** Single-gastruloid flow cytometry-derived population composition of mosaic WT + WT (**h**) or WT + p53KO gastruloids (**i**).

n = 12 (24 h) or n = 16 gastruloids (48–144 h) from 4 independent experiments, (0 h) n = 3. Individual data points from independent experiments n1, n2 and n3 are marked by round, triangular and square shapes, respectively. **j–l,** Flow cytometric analysis of neighbour phagocytosis, showing 0-h and 120-h examples and gating strategy (**j**). Dark colours indicate cells containing fluorescent material from neighbours (**k**). WT–emiRFP versus p53KO–emiRFP uptake of WT–mCherry debris (**l**) analysed using two-way ANOVA with Šidák's post-hoc test (left to right): $P_{adj} = 5 \times 10^{-4}$, $P_{adj} = 6.6 \times 10^{-3}$, $P_{adj} = 2.5 \times 10^{-2}$, $P_{adj} = 4.4 \times 10^{-3}$, $P_{adj} = 1.3 \times 10^{-3}$. Depicted as *P < 0.05, **P < 0.01, ***P < 0.001; n.s., not significant. In **k** and **l**, n = 12 gastruloids from 3 independent experiments per timepoint. **m**, Experimental workflow for quantifying absolute cell numbers per fluorescent population per gastruloid. **n**, Growth curves of WT–mCherry cells in mosaic gastruloids with 150 WT–emiRFP (blue), 25 p53KO–emiRFP (light green) or 50 p53KO–emiRFP (dark green) cells. n = 12 gastruloids from 3 independent experiments per datapoint. All data presented as mean ± s.e.m. Scale bars, 200 µm (**a** and **b**) and 500 µm (**f** and **g**).

## Influence of physiological, metabolic and signalling parameters on cell competition

To understand how cell competition is communicated, we first tested whether it was perceived over long-range, by coculturing multiple gastruloids in adjacent microwells with shared medium. Monoclonal WT–mCherry gastruloids grew unhindered, independent of whether surrounded by six WT–emiRFP, p53KO–emiRFP or mosaic WT + p53KO gastruloids from 48 to 120 h (Fig. 4a,b), suggesting that cell competition signals are either acting through direct contact or were not sufficiently concentrated in this setting.

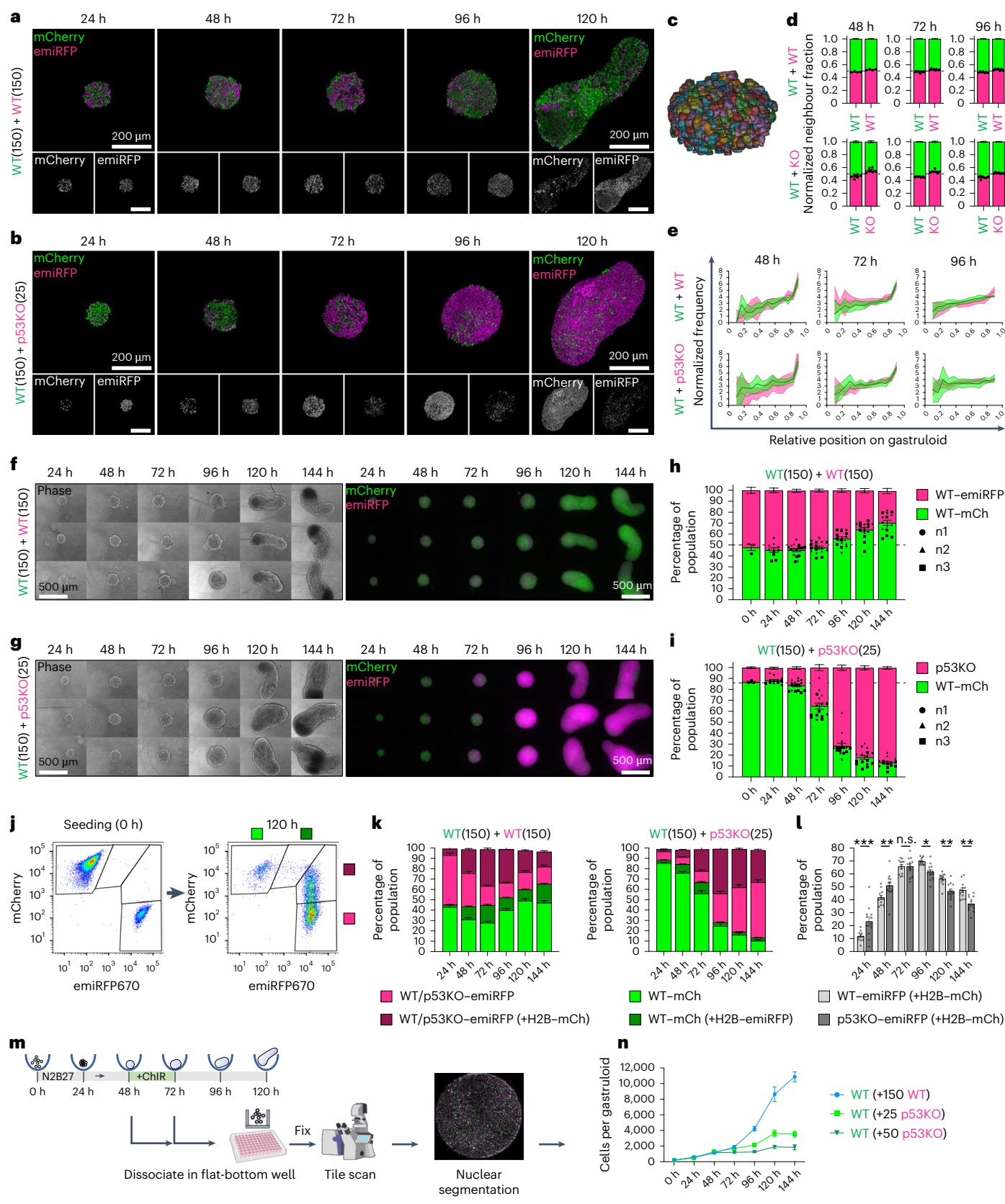

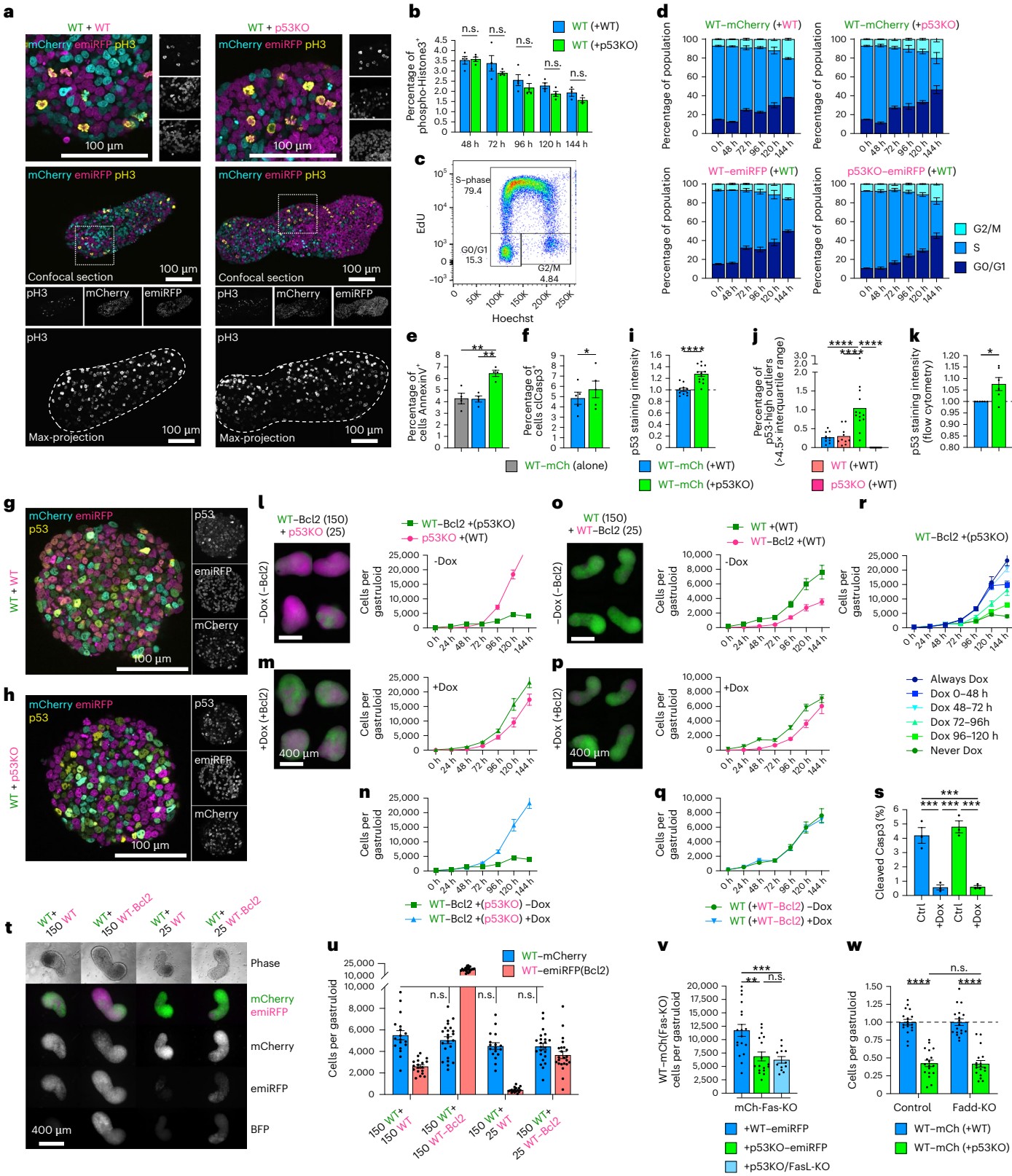

Following this, we conducted 3D segmentation and neighbourhood analysis of cleaved caspase 3-stained apoptotic events in mosaic gastruloids and found that apoptotic WT–mCherry cells were surrounded by more p53KO cells than were viable cells at 72–96 h (Extended Data Fig. 7a–c). Furthermore, cells initiating apoptosis were biased towards central regions of gastruloids (Extended Data Fig. 7d,e), which might resemble increased resource scarcity and compressed

mechanical environments. To alter the gastruloid core size, we seeded mosaic gastruloids with 200%, 50% or 25% of the normal starting cell number (600, 150 and 75 cells). Developmental processes in gastruloids scale with size within certain limits[32,33], and we found that the gastruloid size and absolute cell numbers scaled linearly with the number of initially seeded cells (Fig. 4c–f and Extended Data Fig. 7f,g). However, the relative effect size of cell competition exerted by p53KO

**Fig. 3 | Cell competition in gastruloids occurs through mitochondrial apoptosis without altering cell cycle progression. a**, Representative confocal micrographs of phospho-histone3 (pH3) immunostained gastruloids at 120 h, with the corresponding pH3 maximum-intensity projection shown below. **b**, Flow cytometric quantification of pH3-positive WT–mCherry cells in WT + WT or WT + p53KO gastruloids. **c,d**, Representative example (**c**) and quantification (**d**) of EdU/Hoechst cell cycle profiling in mosaic gastruloids. **e,f**, Percentage of Annexin-V (**e**) and clCaspase-3-positive (**f**) WT–mCherry cells in 72-h mosaic gastruloids. In **e**, top to bottom: $P_{adj} = 3.8 \times 10^{-3}$, $P_{adj} = 3.3 \times 10^{-3}$. In **f**: $P = 3.2 \times 10^{-2}$. **g,h**, Representative p53 immunostaining at 72 h (Supplementary Fig. 2). **i–k**, p53 levels (**i**) and frequency of p53-extreme high (**j**) WT–mCherry cells in 72-h mosaic gastruloids measured by image analysis and flow cytometry (**k**). In **i**: $P < 1 \times 10^{-4}$. In **j**, left to right: $P_{adj} < 1 \times 10^{-4}$, $P_{adj} < 1 \times 10^{-4}$, $P_{adj} < 1 \times 10^{-4}$. In **k**, $P = 2.3 \times 10^{-2}$. **l–n**, Representative 120-h gastruloids containing p53KO–emiRFP and WT–mCherry(inducible-Bcl2) cells without (**l**) or with (**m**) doxycycline with quantified growth curves. WT–mCherry(inducible-Bcl2) curves with and without doxycycline (**n**). **o–q**, As **l–n**, but from WT–mCherry and WT–emiRFP-Bcl2 mosaics. **r**, Temporally controlled *Bcl2* overexpression

in WT–mCherry(inducible-Bcl2) cells in WT + p53KO gastruloids. **s**, Flow cytometry of clCaspase-3-positive WT–mCherry(inducible-Bcl2) cells in 72-h mosaic gastruloids. Top, left to right: $P_{adj} = 5 \times 10^{-4}$, $P_{adj} = 4 \times 10^{-4}$, $P_{adj} = 2 \times 10^{-4}$, $P_{adj} = 2 \times 10^{-4}$. **t,u**, Representative micrographs (**t**) and cell counts (**u**) of 120-h gastruloids containing WT–mCherry and WT–emiRFP(±inducible-Bcl2) cells. **v**, Quantified mCherry–Fas-knockout cells at 120 h in mosaic gastruloids with WT–emiRFP, p53KO–emiRFP, or p53KO–FasLKO–emiRFP cells. Top to bottom: $P_{adj} = 9 \times 10^{-4}$, $P_{adj} = 1.2 \times 10^{-3}$. **w**, Normalized WT–mCherry cell counts in 120-h gastruloids with p53KO–emiRFP cells ±Fadd knockout. Left to right: $P_{adj} < 1 \times 10^{-4}$, $P_{adj} < 1 \times 10^{-4}$. All data presented as mean ± s.e.m. For pooled analyses: $n = 3$ (**s**), $n = 4$ (**b**, **d** and **e**), $n = 5$ (**f**), $n = 6$ (**k**) independent experiments. For per-gastruloid analyses: $n = 9$ (**l–n** and **r**), $n = 12$ (**v**), $n = 13$ (**i** and **j**), $n = 17$ (**u**), $n = 18$ (**w**) from 3 (**i,j,l–n, r, v** and **w**) or 4 (**u**) independent experiments each. Statistical analysis by two-way ANOVA with Šidák's post-hoc test (**b, s, u** and **w**), one-way ANOVA with Tukey's post-hoc test (**e, j** and **v**) or paired (**f**) or unpaired two-tailed *t*-test (**i** and **k**), depicted as *$P < 0.05$, **$P < 0.01$, ***$P < 0.001$, ****$P < 0.0001$; n.s., not significant. Scale bars, 100 μm (**a** and **b**) and 400 μm (**l, m, o, p** and **t**).

cells as measured by the ratio of WT–mCherry cells in WT + WT versus WT + p53KO mosaics, was unaffected by the eightfold difference in gastruloid size (Fig. 4e). Similarly, the temporal kinetics of competition were unaffected by gastruloid size (Fig. 4g). This confirmed that, different from 2D systems, the onset of cell competition is determined by a developmental stage rather than a cell number threshold.

Mechanical cell crowding and YAP play important roles in 2D epithelial cell competition and *Drosophila*[16,19,34]. We thus investigated the local cell densities around apoptotic events by 3D image segmentation. WT + p53KO mosaic gastruloids start at lower cell densities but increase their density to match that of WT + WT gastruloids at 96 h after aggregation (Fig. 4h). Early apoptotic WT–mCherry cells in WT + p53KO gastruloids are surrounded by higher cell densities than viable cells in the same gastruloid or cells in WT + WT gastruloids (Fig. 4i and Extended Data Fig. 7h,i). However, we detected no differences in the nuclear-to-cytosolic YAP ratio in WT–mCherry cells at 72 h independent of their neighbours' p53 status (Fig. 4j and Extended Data Fig. 7j,k). Furthermore, treatment with Rho-associated kinase inhibitors (ROCKi, Y-27632), cytochalasin D or blebbistatin at the highest sublethal concentrations did not significantly alter competition (Fig. 4k,l and Extended Data Fig. 7l,m).

Loss of the tumour suppressor p53 has been linked to higher metabolic activity and a shift towards glycolysis (Warburg effect)[35,36]. To test the influence of glucose availability on cell competition, we generated gastruloids in a pyruvate-poor N2B27 medium with varying glucose concentrations (Fig. 4m,n and Extended Data Fig. 7n,o). While the competition effect size in this home-made medium was less pronounced than in commercial medium, lowering glucose concentrations eightfold from 21 mM to 2.5 mM did not affect cell competition (Fig. 4n), despite lower proliferation of p53KO cells. In this

range, cell competition was thus also independent of the p53KO cell growth rate. Lowering glucose levels further to 0.5 mM appeared to abolish cell competition, but might be caused by the p53KO cells' higher sensitivity to glucose limitation compared with WT cells (Fig. 4m and Extended Data Fig. 7o). Interestingly, cell competition effect size was similarly decreased when inhibiting glycolysis (6 mM 2-deoxy-glucose) or oxidative phosphorylation (1 mM $NaN_3$) in mosaic gastruloids from 24 to 48 h, suggesting a general sensitivity to metabolic perturbation in p53KO cells rather than a glycolysis-specific effect (Fig. 4o,p).

Together, these data suggest that cell competition in this system acts at short range, correlates with denser p53KO-rich neighbourhoods towards the gastruloid centre, is independent of gastruloid size and cannot be explained by YAP/mechanical signalling or glucose limitation.

## Wnt and BMP signals protect from cell competition

Defects or local mismatches in BMP and Wnt signalling activity can create loser cell identities[6,12,13], and treatment of pluripotent stem cells with anteriorizing and posteriorizing signals can influence their developmental fidelity[37]. We thus tested whether modulating developmental signalling pathway activities, and thereby the developmental context, would influence cell competition in mosaic gastruloids. Wnt agonism through ChIR treatment, as done in default gastruloids, surprisingly suppressed cell competition when compared with Wnt inhibition (Fig. 5a and Extended Data Fig. 8a,b). Similarly, BMP4 treatment lowered the amount of cell competition compared with BMP inhibition by LDN193189 treatment (Fig. 5b). By contrast, modulation of Activin-A/Nodal/TGFb signalling by Activin A or SB431542 treatment as well as ERK signalling activation or inhibition by FGF2 or PD0325901 did not influence cell competition (Fig. 5c,d). When comparing all

**Fig. 4 | Cell competition acts over short ranges and is not influenced by gastruloid size, glucose availability or cytoskeletal perturbations. a,b**, Representative micrographs (**a**) and WT–mCherry cell counts (**b**) from contact-free microcavity cocultures. **c**, Representative 120-h gastruloids seeded from 600, 150 or 75 total cells. **d,e**, Absolute (**d**) and normalized (**e**) WT–mCherry cell counts in mosaic 120-h WT + WT (blue) or WT + p53KO (green) gastruloids. All *P* values annotated as '****' signify $P_{adj} < 1 \times 10^{-4}$. **f**, Sum growth curves all cells in WT + WT gastruloids normalized to initial seeding numbers ($N_t/N_0$). **g**, Normalized WT–mCherry growth curves in WT + WT (blue) and WT + p53KO (green) mosaic gastruloids from varying seeding numbers. **h,i**, Mean local density of closest 20 nuclei surrounding viable (**h**) and apoptotic (**i**) nuclei in mosaic gastruloids. Left to right: $P_{adj} = 3.6 \times 10^{-2}$, $P_{adj} = 3.4 \times 10^{-2}$. **j**, Nuclear-to-cytosolic ratio of YAP staining intensity in WT cells from 72 h WT + WT or WT + p53KO gastruloids. **k,l**, Absolute (**k**) and normalized (**l**) cell counts from mosaic 120-h gastruloids treated with cytoskeletal and mechanoperception

inhibitors (48–96 h). In **l**, all *P* values annotated as '****' signify $P_{adj} < 1 \times 10^{-4}$. **m,n**, Absolute (**m**) and normalized (**n**) cell counts from mosaic 120-h gastruloids grown in pyruvate-poor, glucose-limited medium. In **n**, top to bottom, left to right: $P_{adj} = 1 \times 10^{-4}$, $P_{adj} < 1 \times 10^{-4}$, $P_{adj} = 2.2 \times 10^{-3}$, $P_{adj} < 1 \times 10^{-4}$, $P_{adj} < 1 \times 10^{-4}$, $P_{adj} < 1 \times 10^{-4}$. **o,p**, Absolute (**o**) and normalized (**p**) cell counts from mosaic 120-h gastruloids treated with glycolysis or oxidative phosphorylation inhibitors (24–48 h). In **p**, all *P* values annotated as '****' signify $P_{adj} < 1 \times 10^{-4}$. For per-gastruloid analyses: $n = 4$ (**h-i_48h**), $n = 8$ (**h-i_72–96h**), $n = 12$ (**d–g**), $n = 18$ (**k, l, o** and **p**), $n = 22$ (**b**), $n = 24$ (**m** and **n**), from 3 (**b, d–g, k, l, o** and **p**) or 4 (**m** and **n**) independent experiments. $n = 984$ and $n = 1,819$ nuclei from 3 and 4 gastruloids (**j**). All data presented as mean ± s.e.m. Statistical analysis by two-way ANOVA with Šidák's post-hoc test (**e, h, i, l, n** and **p**), one-way ANOVA with Tukey's post-hoc test (**b**) or two-sided unpaired *t*-test (**j**) and is depicted as *$P < 0.05$, **$P < 0.01$, ***$P < 0.001$, ****$P < 0.0001$; n.s., not significant. Scale bars, 1,000 μm (**a**) and 500 μm (**c**).

signalling modulations side by side, we found that ChIR treatment was the most protective, while absence of any treatment (dimethyl sulfoxide (DMSO) control) permitted some of the most pronounced competition (Fig. 5e–g). This absence of ChIR treatment did, however, result in poorly organized gastruloids that lacked *Tbx6* expression and showed elevated levels of *Sox2*, *Otx2* and *Sox3*, indicating lack or delay of paraxial mesoderm development, and in the p53KO gastruloids a tendency towards more neuronal tissues (Extended Data Fig. 8c,d). Together, we observed that Wnt and BMP agonism, but not ERK and

Nodal modulation, prevent cell competition, while the absence of posteriorizing signals is permissive for more extensive cell competition.

## Cell competition is differentiation stage specific

At 48 h after aggregation, gastruloids have exited naive pluripotency and entered a formative-like state (Fig. 6a,b). The standard gastruloid protocol calls for a ChIR (CHIR99021) treatment (Wnt agonism) at 48–72 h after aggregation (Fig. 1c), which coincides with the onset of cell competition and gastrulation. Recent work demonstrated that this Wnt

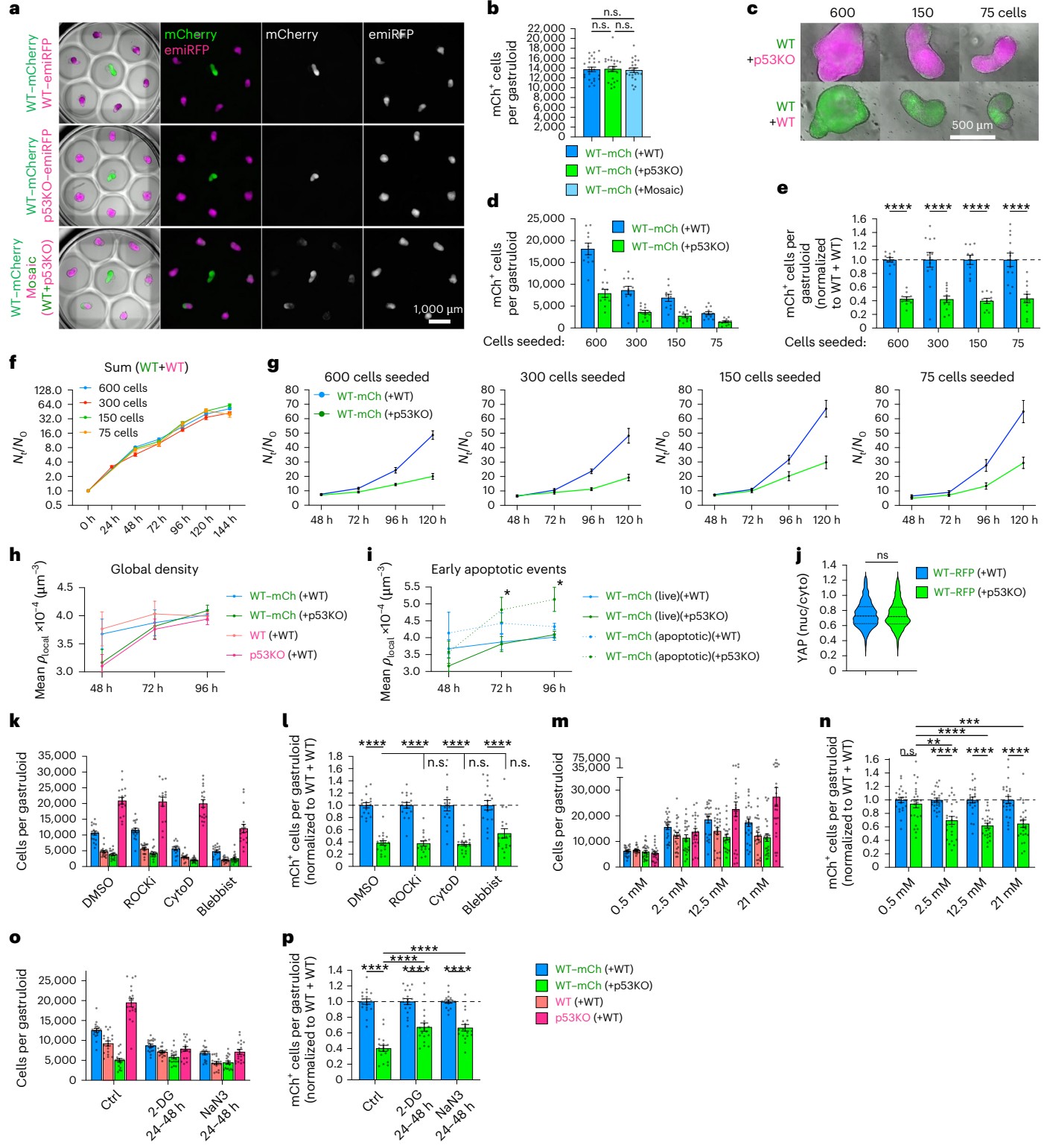

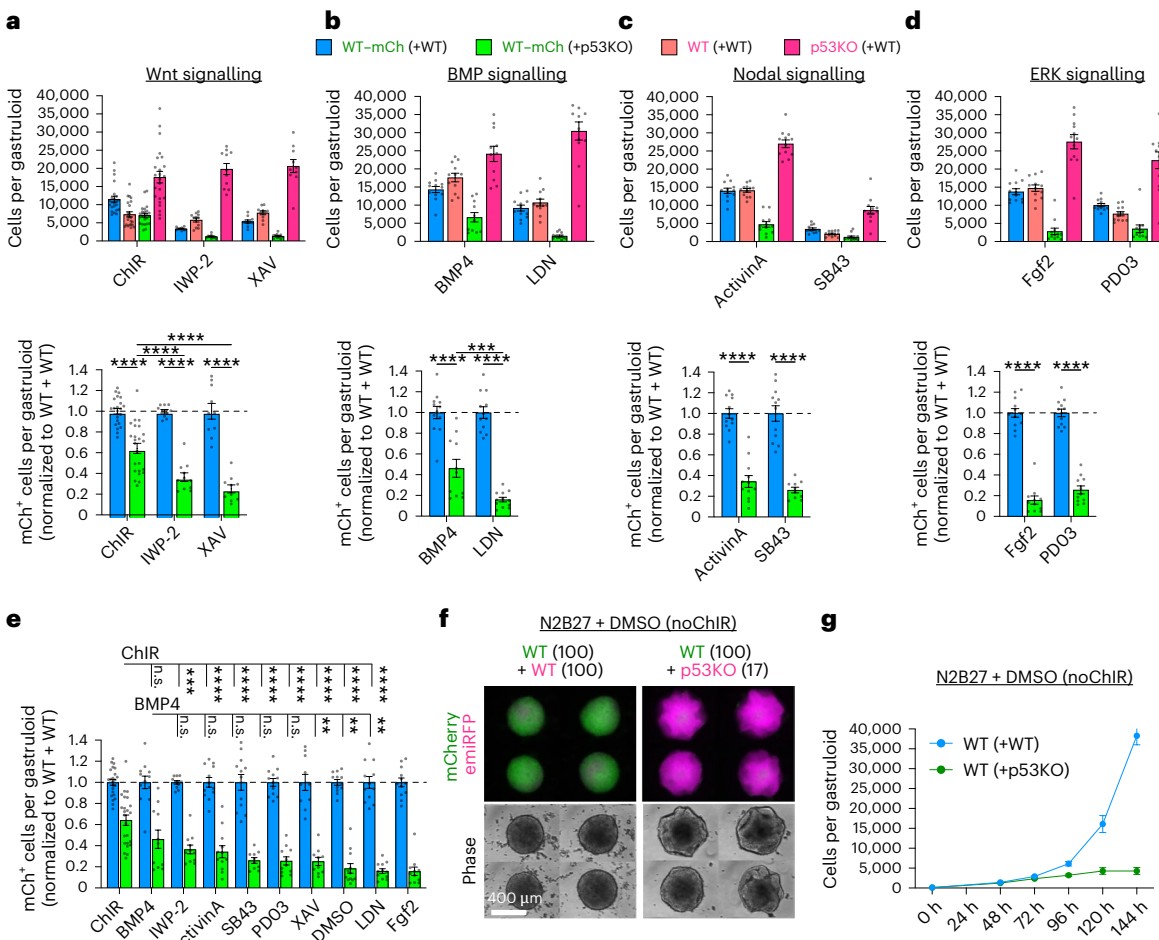

**Fig. 5 | Wnt and BMP signalling, but not ERK or Nodal signalling, protect from cell competition. a–d**, Absolute (top) and normalized (bottom) cell counts in 120-h mosaic WT + WT and WT + p53KO gastruloids treated with 3 μM ChIR (48–72 h), 1 μM IWP-2 (48–120 h) or 1 μM XAV-939 (48–120 h) (**a**), 1 ng ml⁻¹ BMP4 (48–120 h) or 100 nM LDN193189 (48–120 h) (**b**), 100 ng ml⁻¹ Activin A (48–72 h) or 10 μM SB431542 (48–120 h) (**c**), 12.5 ng ml⁻¹ FGF2 (48–120 h) or 1 μM PDO325901 (48–120 h) (**d**). Statistics displayed only for normalized cell competition effect size. All *P* values annotated as '****' signify $P_{adj} < 1 \times 10^{-4}$. '***' signifies $P_{adj} = 9 \times 10^{-4}$. **e**, Comparative overview of normalized WT–mCherry cell counts in all developmental signal perturbations and DMSO control treatment. Only statistical comparisons among the WT–mCherry(+p53KO) counts (effect size of competition) are depicted. All *P* values annotated as

'****' signify $P_{adj} < 1 \times 10^{-4}$. ChIR versus IWP-2 $P_{adj} = 8 \times 10^{-4}$, BMP4 versus DMSO $P_{adj} = 7.4 \times 10^{-3}$, BMP4 versus LDN $P_{adj} = 1.9 \times 10^{-3}$, BMP4 versus Fgf2 $P_{adj} = 2.3 \times 10^{-3}$. In **a–e**, $n = 24$ (ChIR) or $n = 12$ (all other treatments) gastruloids from 6 (ChIR) or 3 (all others) independent experiments. **f**, Representative micrographs of 120-h mosaic WT + WT (left) and WT + p53KO (right) gastruloids grown without ChIR treatment. Scale bar, 400 μm. **g**, Absolute WT–mCherry growth curves in mosaic WT + WT or WT + p53KO gastruloids without ChIR treatment (DMSO control). $n = 9$ gastruloids per datapoint from 3 independent experiments. All data presented as mean ± s.e.m. Statistical analysis by two-way ANOVA with Šidák's post-hoc test, depicted as *$P < 0.05$, **$P < 0.01$, ***$P < 0.001$, ****$P < 0.0001$; n.s., not significant.

signalling pulse not only drives primitive streak development but also accelerates its temporal progression[38]. Flow cytometry analysis of the pluripotency surface markers SSEA-1 and Pecam1 supports this observation (Fig. 6c,d and Extended Data Fig. 9a,b), suggesting that omission of ChIR treatment may prolong the cell-competition-permissive stage in gastruloids (Fig. 5e–g). Flow cytometry also revealed higher SSEA-1 and Pecam1 levels in p53KO cells compared with WT cells, which could indicate a delay in the exit of pluripotency (Fig. 6c and Extended Data Fig. 9b). At the whole-transcriptome level, all populations from mosaic gastruloids follow similar temporal trajectories, as estimated by their Euclidean distances to timed embryonic transcriptomes (Fig. 6e), while principal component analysis clustered all clones closely together until 72 h, after which p53KO cells diverge (Fig. 6f). To test whether a developmental delay in p53KO cells could be related to their competitive advantage, we created heterochronic mosaic gastruloids by first generating pure mCherry or emiRFP gastruloids, dissociating them at different times and reaggregating them as mosaics centred around the 48-h timepoint of WT–mCherry

cells (Fig. 6g). Introducing WT–emiRFP cells from 24-h gastruloids into WT–mCherry 48-h gastruloids did not impair their growth, but rather supported it, indicating that a relative difference in pluripotency/developmental stage between two WT clones does not cause cell competition (Fig. 6h,i). Reintroducing p53KO cells from 72-h gastruloids into 48-h WT gastruloids caused almost identical cell competition pressure on the WT cells as reintroducing homochronic 48-h p53KO cells (Fig. 6j). By expanding our panel of heterochronic shifts, we observed that 96-h p53KO cells no longer exert competition pressure on neighbouring 48-h WT–mCherry cells, and that temporally shifted 72-h WT–mCherry cells no longer receive competition stress from 48-h p53KO cells (Fig. 6k–m and Extended Data Fig. 9c–h). Taken together, relative differences of pluripotency are neither necessary nor sufficient to cause cell competition. Furthermore, cell competition is only executed or communicated when both competing populations reside within a developmental stage permissive for cell competition, which is centred around 72 h, and can be shortened or prolonged by Wnt agonism in gastruloids.

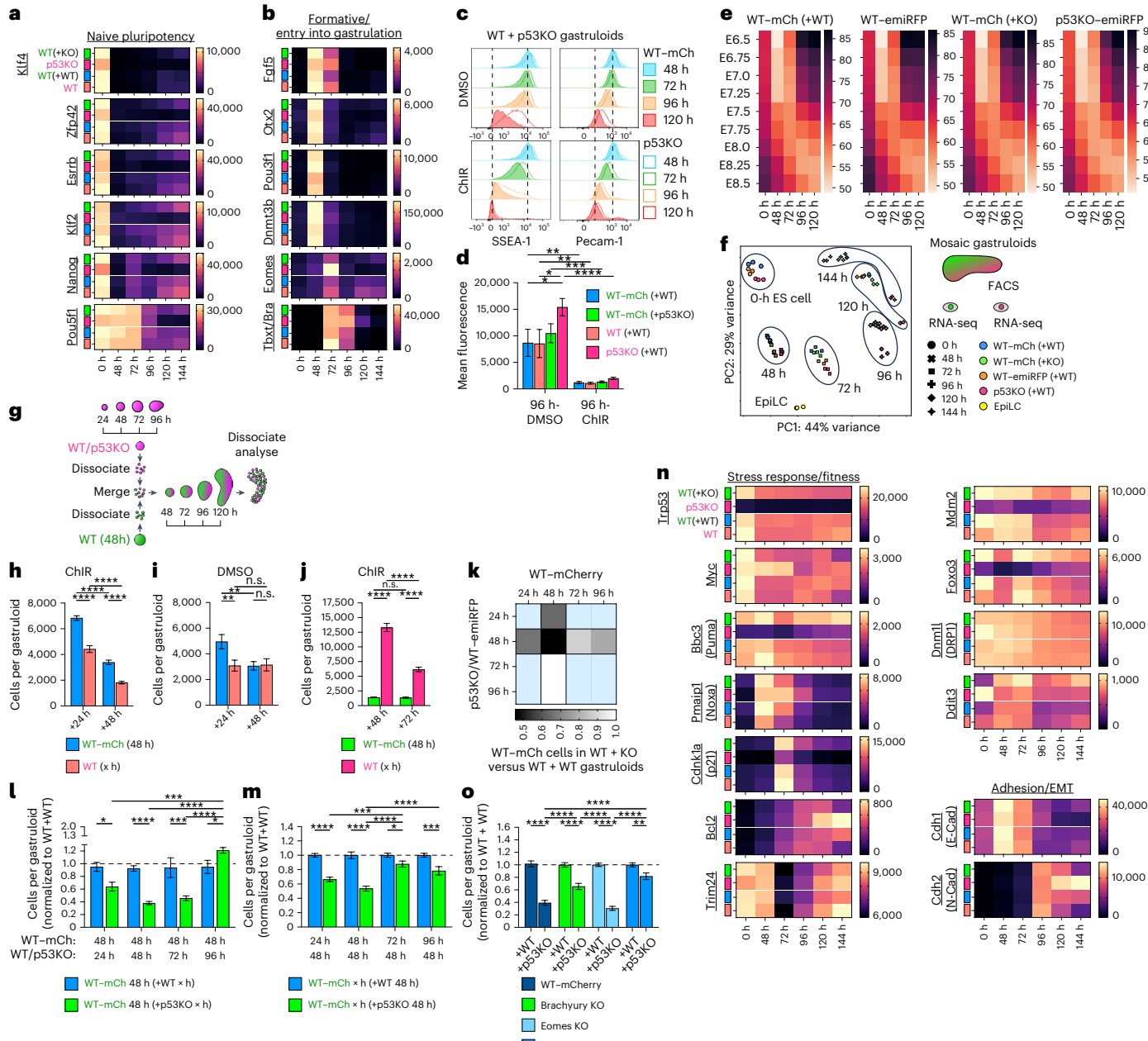

**Fig. 6 | Cell competition occurs exclusively between primed pluripotency and gastrulation onset. a**,**b**, Gene expression dynamics of markers of naive pluripotency (**a**), formative pluripotency (**b**) or entry into gastrulation (**b**) in FACS-isolated cells from mosaic gastruloids. **c**, Flow cytometry detecting surface SSEA-1 and Pecam1 in WT−mCherry (filled) and p53KO−emiRFP (open lines) cells from ChIR- or DMSO-treated mosaic gastruloids at 48–120 h. **d**, Median SSEA-1 surface expression at 96 h. Top to bottom, left to right: $P_{adj} = 2.5 \times 10^{-3}$, $P_{adj} = 2.5 \times 10^{-3}$, $P_{adj} = 4 \times 10^{-4}$, $P_{adj} = 2.3 \times 10^{-2}$, $P_{adj} < 1 \times 10^{-4}$, $P_{adj} = 2.7 \times 10^{-2}$. **e**, Euclidean distance between transcriptomes of cells from mosaic gastruloids and timed embryonic references. Brighter colour indicates shorter distance. **f**, Principal component analysis of FACS-separated mosaic gastruloids at 48–144 h, mES cells before aggregation, and EpiLCs. **g**, Schematic of heterochronic mosaic gastruloid formation. **h**–**j**, Cell counts from 120-h heterochronic mosaic gastruloids reaggregated from 48-h WT−mCherry cells with WT−emiRFP (**h** and **i**) or p53KO−emiRFP (**j**) cells from indicated timepoints. In **h** and **j**, all annotated P values annotated as '****' signify $P_{adj} < 1 \times 10^{-4}$. In **i**, top to bottom: $P_{adj} = 5.1 \times 10^{-3}$, $P_{adj} = 5.6 \times 10^{-3}$. **k**–**m**, Summary heatmap (**k**) and normalized counts (**l** and **m**) from 120-h heterochronic mosaic gastruloids. Darker shading indicates stronger competition measured by the ratio of WT−mCherry cells in WT + KO to WT + WT gastruloids. Light-blue conditions not tested. Conducted in home-made N2B27 medium. All P values top to bottom and left to right, omitting '****', which signifies $P_{adj} < 1 \times 10^{-4}$ (**k**), $P_{adj} = 1 \times 10^{-4}$, $P_{adj} = 1.1 \times 10^{-2}$, $P_{adj} = 1 \times 10^{-4}$, $P_{adj} = 3.6 \times 10^{-2}$. In **m**, $P_{adj} = 6 \times 10^{-4}$, $P_{adj} = 3.5 \times 10^{-2}$, $P_{adj} = 6 \times 10^{-4}$. **n**, Gene expression dynamics of fitness, stress response and epithelial-to-mesenchymal transition (EMT) (as **a** and **b**). **o**, Normalized cell counts of WT, BraKO, EomesKO and Bra/Eomes dKO cells in gastruloids with WT or p53KO neighbours. All P values annotated as '****' signify $P_{adj} < 1 \times 10^{-4}$; ***$P_{adj} = 2.4 \times 10^{-3}$. For pooled experiments: $n = 3$ (**a**, **e** and **n**), $n = 4$ (**c** and **d**) independent experiments. For per-gastruloid quantifications: $n = 12$ (**h**–**j** and **l** + KO), $n = 16$ (**l** + WT), $n = 22$ (**o**_EomesKO), $n = 24$ (**m**), $n = 28$ (**o**_BraKO), $n = 33$ (**o**_dKO), $n = 44$ (**o**_WT-mCh) gastruloids from 3 (**h**–**j**) or 4(**k**–**m**) or 6 (**o**) independent experiments. All data presented as mean ± s.e.m. Statistical analysis by two-way ANOVA with Šidák's post-hoc test and is depicted as *$P < 0.05$, **$P < 0.01$, ***$P < 0.001$, ****$P < 0.0001$; n.s., not significant.

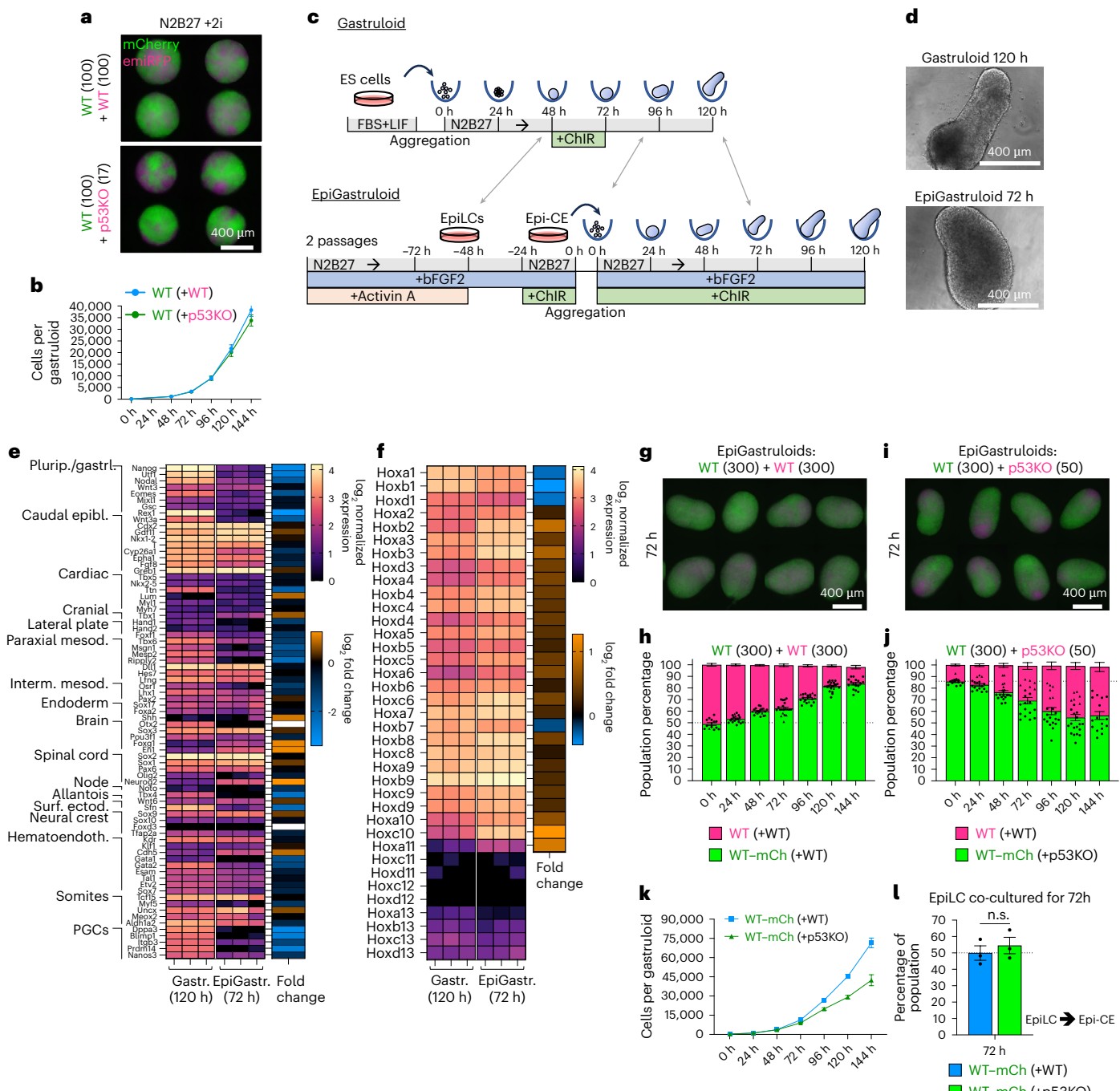

**Fig. 7 | 3D aggregates outside of the permissive developmental stage do not undergo cell competition. a**, Representative widefield micrographs of pluripotent mosaic 3D aggregates at 120 h after aggregation. **b**, WT–mCherry cell growth curves in mosaic pluripotent aggregates with WT–emiRFP or p53KO–emiRFP neighbours. $n$ = 12 gastruloids per timepoint from 3 independent experiments **c,d**, Schematic of mouse gastruloid and EpiGastruloid formation protocols (**c**) and representative micrographs at 120 h (**d**). Grey arrows mark tentative developmental stage equivalence. **e,f**, Developmental lineage marker (**e**) and Hox gene (**f**) expression profiles from bulk transcripts of 120-h WT mouse gastruloids and 72-h WT mouse EpiGastruloids. $n$ = 3 independent experiments. log$_2$ fold change depicted in adjacent blue–orange heatmaps.

**g–j**, Fluorescent widefield micrographs (72 h) (**g** and **i**) and flow cytometric quantification of population composition (**h** and **j**) of WT + WT (**g** and **h**) and WT + p53KO (**i** and **j**) EpiGastruloids. $n$ = 13–20 EpiGastruloids per timepoint from 4 independent experiments. **k**, Absolute growth curves of WT–mCherry cells in mosaic EpiGastruloids seeded together with 300 WT–emiRFP (blue), or 50 p53KO–emiRFP (green) cells. $n$ = 17 gastruloids per datapoint from 3 independent experiments. **l**, Percentage of WT–mCherry cells after cotransitioning from EpiLC to EpiCE together with WT–emiRFP (blue) or p53KO–emiRFP (green) cells. $n$ = 3 independent experiments. All values depicted as mean ± s.e.m. Scale bars, 400 μm.

To interrogate the temporal regulation of competition, we transcriptionally profiled stress and apoptosis regulators across time. Consistent with previous work[39], mosaic gastruloids demonstrate DRP1 (*Dnm1l*) downregulation and pro-apoptotic Puma (*Bbc3*) upregulation

during primed pluripotency[15] (Fig. 6n). We also observe an even more acute early upregulation of pro-apoptotic Noxa (*Pmaip1*) at 48 h (Fig. 6n), unexpectedly even in p53KO cells, followed by p21 (*Cdkn1a*) at 72 h, marking the temporal centre of cell competition, and

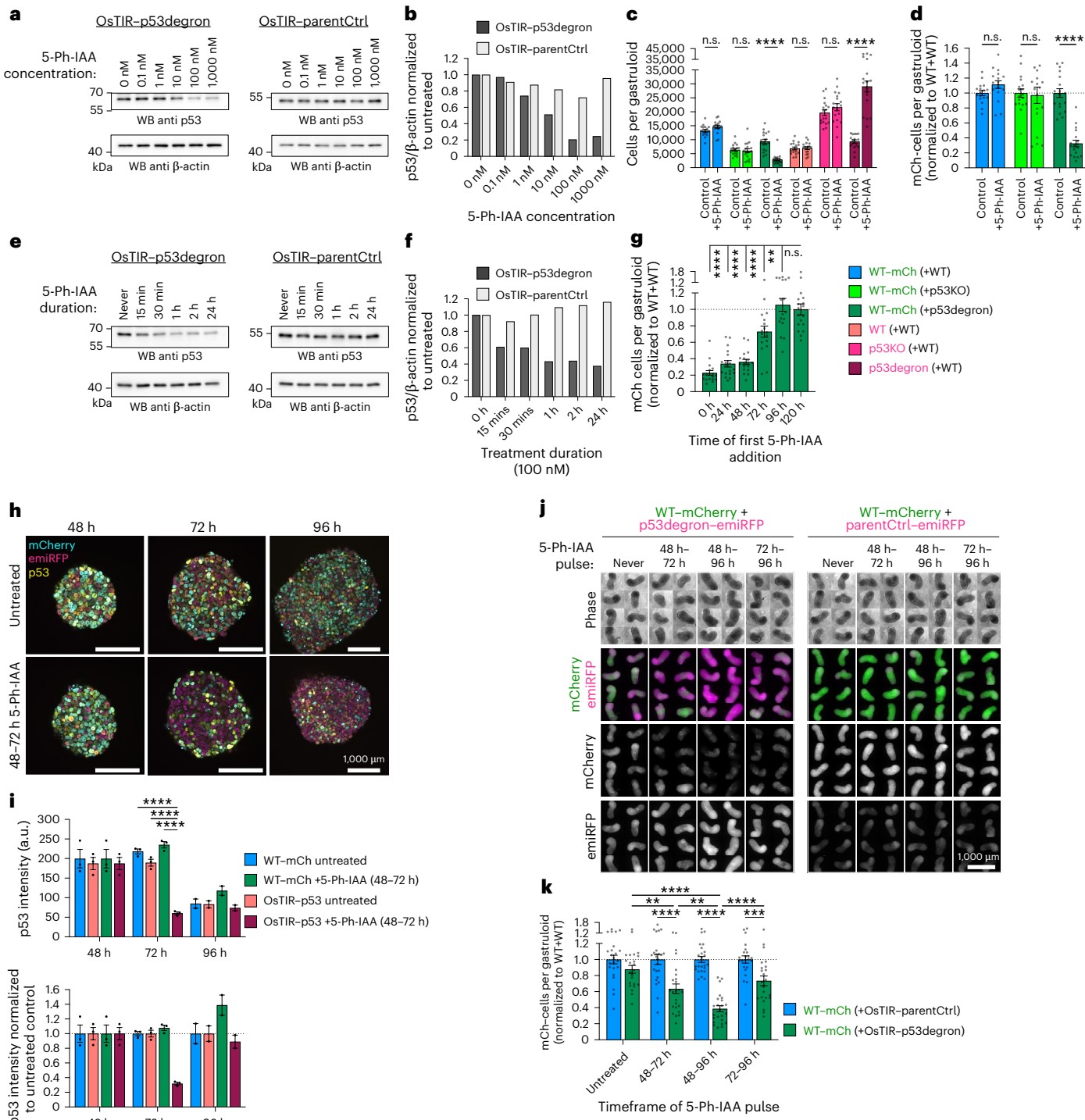

**Fig. 8 | Transient reduction of p53 protein at the onset of gastrulation acutely causes supercompetition. a,b**, Western blot (WB) confirmation of 5-Ph-IAA dose-dependent p53 degradation in OsTIR1-based p53-degron or OsTIR control clone (**a**), and densitometric ratio (**b**) of p53 and β-actin loading control. **c,d**, Absolute (**c**) and normalized (**d**) cell counts in 120-h mosaic WT + WT, WT + p53KO or WT + p53degron gastruloids treated with 1,000 nM 5-Ph-IAA, or DMSO control. '****' signifies $P_{adj} < 1 \times 10^{-4}$. **e,f**, Western blot analysis of p53 degradation time course after 100 nM 5-Ph-IAA treatment in p53-degron or OsTIR control clone (**e**), and densitometric ratio (**f**) of p53 and β-actin loading control. **g**, WT–mCherry cell counts in 120-h mosaic WT + p53degron gastruloids with varying starting times of 5-Ph-IAA treatment normalized to untreated (120 h) control. Statistical comparison of all conditions to untreated '120 h' reference. '****' signifies $P_{adj} < 1 \times 10^{-4}$, $**P_{adj} = 3.6 \times 10^{-3}$. **h**, Representative Z-slices of confocal stacks of p53-immunostained WT + p53degron gastruloids at indicated times with or without 48–72 h 100 nM 5-Ph-IAA treatment. **i**, Image analysis-based

quantification of p53 levels in mosaic degron gastruloids (as **h**). Each datapoint represents the depth-corrected mean intensity of all cells within one gastruloid. $n = 3$ (48–72 h) or $n = 2$ (96 h) gastruloids per condition and timepoint. P53 intensities normalized to untreated gastruloids below. '****' signifies $P_{adj} < 1 \times 10^{-4}$. **j**, Short-term reversible p53 degradation in WT–mCherry +OsTIR–p53degron or +OsTIR–parentCtrl mosaic gastruloids. Time annotations indicate addition and washout of 100 nM 5-Ph-IAA. **k**, Normalized WT–mCherry cell counts from 120-h mosaic gastruloids with short-term degron induction as depicted in **j**. '****' signifies $P_{adj} < 1 \times 10^{-4}$. Left to right: $P_{adj} = 5.8 \times 10^{-3}$, $P_{adj} = 3.9 \times 10^{-3}$, $P_{adj} = 3 \times 10^{-4}$. All data presented as mean ± s.e.m. $n = 18$ (**c**, **d** and **g**) or $n = 24$ (**k**) gastruloids from 3 (**c**, **d** and **g**) or 4 (**k**) independent experiments. Data in **j** are representative of four independent experiments. Statistical analysis by two-way ANOVA with Šidák's post-hoc test (**c**, **d**, **i** and **k**) or one-way ANOVA with Dunnet's post-hoc test (**g**) depicted as *$P < 0.05$, **$P < 0.01$, ***$P < 0.001$, ****$P < 0.0001$; n.s., not significant. Scale bars denote 1,000 μm.

lastly anti-apoptotic *Bcl2* upregulation from 96 h onwards, coinciding with gastrulation and an E-to-N-Cadherin switch. The 72-h centre of cell competition also aligns with sharp downregulation of *Trim24*, an epigenetic repressor of p53 targets[40]. This suggests a temporally regulated transcription programme preparing the stress response machinery at primed pluripotency, releasing it at the onset of gastrulation and restraining it by mid-gastrulation.

The transcription factors Brachyury and Eomesodermin are key regulators governing the transition from primed pluripotency to gastrulation in the posterior and anterior primitive streak respectively[41,42]. We tested whether their deletion interrupted the cell competition machinery. When exposed to p53KO cells in mosaic gastruloids, Brachyury-knockout (BraKO) but not Eomesodermin-knockout (EomesKO) cells responded less than WT control cells (Fig. 6o). Brachyury/Eomesodermin double-knockout (dKO) cells were almost entirely insensitive to neighbouring supercompetitor cells (Fig. 6o). dKO cells lose the ability to specify mesendodermal fates and remain in primed pluripotency until neuroectodermal differentiation[41,42]. Despite low Myc levels, dKO cells did not express Puma, Noxa or p21 at 72 h of gastruloid formation, but instead express high levels of Bcl2 (Extended Data Fig. 9i). Together, these findings suggest either that the cell competition and stress machinery at the onset of gastrulation depends on Brachyury's transcriptional programme, or that the combined absence of Brachyury and Eomesodermin causes cells to bypass the developmental window permissive for cell competition.

The need for competing cells to be in this permissive stage is reinforced by the observation that p53KO and WT cells do not compete when they grow in naive pluripotent 3D aggregates, or in EpiGastruloids, a 3D developmental model resembling posterior development of a slightly later stage than classical gastruloids (Fig. 7 and Extended Data Fig. 10).

### Dynamic and reversible p53 protein levels at the onset of gastrulation elicit cell competition

Having identified a developmental window for cell competition, we tested whether fitness sensing through p53 protein levels was truly a dynamic and acute process. For this, we developed a p53-degron that allows rapid temporal control over p53 protein levels by degron induction via 5-phenyl-indole-3-acetic acid (5-Ph-IAA) addition (Methods; Fig. 8a–f). By initiating the p53 degradation at different times in WT + Degron mosaic gastruloid formation, we observed that no additional cell competition is gained when degrading p53 earlier than 48 h, and no competition was observed when p53 protein was degraded starting at 96 h (Fig. 8g). Using the reversible nature of this system and confocal image analysis, we confirmed that degron induction resulted in degradation of p53 protein even in deep layers of gastruloids, which could be fully restored after 5-Ph-IAA washout (Fig. 8h,i and Supplementary Fig. 2). Dynamic reduction in p53 protein in OsTIR–p53-degron cells was followed by an increase of p53 levels in neighbouring WT–mCherry cells. We tested different times and durations of p53 degradation in mosaic gastruloids and observed that reduction of p53 protein levels for 24 h from 48 to 72 h or 72 to 96 h both caused cell competition against neighbouring WT cells, but that a 48–96-h pulse resulted in the maximal response (Fig. 8j,k). These results demonstrate that the acute and dynamic p53 protein levels during the onset of gastrulation determine cell competition outcomes.

## Discussion

We establish mouse gastruloids as a 3D model system to study mammalian developmental cell competition. We find that competition is restricted to a narrow window at the onset of gastrulation (48–96 h in gastruloids, equivalent to E5.5–E7.5 mice), and heterochronic mosaic gastruloids reveal that both winner and loser cells outside this permissive stage are unable to participate in cell competition. Transient and reversible reductions of p53 protein are sufficient to generate supercompetitor states, providing an experimental validation that acute relative p53 protein levels at gastrulation onset determine competitive outcomes. Together, our findings suggest a stage-gated fitness checkpoint as cells prepare for gastrulation. This may be connected with transcriptional programmes that prime the stress response machinery and depend on Brachyury and/or Eomesodermin.

The gastruloid system allows tight control over the starting cell population and revealed that seeding as few as two p53KO cells per mosaic gastruloid (1.3%) is sufficient to measurably impair growth of neighbouring WT cells (Fig. 1f,g), and that 25 p53KO cells robustly outcompete 150 WT cells without exception. These observations demonstrate an extensive effect size of cell competition in our 3D gastruloid system and validate it as a highly suitable model to study this phenomenon. We show that the winner cell status of p53KO supercompetitor cells is not simply a result of their apoptosis resistance (Fig. 3t,u), and that loser cells respond to fitter cells by stabilizing p53 protein (Fig. 3g–k), which results in their apoptosis[6,19] (Fig. 3e,f). This positions p53 as a fitness-defining factor upstream, and simultaneously a response factor downstream of cell competition.

The interactions between WT and p53KO supercompetitor cells go beyond a one-sided growth impairment. While compensatory growth in response to apoptosis had been suggested in some settings[7,43] but argued against in others[20], mosaic gastruloids do reveal compensatory growth of p53KO cells that depends on neighbour cell apoptosis (Extended Data Figs. 2f and 6e). This suggests two-way communication between competing cells, although phagocytic uptake of apoptotic debris and nutrient recycling may also play a role. In mosaics, p53KO cells exert a very mild cell fate bias towards paraxial mesoderm, intermediate mesoderm and caudal epiblast onto neighbouring cells (Fig. 1h,i), but without fully impairing specific lineages. This suggests that, despite the large effect size of loser cell elimination, cell competition is largely phenotypically silent.

It has been proposed that relative pluripotency levels determine winner or loser status during cell competition[14,22], with Myc expression levels as the functional link[14]. We confirm that Myc expression is progressively downregulated during differentiation, and that p53KO cells maintain prolonged Myc expression. However, by creating heterochronic mosaic gastruloids, which contain different genotypes and developmental stages, we find that relative shifts in pluripotency are neither necessary nor sufficient to convey competitive advantages, an experiment uniquely possible due to the modular nature of gastruloids.

We tested the influence of morphogen signalling on cell competition and observe that Wnt and BMP signalling agonism reduces cell competition, while Nodal and ERK modulation does not influence this process. This draws intriguing parallels to recent work linking posteriorizing signals Wnt and BMP to lower and anteriorizing signals to higher amounts of chromosome missegregation and replication stress in pluripotent stem cells[37]. This raises the possibility that these pathways may modulate DNA replication stress and cell competition in a coordinated manner, focusing quality control towards more error-prone cell populations. The correlation between cell competition and anteriorizing/posteriorizing signals also raises the question whether cell competition displays local differences along the anteroposterior axis in the epiblast.

Transcriptomic analysis in mosaic gastruloids draws a timeline in which cells start expressing an apoptotic stress response machinery during primed pluripotency. This response programme is disrupted by the deletion of Brachyury and Eomesodermin, two transcription factors that govern the transition from primed pluripotency to gastrulation[41,42]. Together, this may describe a mechanism that temporally limits cell competition to pre- or early-gastrulation stages, when lost cells can be replaced without compromising specific cell lineages.

In summary, gastruloids provide a robust 3D platform to study cell competition, overcoming limitations of 2D models while enabling precise spatiotemporal analysis and perturbation. Leveraging their

modularity, we reveal the temporal and mechanistic regulation of cell competition and create a foundation for future investigation into embryonic quality control.

## Online content

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

## Methods

### Ethics

All experiments were performed in vitro using established mES cell lines. No live animals or human participants were used. All work was conducted in accordance with the institutional regulations of the Universitat Pompeu Fabra.

### Cell culture

All cells of this study are derived from E14Tg2A mES cells and were cultured at 37 °C in 5% CO$_2$ in medium containing serum and leukemia inhibitory factor (ES/LIF medium) on gelatin-coated (Sigma-Aldrich, G1890) dishes unless otherwise stated. ES/LIF medium was composed of Glasgow Minimum Essential Medium (Thermo Fisher, 11710-035) supplemented with 15% foetal bovine serum (Thermo Fisher, 10270106), 1× non-essential amino acids (thermo Fisher, 11140-035), 1 mM sodium pyruvate (Thermo Fisher, 11360-039), 2 mM L-glutamine (Thermo Fisher, 25030-024), 0.1 mM 2-mercaptoethanol (Thermo Fisher, 31350010) and 10 ng ml$^{-1}$ LIF (Qkine, Qk018-0100). ES cells were passaged as single cells by trypsin digestion (Life Technologies, 25300-096) every other day, and the medium was exchanged daily. Cells were routinely tested negative for mycoplasma infections. All experiments were performed between passage 12 and 30.

### Generation of WT–mCherry, WT–emiRFP and p53KO–emiRFP clones

All stable fluorescent cell lines were generated from E14Tg2A mESCs at passage 12 using the piggyBac system[44]. Cells were cotransfected with pBase and piggyBac donor constructs using Lipofectamine 3000 (Thermo Fisher, L3000001), and single-cell clones were isolated by FACS into 96-well plates. WT–mCherry cells express EF1α–H2B–mCherry and WT–emiRFP cells express CAG–H2B–emiRFP670[45]. For *Trp53* knockout, WT–emiRFP cells were cotransfected with FUCas9Cherry (Addgene, #70182) and a single guide RNA (5′-GGCAACTATGGCTTCCACCT-3′) as previously described[46]. p53 loss of function was selected by 3-day treatment with 10 µM Nutlin-3a (Quimigen, HY-10029), followed by single-clone derivation, genomic sequencing and western blot verification. The chosen p53KO clone (A12-8) carries a homozygous +1 frameshift in *Trp53* exon 4. Normal 40 XY karyotypes of parental E14Tg2A cells, WT–mCherry, WT–emiRFP and p53KO–emiRFP clones were validated by GTG banding in 20 metaphases each. Alkaline phosphatase assays were performed according to the manufacturer's protocols (Sigma-Aldrich, 86R-1KT).

### Generation of other overexpression and knockout clonal cell lines

For inducible *Bcl2* overexpression, cells were cotransfected with pBase, pPB-CAG-rtTA3G-IRES2-NeoR and pPB-TRE3G-Bcl2-IRES-TagBFP (PuroR replaced with TagBFP, original construct gifted by Dr Tristan Rodriguez). TagBFP expression was induced with 1 µg ml$^{-1}$ doxycycline, and single clones were FACS-isolated and chosen for high inducible expression without basal leakiness.

*Fas*-- and *FasL*-knockout clones were generated by CRISPR–Cas9-mediated dual-guide excision using 2X_pX458_pSpCas9(BB)-2A-GFP (Addgene, #172221). For *Fas* knockout in WT–mCherry cells, the guides 5′-CAGTTAAGAGTTCATACTCA-3′ and 5′-GGCATGGTTGACAGCAAAAT-3′ excised the majority of exon 2. For *FasL* knockout in p53KO–emiRFP cells, the guides 5′-CTGTCTACCCAGAAGATCTG-3′ and 5′-TCAGCTCTTCCACCTGCAGA-3′ excised parts of exon 1 including its 3′ exon junction. For *Fadd* knockout in WT–mCherry cells, the guide 5′-CAGCACCAGGAATGGGTCCA-3′ targeting exon 1 was used. The creation of frameshifts and premature stop codons was verified by sequencing for all clones.

Brachyury-knockout (BraKO), Eomes-knockout (EomesKO) and Brachyury/Eomes-dKO cells are based on the same E14Tg2A background and were characterized previously[41,42].

Membrane-labelled WT-GPI-RFP cells used for YAP analysis have been described previously[33].

### Generation of OsTIR-based p53-degron system

The auxin-inducible OsTIR1(F74G) degron system[47,48] degrades mAID2-tagged proteins upon addition of 5-Ph-IAA, a derivative of auxin. A piggyBac plasmid expressing CAG-OsTIR1(F74G)-t2A-H2B-emiRFP670 was synthesized by VectorBuilder. After cells were cotransfected with pBase and this plasmid, a single clone was FACS-isolated ('OsTIR parent'). A mAID2-tag was inserted at the endogenous *Trp53* locus immediately upstream of the stop codon, by Cas9-mediated cut and homology-directed repair using the gRNA 5′-GTGATGGGGACGGGATGCAG-3′ and a synthesized donor plasmid containing mAID2 and homology arms. Correct clones were enriched by cotreatment with 30 µM Nutlin-3a (Quimigen, HY-10029) and 100 nM 5-Ph-IAA (Sigma Merck, SML3574) and confirmed by sequencing. Unimpaired p53 function was revalidated by Nutlin-3a-induced death in absence (but not presence) of 5-Ph-IAA. In gastruloids, 5-Ph-IAA was administered at 100 nM if not otherwise stated, while even 1,000 nM had effects on growth or cell competition in non-degron clones. For washout/pulse experiments, four sequential 150-µl medium exchanges reduced 5-Ph-IAA to ~0.2 nM, which does not cause detectable p53 degradation.

### SDS–PAGE and western blot

Whole-cell lysates were generated and analysed by western blot as previously described[49] under omission of the n-dodecyl β-D-maltoside detergent. In short, cells were lysed in RIPA buffer (Thermo Fisher, 89900) supplemented with complete protease inhibitors (Roche, 11836170001). Chromatin was sheared by sonication, lysates were precleared by centrifugation at 15,000*g* for 10 min at 4 °C, and protein content was determined using a Protein Assay Kit (Bio-Rad, 500-0006). Samples were boiled and reduced in Laemmli buffer (Bio-Rad, 1610747) supplemented with β-mercaptoethanol, and 5 µg protein per sample was separated by SDS–PAGE. After wet transfer, nitrocellulose membranes were analysed by Ponceau staining and subsequently blocked with 5% milk powder in PBS-T (phosphate-buffered saline (PBS) containing 0.1% Tween-20). Primary antibodies against p53 (1:1,000; Leica, NCL-L-p53-CM5p) and β-actin (1:2,000; Santa Cruz, sc-47778) were incubated overnight at 4 °C. HRP-conjugated secondary anti-rabbit (Santa Cruz, sc2357) and anti-mouse (Santa Cruz, sc-516102) antibodies were incubated for 1 h at room temperature (RT), and bands were visualized using the ECL Prime chemiluminescence system and imaged on a ChemiDoc MP (Bio-Rad). p53 and β-actin staining was sequentially conducted on the same membranes.

### Transwell assays

Transwell assays were performed in 24-well plates with WT–emiRFP or p53KO–emiRFP cells in the bottom well and WT–mCherry cells seeded into polyethylene terephthalate transwell inserts with high-density 0.4-µm perforations (Corning, 353495). Wells and inserts were coated for at least 2 h at 37 °C with 0.1% gelatin (Sigma-Aldrich, G1890) for naive conditions, or 2 µg cm$^{-2}$ fibronectin (Bio-Techne, 1918-FN-02M) for primed conditions. After rinsing once with Dulbecco's PBS (DPBS) −/−, 10,000 cells were seeded at the bottom well and 1,600 cells into the insert, to maintain equivalent seeding densities. Cells were cultured in ES/LIF (naive condition) or N2B27 (Takara, XA0530) + 12 ng ml$^{-1}$ bFGF2 (Preprotech, 450-33) + 25 ng ml$^{-1}$ Activin A (QKine, Qk005-100ug) (primed condition). Medium was exchanged daily, and WT–mCherry cells from inserts were dissociated and counted at 48 h, 96 h and 144 h after seeding using a Countess 3 (Thermo Fisher Scientific).

### Mosaic gastruloid formation

Gastruloids were generated as previously described[25], unless otherwise mentioned. In brief, ES cells grown in ES/LIF medium (described above)

were seeded at 45,000 cells per 6-well plate 3 days before gastruloid formation, and medium was exchanged daily. On the day of gastruloid seeding, cells were washed with DPBS without $Ca^{2+}$ and $Mg^{2+}$ ($PBS^{-/-}$) and enzymatically dissociated using prewarmed trypsin for 1.5 min. Cells were gently pipetted up and down to form a single-cell suspension, and trypsin was inactivated by adding an excess of ES medium, followed by pelleting the cells at $200g$ for 2 min. Cells were washed twice with $PBS^{-/-}$ and resuspended in N2B27 (NDiff227-custom, Takara, XA0530). Cells were counted and resuspended in a master mix to yield the desired number of cells per gastruloid in 40 µl per U-bottom 96-well plate (ultralow-attachment plates Greiner, 650970). For single-culture gastruloids, 300 cells were seeded per well, except for p53-KO cells, for which seeding 50 cells produced gastruloids of comparable size at 120 h after aggregation. For mosaic gastruloids, 50% of these numbers were used, resulting in 150 + 150 cells of WT + WT cell mosaics, and 150 + 25 cells of WT + KO mosaic gastruloids. Gastruloids were not perturbed for the first 48 h after aggregation. At 48 h, 150 µl of fresh N2B27 containing 3 µM of ChIR (Chi99021, Sigma, SML1046-5MG) was added. At 72 h, 96 h and 120 h each, 150 µl of medium was removed and exchanged with fresh 150 µl N2B27.

### Microwell gastruloid cocultures

For coculture of gastruloids in a shared medium without direct cell–cell contacts, we used commercially available 'Gri3D' hydrogel microcavity plates with 7 microwells per 96-well plate (InSphero, Gri3D-96IBI-S-96-1600). Gastruloids from individual clones or mosaic mixes were aggregated in ultralow-attachment 96-well plates as described above, followed by transfer of individual gastruloids into microwell cavities of the Gri3D plates under a dissection microscope at 48 h. After 120 h of total growth time, central WT–mCherry gastruloids were collected using a p1000 pipette and dissociated for image analysis-based counting of cells as described below.

### Morphological feature analysis

The size and aspect ratio of gastruloids at 120 h were quantified from phase-contrast and widefield fluorescence images using Fiji (ImageJ) in batch mode. For each image, the intensities of the mCherry and emiRFP fluorescence channels were summed, and a Gaussian blur with sigma 0.6 was applied, followed by intensity thresholding using the maximum entropy method and conversion to binary masks. The Analyze Particles function was used with a size threshold of 5000-infinity pixels to exclude artefacts while selecting gastruloids as the region of interest. After using the Clear Outside, and Fill Holes functions, the size (area in 2D) and aspect ratio were measured. Gross morphology of gastruloids—single-axis, multi-axis or non-elongated—was categorized by visual inspection.

### Dissociation and image-analysis based quantification of cell numbers per gastruloid

This method was adapted following ref. [32]. Individual gastruloids were transferred to Eppendorf tubes, washed in $PBS^{-/-}$ and incubated in 50 µl Accutase (Capricorn Scientific, ACC-1B; prewarmed for 1 min) without agitation for 15 min at RT. The enzymatically loosened but not disintegrated gastruloids were transferred with the Accutase to a 96-well flat-bottom PhenoPlate (PerkinElmer, 6055302) and pipetted up and down 20 times. After inspection, additional pipetting was performed for 96–144-h gastruloids as needed. Pipette tips were prewetted in Accutase before handling gastruloids or cells. Cells were fixed in the same well by adding 100 µl of 1% paraformaldehyde (PFA) in $PBS^{-/-}$ without aspiration. PhenoPlates were tile-imaged using an Opera Phenix (Revvity/PerkinElmer) to detect the H2B–mCherry or H2B–emiRFP670 nuclei. Unlabelled clones were detected by adding Hoechst dye before imaging. Tile images were stitched, and nuclei were automatically segmented and counted using an ImageJ macro. For each condition or timepoint, a minimum of four gastruloids were used as technical repeats, and all conditions were repeated in multiple independent experiments.

### Signalling modulator treatments

Gastruloids were either treated from 48 to 72 h after aggregation with a pulse of 3 µM ChIR (Chi99021, Sigma, SML1046-5MG), or received alternative treatments in the absence of ChIR. Alternative treatment regimens were conducted as follows: 48–72 h Activin A (100 ng ml$^{-1}$, QKINE, Qk005); 48–120 h IWP-2 (1 µM, Sigma, I0536); 48–120 h XAV-939 (1 µM, Tocris, 3748); 48–120 h SB431542 (10 µM, Tocris, 1614); 48–120 h BMP4 (1 ng ml$^{-1}$, R&D, 5020-BP); 48–120 h LDN193189 (100 nM, Tocris, 6053); 48–120 h Fgf2 (12.5 ng ml$^{-1}$, Peprotech, 450-33); 48–120 h PD0325901 (1 µM, Sigma, Pz0162).

### Glucose limitation and metabolic perturbation

Glucose limitation experiments were conducted as described above, but in a custom glucose-free and pyruvate-poor N2B27 medium. Five-hundred millilitres of this N2B27 medium were made by mixing 242.5 ml Neurobasal-A Medium, no D-glucose, no sodium pyruvate (Fisher Scientific, A2477501), 245 ml SILAC Advanced DMEM/F-12 Flex Media, no glucose, no phenol red (Fisher Scientific, A2494301), 5 ml B27 supplement with vitamin A (Gibco, 17504-044), 2.5 ml N2 supplement (Gibco, 17502-048), 5 mL L-glutamine (Gibco, 25030-024), and 0.5 ml β-mercaptoethanol (50 mM, Life Technologies, 31350010). Varying glucose levels were established by adding glucose (Life Technologies, A2494001) to this base medium as annotated.

Inhibition of glycolysis was performed by treating gastruloids in commercial N2B27 medium with 6 mM of 2-deoxyglucose (Sigma Merck, D8375) from 24 to 48 h of gastruloid formation. Inhibition of oxidative phosphorylation was performed by treating gastruloids with 1 mM of sodium azide ($NaN_3$; Sigma Merck, S8032) from 24 to 48 h of gastruloid formation.

### Heterochronic gastruloids

Pure single-culture gastruloids were generated from 300 WT–emiRFP cells, 300 WT–mCherry cells or 75 p53KO–emiRFP cells per well as described above, on different days, and left to grow until indicated timepoints. For example, to create 24 h + 48 h heterochronic mixes, gastruloids of two different clones were aggregated 24 h and 48 h before heterochronic mixing respectively, with the 48-h population always being the constant reference. On the day of heterochronic assembly, gastruloids were collected and pooled, rinsed once in warm $PBS^{-/-}$ and incubated for 5 min in Accutase cell detachment solution (LabClinics, ACC-1B) at 37 °C under agitation. For each condition, a single-cell suspension was obtained by pipetting up and down ~15 times. Single cells were pelleted by 3-min centrifugation at $200g$, resuspended in 100–250 µl N2B27 and counted. Different heterochronic conditions were generated by mixing cells originating from gastruloids of varying timepoints, centred around a population of interest at 48 h. The individual seeding numbers for each clone during heterochronic reaggregation followed previously determined counts of each clone at 48 h with 10% excess to account for loss during reaggregation. Cell mixes were seeded into 96-well ultralow-attachment plates in 150 µl volume, which were centrifuged for 1 min at $150g$ to assure rapid reaggregation and placed back in the incubator. Fresh N2B27 was added to the gastruloids every 24 h until the 48-h reference population had reached 120 h.

### EpiGastruloids

The protocol for generating EpiGastruloids was first established in the doctoral thesis of Shlomit Edri[50]. mES cells in ES/LIF medium were seeded on fibronectin (2 µg cm$^{-2}$) (Bio-Techne, 1918-FN-02M) or vitronectin (0.5 µg cm$^{-2}$) (Fisher Scientific, A14700) coated wells. The following day, medium was exchanged for N2B27 (Takara, XA0530) supplemented with 12 ng ml$^{-1}$ bFGF2 (Peprotech, 450-33) and 25 ng ml$^{-1}$ Activin A (QKine, Qk005-100ug). Twenty-four hours later, colonies

were lifted as clumps using 0.5 mM EDTA in PBS$^{-/-}$ and passaged onto a freshly coated well in the same medium. After three passages, the cells reached a state of Epi-like cells (EpiLC), a transition state towards Epi-stem cells (EpiSCs, ~E7.5 (ref. 51)). In this stage, cells were transcriptionally comparable to 48–72-h gastruloids (high *Fgf5*, *Otx2*, *Oct6*, *Sox3* and *Dnmt3b*). These cells were dissociated using Accutase and seeded as $5 \times 10^4$ cells per 6-well plate in N2B27 + AA + FGF supplemented with 5 μM ROCK inhibitors (Y-27632, Selleckchem, S1049-10MG). On day 2, the medium was exchanged to N2B27 + 20 ng ml$^{-1}$ bFGF2. On day 3, the medium was changed to N2B27 + 20 ng ml$^{-1}$ bFGF2 + 3 μM ChIR (Chi99021, Sigma, SML1046-5MG), comparable to ChIR pretreatment of human gastruloids[52]. After 24 h, these caudal epiblast-like cells (EpiCE)[53] were dissociated using Accutase, washed with PBS$^{-/-}$, resuspended in EpiGastruloid medium (N2B27 + 20 ng bFGF +3 μM ChIR) and counted. EpiGastruloids were aggregated as 600 cells in 40 μl per ultralow-attachment U-bottom 96-well plate (Greiner, 650970). WT + WT and WT + KO mosaic EpiGastruloids were seeded as 300 + 300 and 300 + 50 cells, respectively. Twenty-four hours after aggregation, 150 μl fresh EpiGastruloid medium was added, followed by daily exchange of 150 μl medium. The 72-h EpiGastruloids are temporal equivalents to 120-h classical gastruloids, but with enrichment in *Neurog2*, *Sox1*, *Sox2*, *Foxg1* and *En1* expression, indicating spinal cord and brain trajectories, and a shift towards posterior late hox genes (less *Hoxa1*, *Hoxb1* and *Hoxd1*, and more *Hoxa10*, *Hoxc10* and *Hoxa11*).

## Microscopy

Epifluorescence widefield microscopy was conducted on live cells at 37 °C and 5% CO$_2$ using a Zeiss AxioObserver. For immunofluorescence staining and confocal microscopy, gastruloids were collected at indicated timepoints and washed once in PBS +Ca$^{2+}$/+Mg$^{2+}$ (PBS$^{+/+}$) followed by fixation in 4% PFA (Aname EMS, 15710) overnight at 4 °C. Gastruloids were washed three times in PBS$^{-/-}$ with 0.2% Triton X-100 (Sigma, T8787) (PBS-T) the following day and blocked in PBS-T with 10% bovine serum albumin (BSA, Merck Life Science, 3117332001) for 1.5 h at RT. Primary antibodies were diluted in PBS-T + 2% BSA and incubated with gastruloids at 4 °C overnight under agitation. Gastruloids were washed 6 × 10 min in PBS-T the following day. Fluorophore-conjugated secondary antibodies were diluted 1:500 in PBS-T + 2% BSA together with 1 μg ml$^{-1}$ 4′,6-diamidino-2-phenylindole (DAPI; Thermo Fisher, D1306) and incubated with gastruloids at 4 °C overnight under agitation. Gastruloids were washed 6 × 10 min in PBS-T the following day and mounted in Vectashield antifade mounting medium (Palex Vector, H-1000-10). Microscopy was conducted using a Zeiss LSM980 confocal laser scanning (Airyscan) microscope.

Alternatively, for high-throughput imaging, immunostained gastruloids were transferred in PBS$^{-/-}$ into 96-well PhenoPlates (Revvity/PerkinElmer, 6055302) with up to 12 gastruloids per well, followed by automated *Z*-stack tile imaging on an Opera Phenix (Revvity/PerkinElmer) in spinning disk configuration.

The following antibodies were used for indirect immunofluorescence: mouse anti-p53 (1:4,000, clone 1C12; Cell Signaling, 2524S), rabbit anti-phospho-histone 3 (1:1,600, clone D2C8, Cell Signaling, 3377), rabbit anti-cleaved caspase 3 (1:5,000, clone 5A1E, Cell Signaling, 9664), rabbit anti-YAP (1:200, clone D8H1X, Cell Signaling, 14074), goat anti-E-Cadherin (1:1,000, R&D, AF648), rabbit anti-N-Cadherin (1:200, Abcam, ab18203), rabbit anti-FoxA2 (1:400, clone D56D6, Cell Signaling, 8186), goat anti-Tbx6 (1:200, R&D, AF4744), goat anti-Brachyury (1:100, R&D, AF2085), rabbit anti-Sox2 (1:200, Abcam, ab92494), goat anti-Otx2 (1:200, R&D, AF1979), rabbit anti-Sox3 (1:300, Invitrogen, PA5-35983), rat anti-Nanog (1:250, Invitrogen, 14-5761-80), goat anti-Pecam1 Alexa Fluor 488-conj. (1:400, R&D, FAB3628G-025), mouse anti-SSEA-1 Alexa Fluor 405-conj. (1:400, R&D, FAB2155V-100UG), donkey anti-Goat Alexa Fluor 488-conj. (1:500, Invitrogen, A-11055), goat anti-rabbit Alexa Fluor 488-conj. (1:500, Invitrogen, CA-11034),

goat anti-rat Alexa Fluor 488-conj. (1:500, Invitrogen, A-11006) and donkey anti-mouse Alexa Fluor 488-conj. (1:500, Invitrogen, A-32766).

## Flow cytometry from gastruloids

Gastruloids were collected, washed in DPBS$^{-/-}$ and enzymatically dissociated using Accutase (Lab Clinics Capricorn Scientific, ACC-1B) at 37 °C in a water bath under agitation. After 5 min, flow buffer (PBS$^{-/-}$, 2% BSA, 2 mM EDTA) was added, followed by gentle mechanical dissociation using a p1000 pipette. Cells were washed once in flow buffer, and single-gastruloid flow cytometry was conducted using live cells at this step. For antibody-stained flow cytometry, multiple gastruloids were pooled, dissociated and stained together. For this, single cells were fixed 15 min in 4% PFA, washed three times in flow buffer and blocked in 10% BSA in PBS$^{-/-}$ containing 0.1% Triton X-100 for 30 min at room temperature. After setting aside 10% of the sample as negative and secondary antibody controls, cells were incubated with primary antibodies diluted in 2% BSA in PBS$^{-/-}$ with 0.1% Triton X-100 (staining buffer) for 1 h shaking at RT, followed by three washes in staining buffer. Fluorophore-conjugated secondary antibodies were diluted 1:500 in staining buffer and incubated with cells for 30 min at RT shaking. After three more washes with staining buffer, cells were analysed using a BD Bioscience LSRFortessa system. Antibodies used for flow cytometry: mouse anti-p53 (1:2,000, clone 1C12, Cell Signaling, 2524S), rabbit anti-phospho-histone 3 (1:1,600, clone D2C8, Cell Signaling, 3377) and rabbit anti-cleaved caspase 3 (1:5,000, clone 5A1E, Cell Signaling, 9664).

Apoptosis and necrosis staining was performed using a commercial Annexin V kit (Invitrogen, A13201) in combination with DAPI staining.

Cell cycle analysis was performed using a Click-iT Plus EdU Flow Cytometry Kit (Thermo Fischer Scientific, C10632) according to the manufacturer's protocol. In short, gastruloids were incubated with 10 μM of EdU diluted in N2B27 at 37 °C for 1 h. Full penetrance of EdU to the centre of gastruloids was validated by microscopy. After EdU incorporation, gastruloids were dissociated as described above, fixed, stained according to the manufacturer's protocol and incubated with 1 μg ml$^{-1}$ Hoechst33342 for 30 min at RT. Samples were analysed using a BD Bioscience LSRFortessa system, measuring DNA content on a linear scale.

## Transcriptomic data preprocessing and analysis

The quality of the raw sequencing files was assessed using FastQC (v.0.11.9) (Andrews, 2010; https://qubeshub.org/resources/fastqc). Adapter trimming and low-quality filtering were conducted with Trim-Galore! (v.0.6.7) (Felix Krueger, https://github.com/FelixKrueger/TrimGalore), removing reads with a Phred score below 20. After trimming, the cleaned reads were reevaluated with FastQC.

Processed sequencing data were aligned to the mouse genome (GRCm38.p6) using Gencode annotation (v.25) and the STAR aligner (v.2.7.11b)[54]. Alignments were performed with default settings, and uniquely mapped read counts were generated using the --quantMode option in STAR. A total of 3,695,039,036 uniquely mapped reads were obtained, which served as input for downstream analyses. The final count matrix was created by concatenating the assembled reads in a Python 3.12.8 environment. Raw counts were normalized using the PyDESeq2 module (v.0.4.0), applying median ratio normalization followed by a log1p transformation for downstream analyses.

For dimensionality reduction, highly variable genes were selected and subjected to principal component analysis using the scikit-learn library (v.1.2.2).

To temporally align gastruloid samples with in vivo data, a pseudobulk expression matrix was generated from the Pijuan-Sala dataset (E6.5–E8.5)[55]. Euclidean distances between gastruloid and embryonic samples were computed and visualized as a heatmap.

For spatial comparisons, data from a published embryo atlas[56] containing dissected anterior and posterior portions of E8.5 embryos were used as anterior and posterior references. These anterior and posterior

transcriptomes were mixed at different proportions to generate a pseudobulked anterior-to-posterior signature gradient. Transcriptomes of 120-h gastruloid samples were then embedded into this gradient via principal component analysis to assess closest spatial correspondence.

Visualizations were created using seaborn (v.0.13.2) and matplotlib (v.3.9.2).

The complete code is available via GitHub at https://github.com/stembryo-lab/cell_competition_gastruloids.

GSEA was conducted using the GSEA_4.3.3 desktop application. After loading normalized reads of all samples, GSEA was performed using the mh.all.v2025.1.Mm mouse gene set collection. Analyses used 1,000 phenotype permutations with weighted enrichment statistics, excluded gene sets with fewer than 15 genes, ranked gene sets by Signal2Noise and used a false discovery rate $q$ value <0.1 as the cut-off.

## 3D image segmentation for image analysis

The 3D cell segmentation was performed using the Python package qlivecell (https://github.com/dsb-lab/qlivecell). We used the 2d\_versatile\_fluo pretrained StarDist 2D model[57] and the Cellpose cyto3 model[58,59], and 3D segmentation masks were generated by concatenating the results of 2D segmentations across $z$-planes using qlivecell.

Segmented objects were classified as cells on the basis of two criteria: (1) presence in at least two consecutive $z$-planes and (2) an area exceeding a minimum size threshold in the centre plane.

The centre plane of each object was defined as the $z$-plane with the highest summed fluorescence intensity within the mask. The 2D centroid of the mask in this plane was computed and used, together with the corresponding $z$-coordinate, as the cell centre for all subsequent analyses. Fluorescence quantifications were performed in the centre plane of each cell to minimize crosstalk (Supplementary Fig. 3a–c). The minimum size threshold was determined from the distribution of mask areas measured in the centre plane across all segmented objects. To this end, a kernel density estimate was fitted to the area histogram using the smallest kernel bandwidth that yielded a single local minimum (Supplementary Fig. 3d). This minimum ($31.75 \pm 2.8 \ \mu m^2$) was used to discriminate segmented cells from debris. An upper size threshold of $200 \ \mu m^2$ was applied.

## Classification and quantification of apoptotic events for image analysis

Apoptotic events were quantified on the basis of the segmentation of cleaved caspase-3 (Casp3) immunostaining, followed by manually curation using the qlivecell software to ensure accurate identification. Apoptotic cells were classified into three stages based on a combination of nuclear morphology and Casp3 staining patterns: (1) early apoptosis (intact nucleus surrounded by cytoplasmic Casp3 staining); (2) mid-apoptosis (with nuclear membrane disintegration, Casp3 signal overlapping the nucleus, formation of chromatin granules, and a more intense, fragmented nuclear signal); and (3) late apoptosis (reduced nuclear size and fully compacted chromatin). Examples in Extended Data Fig. 7b. This classification is based on ref. 60, where early apoptosis corresponds to stages 1 and 2, mid-apoptosis to stages 3 and 4, and late apoptosis to stage 5. Early and mid-apoptotic events were segmented manually, whereas late apoptotic events were segmented using StarDist, while applying intensity- and size-based thresholds to distinguish apoptotic debris, following an approach similar to the above-described kernel density estimation approach. The intensity threshold was defined as above the 90th percentile of the mean Casp3 signal measured in early apoptotic events. Apoptotic cells were subsequently assigned to mCherry or emiRFP populations based on their nuclear fluorescence in these channels, after normalization of the mean fluorescent intensities in both channels

## Radial distribution of cells for image analysis

Because gastruloids exhibit irregular 3D shapes, we defined a relative radial position metric ($P$) for each cell. For a cell $i$, we computed the distance to the 3D centroid of the gastruloid, $D_c^i$, and to the closest gastruloid edge, $D_e^i$. The relative position was defined as

$$P^i = \frac{D_c^i}{D_c^i + D_e^i}.$$

Cells located at the centroid have $P = 0$, whereas cells located at the gastruloid boundary have $P = 1$ (Supplementary Fig. 3e). Distances were computed in three dimensions using the cell centre coordinates. The gastruloid edge was identified independently for each $z$-plane using the scikit-image implementation of morphological active contours without edges (Chan–Vese model)[61].

## Local cell density and neighbourhood composition for image analysis

Local cell density ($\rho_{local}$) was estimated using a nearest-neighbours approach. For each cell $i$, we identified its $n$ closest neighbours in three dimensions using the $k$-nearest neighbours algorithm[62]. Let $d_{ij}$ denote the Euclidean distance between cell $i$ and its $j$th neighbour. The local density was defined as the inverse of the cubic mean distance to the neighbours:

$$\rho_{local}^i = \left( \frac{1}{n} \sum_{j=1}^{n} d_{ij}^3 \right)^{-1}.$$

To avoid artefacts, neighbouring cells located closer than 0.25 cell diameters were excluded from the neighbourhood before quantification.

Neighbourhood composition was defined as the fraction of neighbouring cells belonging to a given population within the local neighbourhood of each cell. Neighbourhoods were determined using the $k$-nearest neighbours algorithm[62] in three dimensions.

## YAP quantification for image analysis

Plasma membranes and nuclei of each cell were segmented separately, using GPI–RFP and DAPI signals respectively (Extended Data Fig. 7j). The cytoplasmic mask was defined as the membrane mask excluding the nuclear region. Before YAP quantification, background subtraction was performed independently for the nuclear and cytoplasmic compartments using YAP signal intensity from secondary-antibody-only samples. YAP analysis resulted in the ratio of nuclear to cytoplasmic fluorescence intensity. Cells and their intensities were classified into two populations, based on the H2B–emiRFP channel, which exhibited a clear bimodal distribution.

## p53 quantification for image analysis

Nuclear p53 levels were quantified from the mean nuclear fluorescence intensity in the p53 immunostaining channel.

Nuclei were segmented using the H2B–mCherry and H2B–emiRFP channels. The resulting nuclear masks were used to extract per-pixel intensities from all channels.

We corrected for spectral spillover by using a linear compensation model calibrated on p53 secondary-antibody-only controls, where true p53 signal is absent. Let $C_{obs}(z)$ denote the observed p53 intensity and $B(z)$ the H2B–mCherry intensity at a pixel in plane $z$; on in-cell pixels of the control we fitted

$$C_{sec}(z) = b_0(z) + sB(z)$$

and took the slope $s$ (spillover coefficient) as global for the session. Because the p53 baseline varies with depth, for each image stack we estimated a per-plane offset $b_0(z)$ by computing a robust baseline (median) of the residuals $C_{obs}(z) - sB(z)$ over in-cell pixels at each $z$. All experimental images were then corrected per pixel as

$$C_{corr}(z) = C_{obs}(z) - b_0(z) - sB(z),$$

and per-cell nuclear p53 was defined as the mean of $C_{corr}$ over the pixels inside each nuclear mask.

We verified that the dependence of p53 on H2B–mCherry intensity was removed by the correction (that is, slope of $C_{corr}$ versus $B$ ~0 and distribution of $C_{corr}$ in secondary-only images ~0 across $z$; Supplementary Fig. 3f). Cells with abnormally high p53 were classified using a robust, per-plane threshold based on Tukey fences: for each $z$, letting $\mathscr{M}_z = (m_{i,z})$ be the distribution of p53 values on all cells (mean $C_{corr}$), we calculated $Q1_z$, $Q3_z$, and $IQR_z = Q3_z - Q1_z$, and set

$$\tau_z = Q3_z + 4.5 IQR_z.$$

Cells were labelled p53-high if their p53 intensity exceeded $\tau_z$. Thresholds were computed independently for each plane. Some cells were segmented in both fluorescence channels, as indicated by detectable mCherry signal in cells segmented in the emiRFP channel and vice versa. To correct for this, cells were clustered using $k$-means clustering ($k = 2$) based on their mCherry and emiRFP fluorescence intensities. Cells whose segmentation channel did not match their cluster assignment were excluded from further analysis (Supplementary Fig. 3g).

### Statistics and reproducibility
Unless otherwise stated, all statistical comparisons of exactly two samples were conducted by two-sided $t$-test, and all comparisons containing more than two samples were conducted by one-way analysis of variance (ANOVA) with Tukey's post-hoc multiple comparisons test, or two-way ANOVA with Tukey's or Šidák's post-hoc multiple comparisons test, depending on the dimensionality and nature of the dataset. Data throughout this Article are depicted as mean ± standard error of the mean (s.e.m.) unless otherwise stated. Data distribution was assumed to be normal, but this was not formally tested. All experiments have been conducted in at least three independent repeats, except for one timepoint in Fig. 8i, with only two repeats due to our laboratory closure. Individual gastruloids were excluded from the analysis only in the case of technical failure during processing, such as loss of cells during dissociations, and never for other reasons. No statistical method was used to predetermine sample size. The experiments were not randomized. Data collection and analysis were not performed blind to the conditions of the experiments. All microscopy micrographs in this Article are representatives of at least three independent repeats. Western blot membranes in Figs. 1a and 8a are representative of two independent repeats, and Fig. 8e depicts single experiments. No statistical analyses were derived from western blot data.

### Reporting summary
Further information on research design is available in the Nature Portfolio Reporting Summary linked to this article.

### Data availability
RNA sequencing data that support the findings of this study have been deposited in the Gene Expression Omnibus (GEO) under accession code GSE294530. All other data supporting the findings of this study are available from the corresponding author on reasonable request. Source data are provided with this paper.

### Code availability
All custom code is available via GitHub at https://github.com/stembryo-lab/cell_competition_gastruloids. Information about the qlivecell package used to handle image segmentation and quantifications is available via GitHub at https://github.com/dsb-lab/qlivecell. All code used for the image analysis is available via GitHub at https://github.com/stembryo-lab/cell_competition_gastruloids/tree/main/image-analysis. Executable examples for the image analysis are available via GitHub at https://github.com/stembryo-lab/cell_competition_gastruloids/tree/main/image-analysis/examples, together with an example dataset that can be used to test the analysis.

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

## Acknowledgements
We thank T. Rodriguez for sharing a *Bcl2* overexpression plasmid, V. Trivedi and the EMBL-Barcelona for access to imaging equipment,

A. Janic for sharing the *Trp53* gRNA, the UPF/CRG Flow Cytometry, Advanced Light Microscopy and Genomics core facilities, and, above all, T. Balayo for universal support. J.F. was supported by an EMBO postdoctoral fellowship (ALTF 605-2022) and a 'la Caixa' Foundation (ID 100010434) Junior leader fellowship (LCF/BQ/PI23/11970017). S.B. was supported by an Boehringer Ingelheim Fonds PhD fellowship. A.M.A., J.B., P.P.M. and G.R. are funded by an ERC AdG (MiniEmbryoBlueprint_834580) and A.M.A. also by the 'Maria de Maeztu' Program for Units of Excellence in R&D (grant no. CEX2018-000792-M). P.C.G. and J.G.O. were supported by project PID2021-127311NB-I00 by the Spanish Ministry of Science and Innovation, the Spanish State Research Agency and the European Regional Development Fund (FEDER) and the ICREA Academia programme. Figures 1 and 2 contain illustrations from the public domain NIH Bioart repository (https://bioart.niaid.nih.gov/).

## Author contributions

Conceptualization, J.D.F. and A.M.A.; investigation, J.D.F., S.B., J.B.J.G. and G.R.; formal analysis, J.D.F., P.C.G. and P.P.M.; methodology, J.D.F. and S.E.; project administration, J.D.F. and A.M.A.; software, P.C.G. and P.P.M.; writing – original draft, J.D.F.; writing – review and editing, J.D.F., A.M.A., S.B., J.B.J.G., S.E., P.C.G., P.P.M., G.R. and J.G.O.; resources, J.D.F., A.M.A., P.C.G., G.R., A.E.W. and S.J.A.; visualization, J.D.F.; funding acquisition, J.D.F., J.G.O. and A.M.A.; supervision, J.D.F. and A.M.A.

## Competing interests

The authors declare no competing interests.

## Additional information

**Extended data** is available for this paper at https://doi.org/10.1038/s41556-026-01923-x.

**Correspondence and requests for materials** should be addressed to Joshua D. Frenster or Alfonso Martinez Arias.

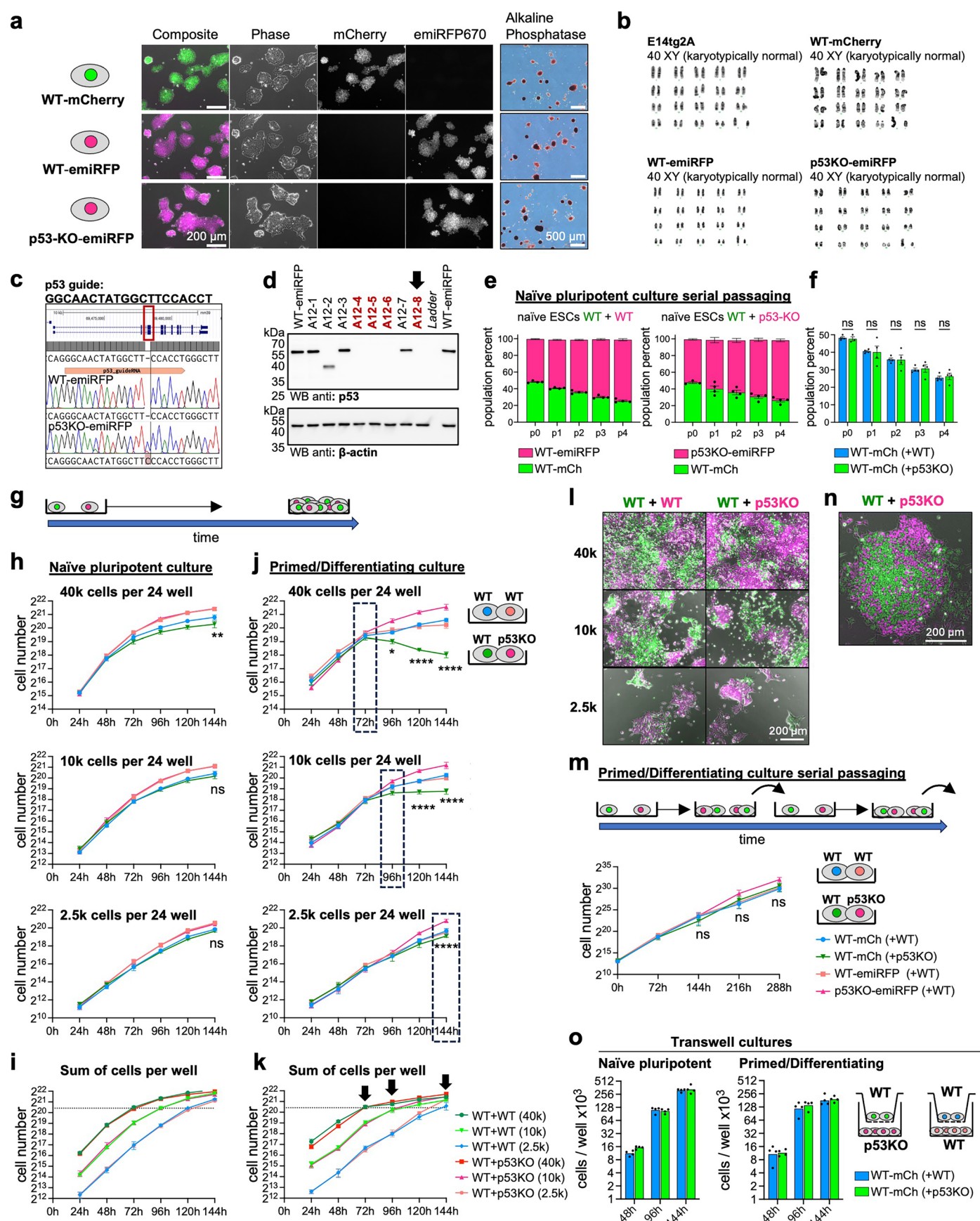

**Extended Data Fig. 1 | See next page for caption.**

**Extended Data Fig. 1 | 2D cell competition occurs only at confluence during differentiation. a**, Overview of H2B-mCherry and H2B-emiRFP670-tagged mESC clones of this study. **b**, Confirmation of normal karyotypes by G-banding. **c**, Sequencing alignment of WT cells and p53KO-emiRFP clone "A12-8" to *TrpS3* reference gene detects homozygous +1 frameshift in exon 4 of *Trp53* gene. **d**, Western blot screen of parental WT-emiRFP and daughter clones stained against p53 and β-actin. Confirms loss of p53 in clone A12-8 (arrow), called "p53KO-emiRFP" in this study. **e**, **f**, Flow cytometric analysis of population composition of mixed naïve pluripotent WT + WT or WT+p53KO co-cultures (**e**), and comparison of WT-mCherry percentage across co-cultures (**f**). **g**, Schematic of co-culture under naïve or primed pluripotency without passaging. **h**–**k**, Growth curves from naïve (**h**, **i**) or primed pluripotent (**j**, **k**) 2D co-cultures without passaging. 1:1 mixes of WT-mCherry (blue) and WT-emiRFP (orange), or WT-mCherry (green) and p53KO-emiRFP (magenta) cells, seeded as 40,000, 10,000, or 2,500 total cells. Sum of WT + WT or WT + KO cells per well from (**h**, **j**) depicted in (**i**, **k**). Dashed boxes (**j**) and black arrows (**k**) indicate onset of competition. Statistical comparison only for WT-mCherry cells from WT + WT vs WT+p53KO. (**h**) (40k_144 h) $p$adj = 9.3×10$^{-3}$. (**j**) (40k_96h) $p$adj = 9.3×10$^{-3}$, (40k_120h/144 h) $p$adj < 1×10$^{-4}$, (10k_120h/144h) $p$adj < 1×10$^{-4}$, (2.5k_144h) $p$adj < 1×10$^{-4}$. **l**, Representative micrographs of 72 h co-cultures quantified in panel (**j**). **m**, Growth curves of co-cultured clones in primed pluripotency, with EpiSc-like clump passaging every 72 h. **n**, Representative micrograph of a mixed colony at second passage of primed pluripotent co-cultures, as in panel (m). **o**, WT-mCherry cell counts from transwell co-cultures. All data presented as mean ± SEM. n = 3 (**h**–**k**, **m**) or n = 4 (**e**, **f**, **o**) indep. experiments. Statistical analysis by two-way ANOVA with Sidak's post hoc test and is depicted as *, $p < 0.05$; **, $p < 0.01$; ***, $p < 0.001$; ****, $p < 0.0001$; ns, not significant. Scale bars denote 200 μm, or 500 μm as annotated.

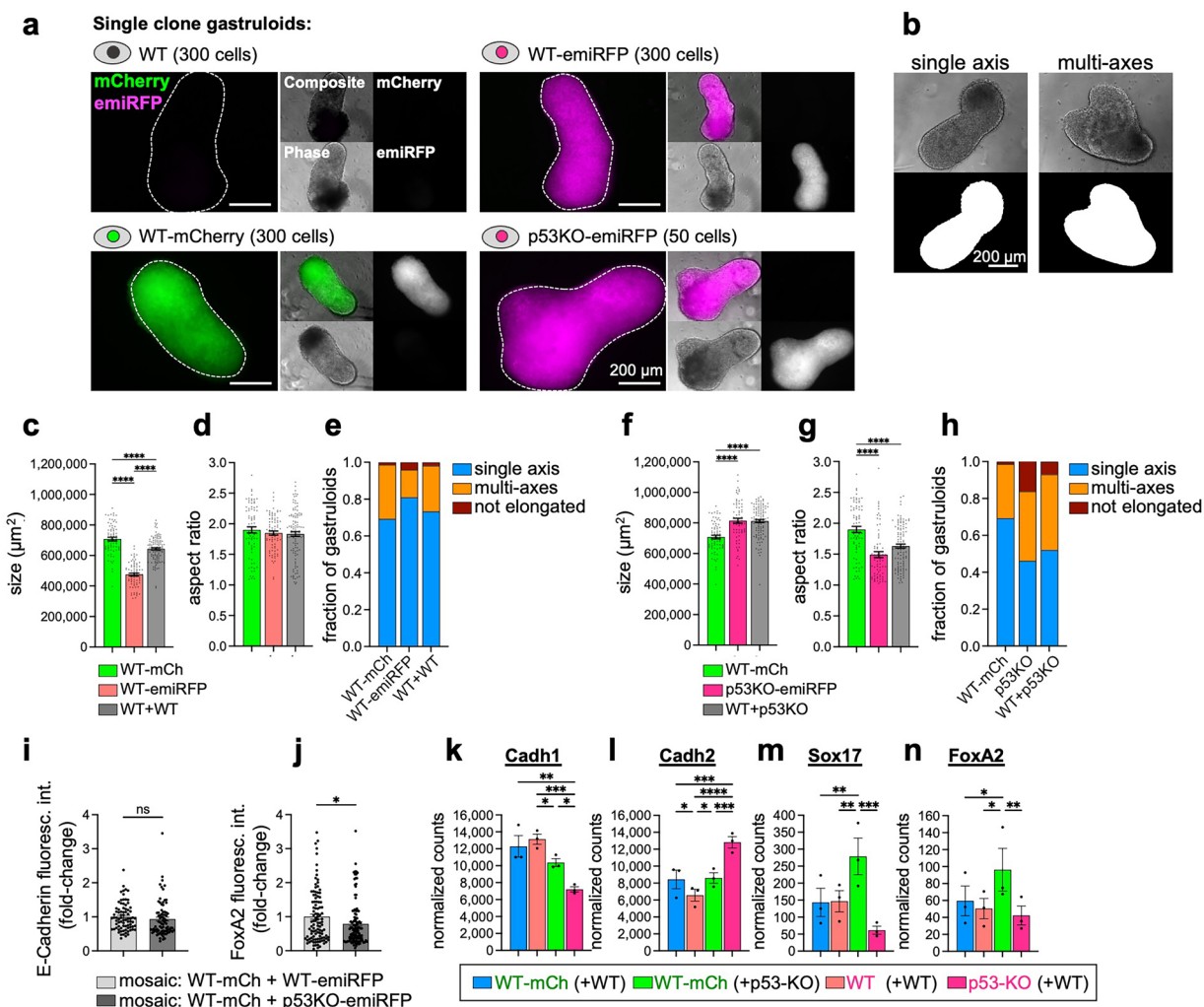

**Extended Data Fig. 2 | Characterization of mosaic gastruloid morphology and cell fates. a**, Representative widefield micrographs of 120 h gastruloids from parental WT, WT-mCherry, WT-emiRFP, or p53KO-emiRFP cells. **b**, Examples of 120 h single- and multi-axes gastruloids and masks used for morphology analysis. **c–h**, Morphological characterization of pure and mosaic gastruloids by size (**c, f**), aspect ratio (**d, g**), and frequency of single- vs multi-axes formation (**e, h**). n = 68(p53KO-emiRFP), n = 72(WT-mCh) n = 78(WT-emiRFP), n = 108 (WT + KO), n = 111(WT + WT) gastruloids per condition. (**c, f, g**) **** signifies *p*adj < 1×10⁻⁴. **i, j**, Quantified intensities of E-cadherin (**i**) and FoxA2 (**j**) immunostaining in mosaic gastruloids, fold change of WT+p53KO gastruloids compared to WT + WT controls. n = 79 (ECad:WT + WT), n = 81 (ECad:WT + KO), n = 105 (FoxA2:WT + WT), n = 122 (FoxA2:WT + KO) gastruloids from 3 indep.

experiments. (**j**) *p*adj = 2.5×10⁻² **k–n**, Normalized transcript levels of Cadh1 (**k**), Cadh2 (**l**), Sox17 (**m**), and FoxA2 (**n**) from FACS-separated populations of 120 h mosaic WT + WT and WT+p53KO gastruloids from n = 3 indep. experiments. *p*-values from top to bottom and left to right: (**k**) *p*adj = 1×10⁻³, *p*adj = 4×10⁻⁴, *p*adj = 2×10⁻², *p*adj = 1.1×10⁻². (**l**) *p*adj = 3×10⁻⁴, *p*adj < 1×10⁻⁴, *p*adj = 2.5×10⁻², *p*adj = 1.7×10⁻², *p*adj = 4×10⁻⁴. (**m**) *p*adj = 8.8×10⁻³, *p*adj = 9.9×10⁻³, *p*adj = 7×10⁻⁴. (**n**) *p*adj = 1.6×10⁻², *p*adj = 4.2×10⁻², *p*adj = 7.4×10⁻³. All data presented as mean ± SEM. Statistical analysis either by unmatched (**c, d, f, g**) or row-matched (**k, l, m, n**) one-way ANOVA with Tukey's post hoc test, or unpaired two-tailed t-test (**i, j**) and depicted as *, *p* < 0.05; **, *p* < 0.01; ***, *p* < 0.001; ****, *p* < 0.0001; ns, not significant. Scale bars denote 200 µm.

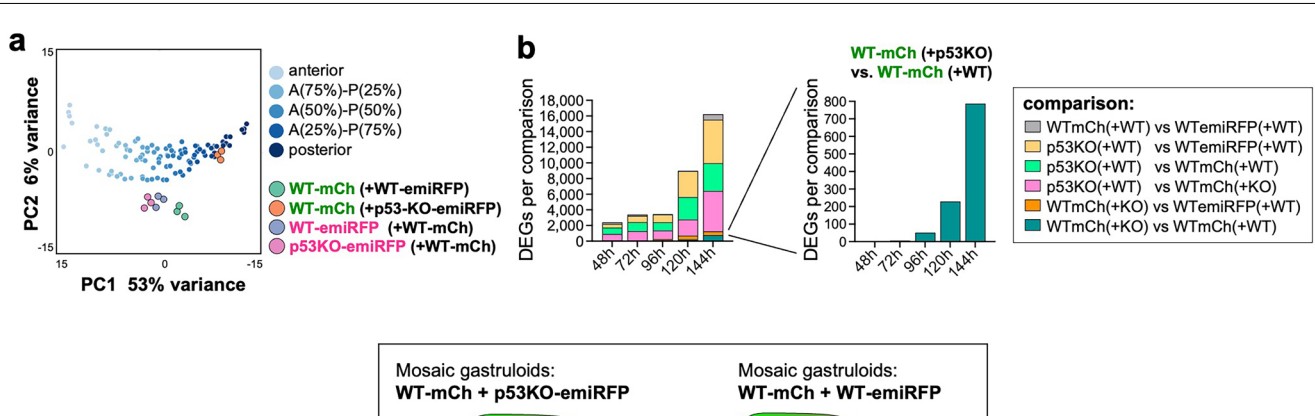

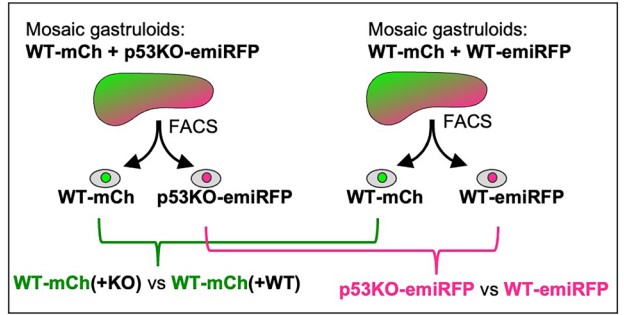

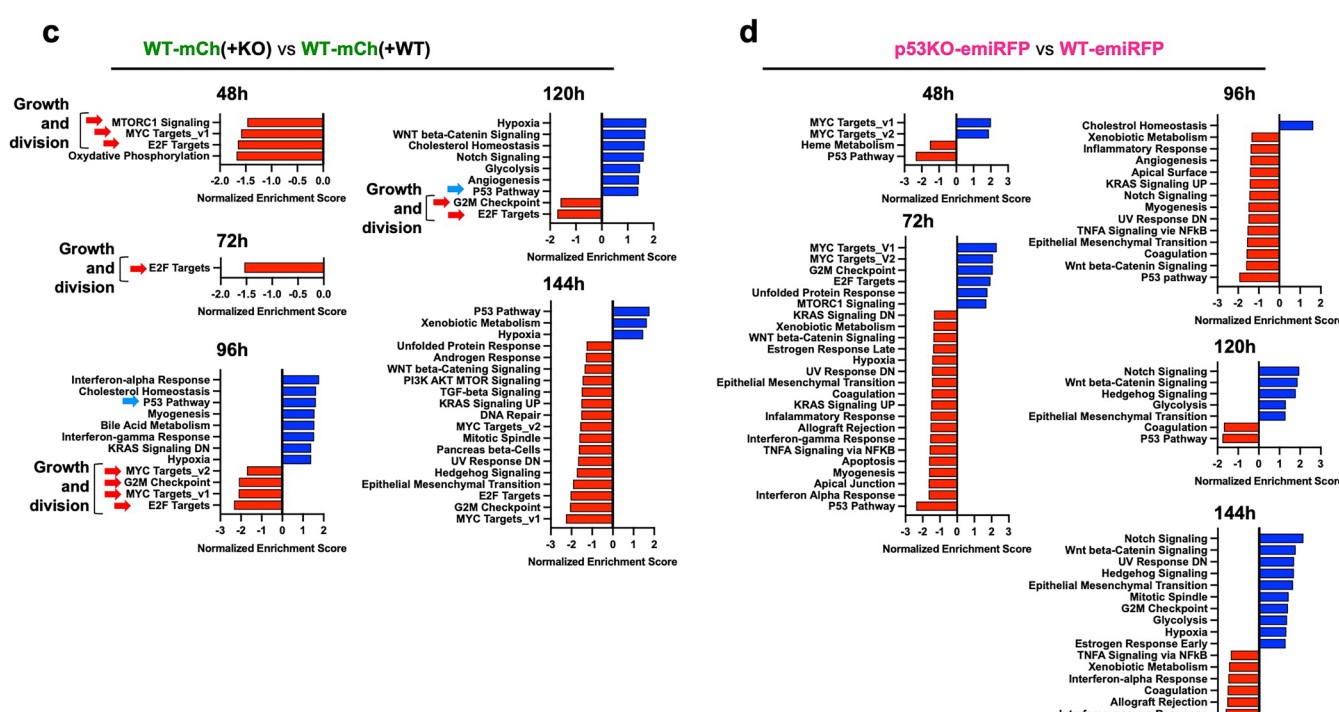

**Extended Data Fig. 3 | Transcriptomic analysis of competing populations.**
**a**, Principal component analysis of FACS-separated populations from 120 h mosaic WT + WT and WT+p53KO gastruloids with anterior-to-posterior pseudobulked subsets of embryonic stage E8.5 transcriptomes (blue shades). **b**, Number of significant differentially expressed genes (DEG) between pairs of FACS-isolated clonal populations from mosaic gastruloids harvested at varying timepoints. **c, d**, Gene Set Enrichment Analysis (GSEA) comparing WT-mCherry (**c**) or WT/p53KO-emiRFP cells (**d**) from mosaic WT + WT vs WT+p53KO gastruloids at varying timepoints. MSigDB Hallmark pathways and significance cutoff FDR q < 0.1 applied. All analysis is based on transcriptomes from 3 indep. experiments.

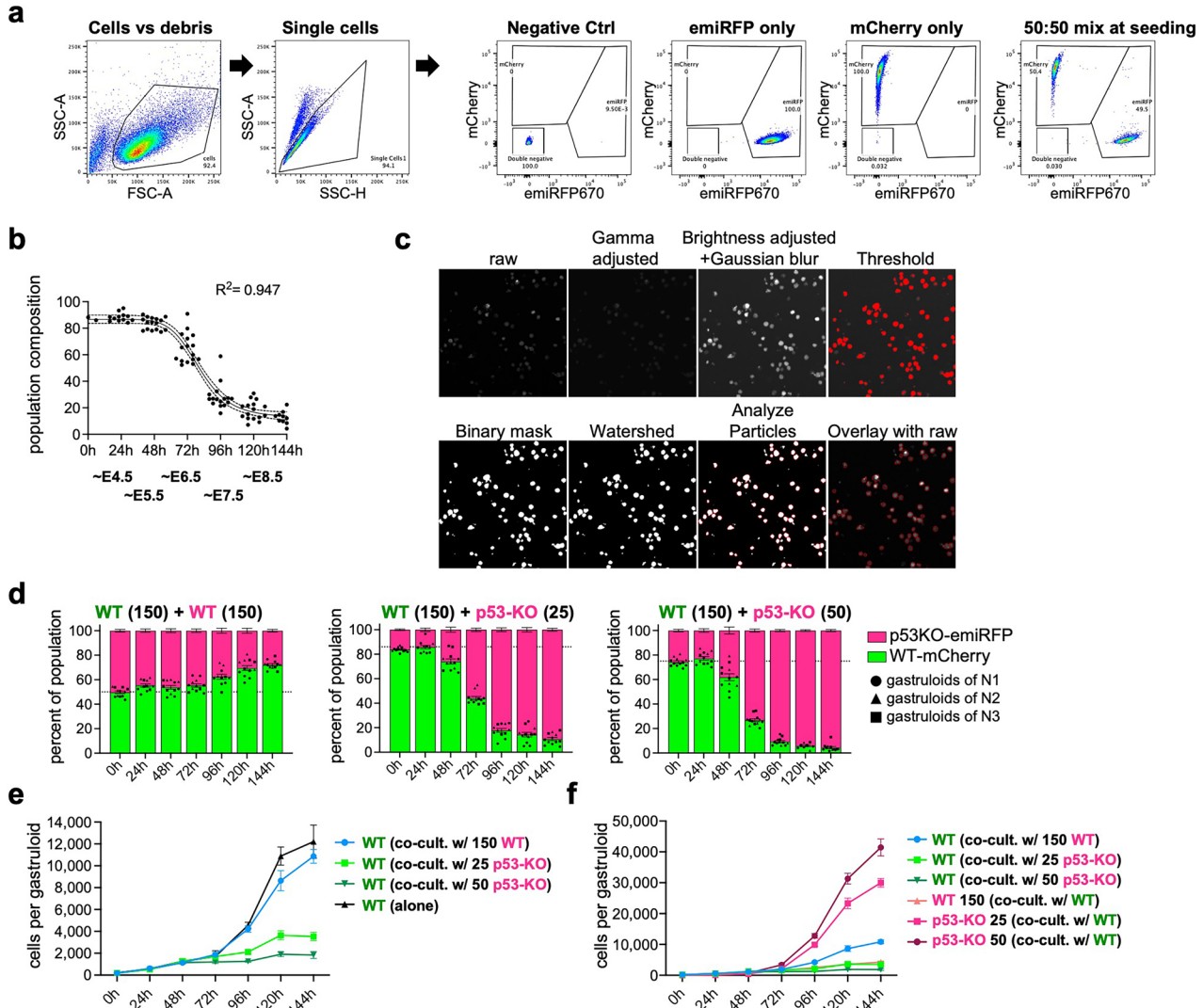

**Extended Data Fig. 4 | Flow cytometry and simple image analysis-based single gastruloid cell population quantification. a**, Flow cytometry gating strategy to distinguish mCherry vs emiRFP cells in mosaic gastruloids. No bleed-through is observed between mCherry and emiRFP channels. **b**, Corresponds to Fig. 2i. (Nonlinear) sigmoidal-4PL (four parameter logistic) curve fit to WT-mCherry population percentages in WT+p53KO mosaic gastruloids. Curve fit with $R^2 = 0.947$. Equation: $Y = 73.98/((80.74/X)^{(-7.265)} + 1) + 12.75$. **c**, Overview of image processing steps during simple nuclear segmentation of 2D tiled images of individually dissociated gastruloids. Corresponds to Fig. 2m. **d**, Population

percentages of mCherry and emiRFP cells in mosaic gastruloids derived from absolute cell counts with methodology outlined in panel Fig. 2m. **e**, **f**, Absolute growth curves of WT-mCherry cells in mosaic gastruloids seeded together with 150 WT-emiRFP (blue), 25 p53KO-emiRFP (light green), 50 p53KO-emiRFP (dark green) cells, or WT-mCherry cells alone (black line) (**e**), and counts of WT-emiRFP and p53KO-emiRFP cells (**f**) from the same mosaic gastruloids (Corresponds to Fig. 2n). All data presented as mean ± SEM. n = 12 (**b**, **d**–**f**) gastruloids from n = 3 (**d**–**f**), or n = 4 (**b**) indep. experiments.

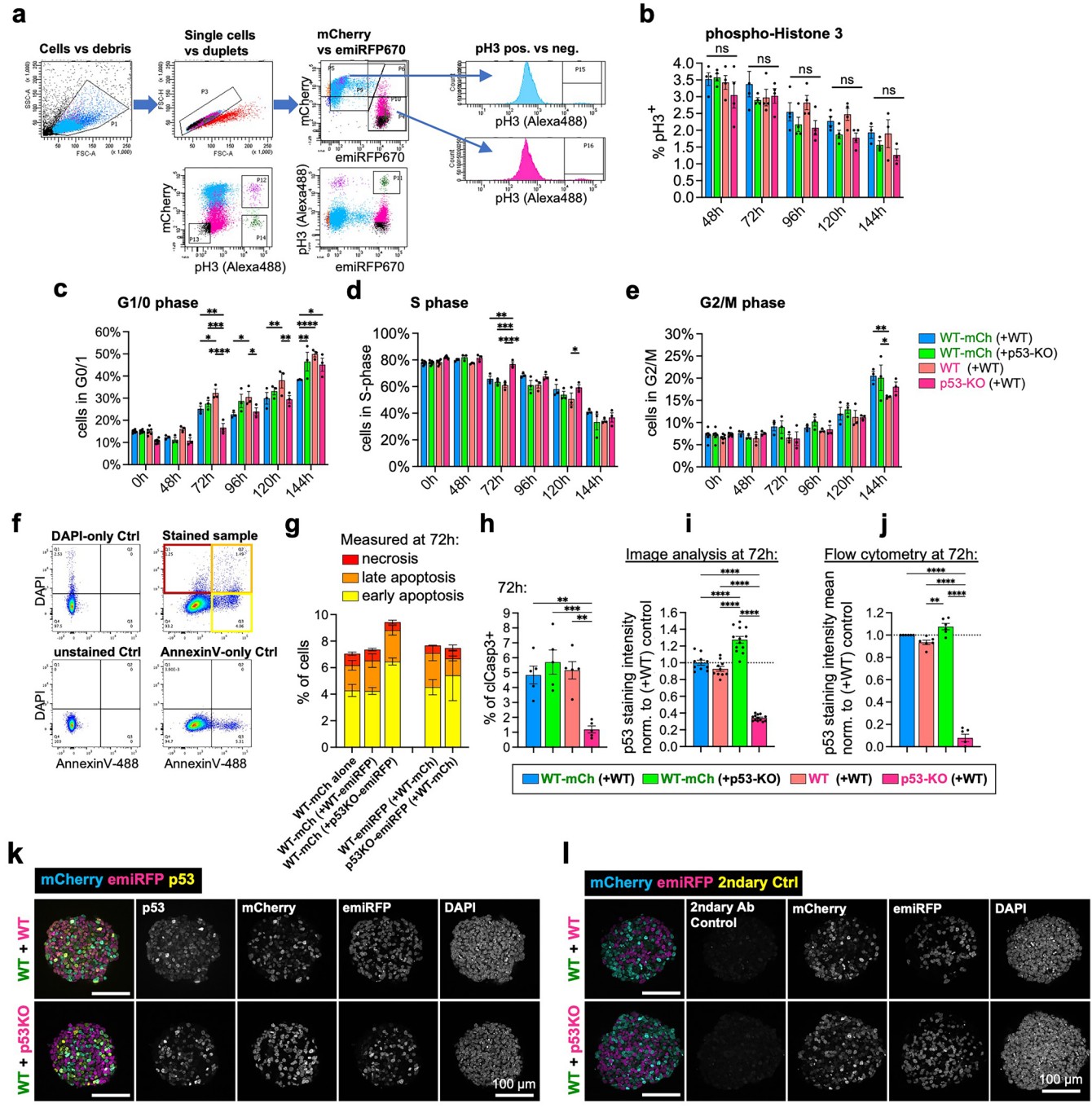

**Extended Data Fig. 5 | Cell competition in gastruloids is proliferation-independent and stress-based. a**, Flow cytometry gating strategy for quantifying phospho-histone3 (pH3⁺)-positive percentage of mCherry or emiRFP cells within mosaic gastruloids. **b–e**, Flow cytometric quantification of pH3⁺ cells (**b**) and cell cycle distributions (**c–e**) in WT + WT or WT+p53KO gastruloids. Two-Way ANOVA with post-hoc Tukey's multiple testing within each timepoint; (**b**) ns, not significant. (**c–e**) $p$-values from top to bottom and left to right: (**c**) $p$adj = 6.1×10⁻³, $p$adj = 3.8×10⁻², $p$adj = 3×10⁻⁴, $p$adj = 8.8×10⁻³, $p$adj < 1×10⁻⁴, $p$adj = 2.5×10⁻², $p$adj = 1.2×10⁻², $p$adj = 4.9×10⁻³, $p$adj = 8.4×10⁻³, $p$adj = 1×10⁻⁴, $p$adj = 4.9×10⁻². (**d**) $p$adj = 3.3×10⁻³, $p$adj = 3×10⁻⁴, $p$adj < 1×10⁻⁴, $p$adj = 3.2×10⁻². (**e**) $p$adj = 5.5×10⁻³, $p$adj = 1.4×10⁻². **f**, Example of apoptosis stage analysis by flow cytometry with single-color and unstained controls. **g**, Percentages of early apoptotic, late apoptotic, and necrotic cells in mosaic 72 h gastruloids. **h**, Percent of cleaved-caspase-3⁺ cells in 72 h mosaic gastruloids measured by

flow cytometry. **i**, Normalized p53 staining intensity determined through image analysis in 72 h mosaic gastruloids, expressed as mean values per gastruloid from n = 10 (WT + WT), n = 13 (WT + KO) gastruloids from 3 independent experiments. **j**, Flow cytometric quantification of p53 levels normalized to WT-mCherry cells in 72 h WT + WT gastruloids. $p$-values from top to bottom and left to right: (**h**) $p$adj = 2.5×10⁻³, $p$adj = 3×10⁻⁴, $p$adj = 1.2×10⁻³. (**i**) **** signifies $p$adj < 1×10⁻⁴. (**j**) **** signifies $p$adj < 1×10⁻⁴, ** $p$adj = 1.6×10⁻³. **k**, Representative confocal micrographs of p53 immunostained 72 h WT + WT and WT+p53KO mosaic gastruloids. **l**, Representative secondary antibody-only control micrograph corresponding to panel (k) and Fig. 3g, h. All data presented as mean ± SEM. n = 3 (**i**), n = 4 (**b–e, g**), n = 5 (**h**), or n = 6 (**j**) independent experiments. Statistical analysis by two-way ANOVA with Sidak's post hoc test (**b–e**), one-way ANOVA with Tukey's post hoc test (**h–j**) and is depicted as *, $p < 0.05$; **, $p < 0.01$; ***, $p < 0.001$; ****, $p < 0.0001$; ns, not significant. Scale bars denote 100 µm.

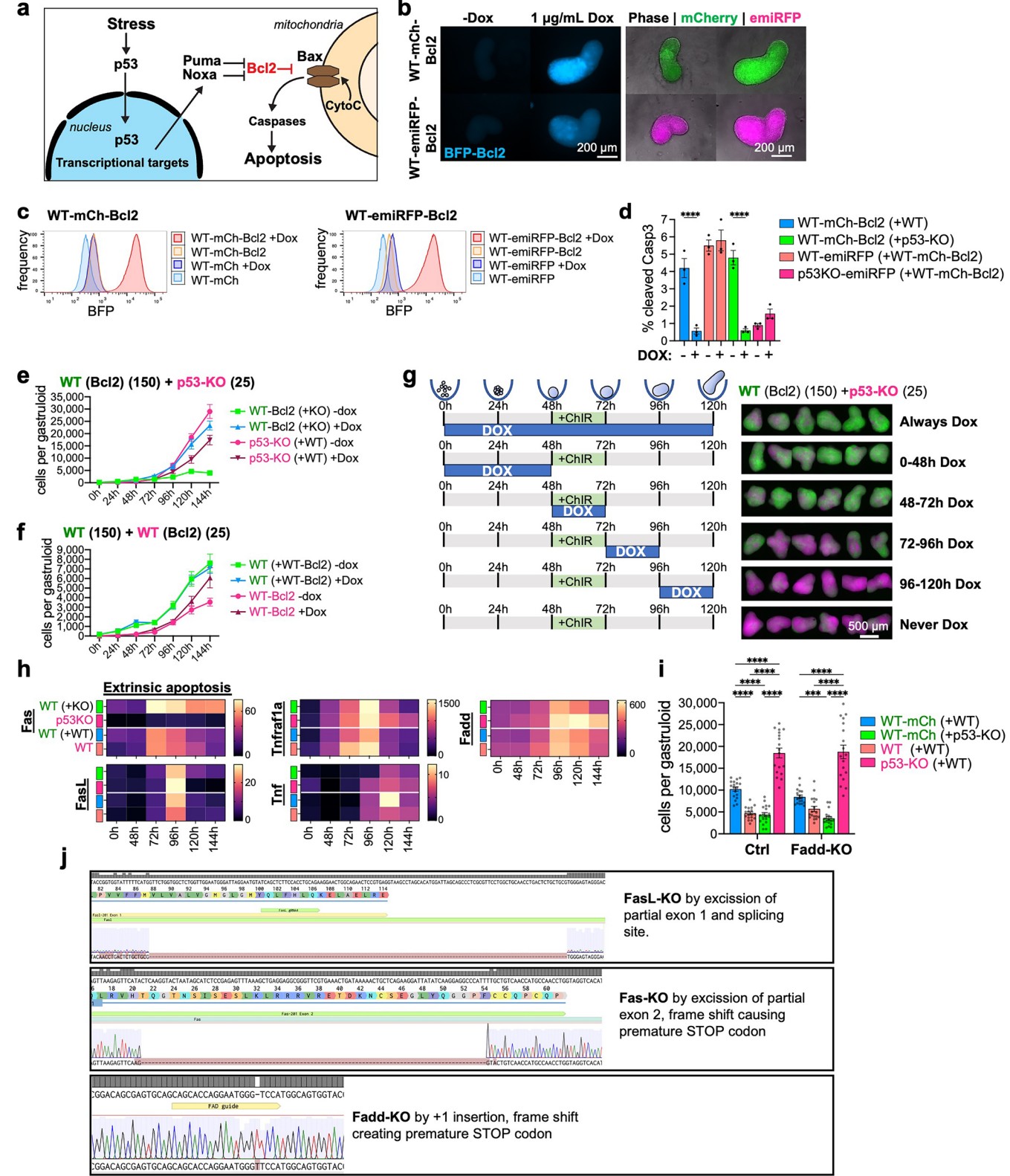

**Extended Data Fig. 6 | See next page for caption.**

**Extended Data Fig. 6 | Cell competition in gastruloids is mitochondrial apoptosis-based. a**, Schematic of stress response/intrinsic apoptosis pathway downstream of p53. **b**, **c**, Fluorescent micrographs (**b**) and flow cytometry (**c**) demonstrating doxycycline-inducible *Bcl2* overexpression clones, as detected by co-expressed BFP. **d**, Percentage of clCasp3-positive cells in 72 h mosaic WT(Bcl2)+WT or WT(Bcl2)+p53KO gastruloids ±doxycycline measured by flow cytometry. **** signifies *p*adj < 1×10⁻⁴. **e**, **f**, Overlay of growth curves of all populations from Fig. 3l–n (**e**) or Fig. 3o–q (**f**). **g**, Schematic of temporal *Bcl2* induction series with representative micrograph montages of resulting 120 h gastruloids. **h**, Heatmap of expression dynamics of extrinsic apoptosis pathway genes in FACS-separated cells from mosaic gastruloids. **i**, Cell counts in 120 h mosaic gastruloids containing WT-mCherry or WT-mCherry-Fadd-knockout

cells. n = 18 gastruloids per condition from 3 indep. experiments. **** signifies *p*adj < 1×10⁻⁴, *** *p*adj = 1×10⁻⁴. **j**, Sequencing confirmation of homozygous gene disruption in *FasL-*, *Fas-*, *Fadd*-knockout clones. *FasL* disruption via partial excision of exon 1 and its splicing site, resulting in nonsense-mediated decay and premature STOP-codon. *Fas* disruption via homozygous partial excision of exon 2 resulting in premature STOP-codon. *Fadd* disruption via homozygous +1 insertion causing frameshift and premature STOP-codon. All data presented as mean ± SEM. n = 3 (**d**, **h**, **i**) indep. experiments. Statistical analysis by two-way ANOVA with Sidak's post hoc test and is depicted as *, *p* < 0.05; **, *p* < 0.01; ***, *p* < 0.001; ****, *p* < 0.0001; ns, not significant. Scale bars denote 200 μm (**b**) or 500 μm (**g**).

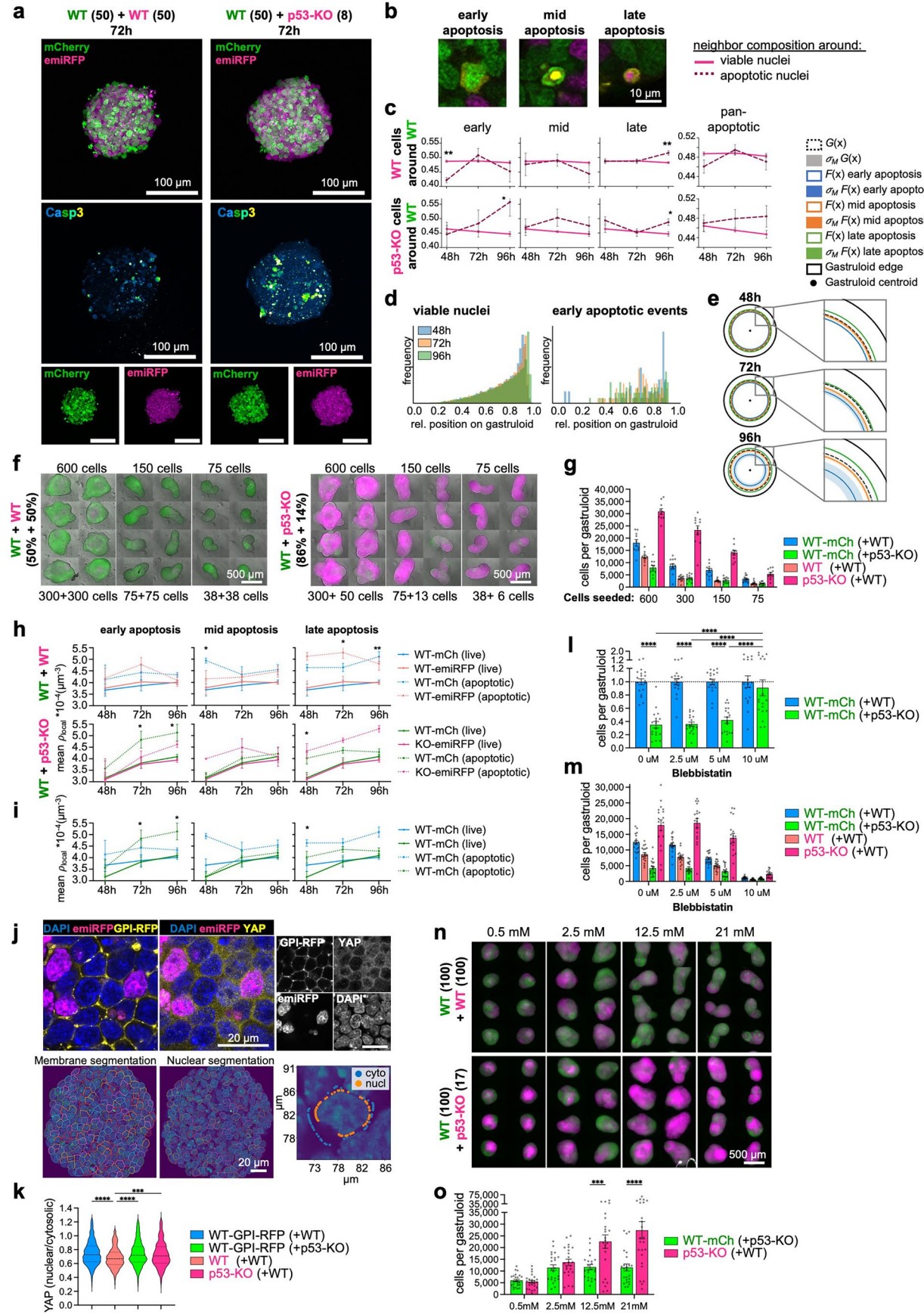

**Extended Data Fig. 7 | See next page for caption.**

**Extended Data Fig. 7 | Apoptotic events are biased toward p53KO neighborhoods and inner regions of gastruloids. a**, Representative projection of clCasp3 immunostained 72 h mosaic WT + WT or WT+p53KO gastruloids. **b**, Examples of apoptotic stages identified by clCasp3 staining (yellow) and nuclear morphology. **c**, 3D neighborhood as normalized percentage of emiRFP nuclei among 20 nearest neighbors centered around viable (solid) and apoptotic (dotted line) mCherry nuclei. n = 4 (WT + WT_48h), n = 6 (WT + KO_48h), n = 8 (WT + WT_72h), n = 7 (others) gastruloids per condition. (early_48h) $p$adj = 5.5×10$^{-4}$, (early_96h) $p$adj = 1.5×10$^{-2}$, (late_96hWT) $p$adj = 1.2×10$^{-3}$, (late_96hKO) $p$adj = 1.1×10$^{-2}$. **d**, Distribution histograms of viable and apoptotic nuclei along center-to-edge axis measured as expanding shell in 3D segmented gastruloids. **e**, Radial distribution of all nuclei (black dotted ring), early (blue), mid (orange) or late (green) apoptotic events within gastruloids. **f**, Representative micrographs of 120 h mosaic WT + WT or WT+p53KO gastruloids seeded from 600, 150, or 75 total cells. **g**, Cells counts in 120 h mosaic gastruloids, as panel **f**. n = 12 gastruloids from 3 indep. experiments. **h**, 3D local densities centered around viable (solid line) or apoptotic (dotted line) nuclei in mosaic WT + WT or WT+p53KO gastruloids. Statistical comparison between viable and apoptotic WT-mCherry cells. n = 4 (WT + WT_48h), n = 6 (WT + KO_48h), n = 8 (WT + WT_72 h), n = 7

(others) gastruloids per condition. (**h, i**) (early_KO_72 h) $p$adj = 1.8×10$^{-2}$, (early_KO_96h) $p$adj = 3.1×10$^{-2}$, (mid_WT_48h) $p$adj = 4.3×10$^{-2}$, (late_KO_48h) $p$adj = 1.5×10$^{-2}$, (late_WT_72h) $p$adj = 3.6×10$^{-2}$, (late_WT_96h) $p$adj = 2.6×10$^{-3}$. **i**, Overlaid WT-mCherry graphs from panel (h). **j**, Representative micrographs of YAP immunostained WT-membrane(GPI)-RFP + p53KO-emiRFP 72h mosaic gastruloids and representative membrane and nuclear segmentation. **k**, quantified nuclear-to-cytosolic YAP ratios from 72h mosaic WT-GPI-RFP + WT-emiRFP and WT-GPI-RFP+p53KO-emiRFP gastruloids. n = 450-1819 nuclei from 3-4 gastruloids per condition. (k) **** $p$adj < 1×10$^{-4}$, *** $p$adj = 1×10$^{-4}$. **l, m**, Normalized (**l**) and absolute (**m**) cell counts from mosaic 120 h gastruloids treated with blebbistatin (48-96 h). n = 18 gastruloids from 3 indep. experiments. (**l**) **** signifies $p$adj < 1×10$^{-4}$. **n**, Representative micrographs of 120 h mosaic gastruloids grown in varying glucose concentrations. **o**, Cell counts from 120 h WT+p53KO mosaic glucose limited gastruloids. n = 24 gastruloids each condition from 4 indep. experiments. **** $p$adj < 1×10$^{-4}$, *** $p$adj = 3×10$^{-4}$. All data presented as mean ± SEM. Statistical analysis by two-way ANOVA with Sidak's post hoc test (**c, g, h, i, l, m, o**) or one-way ANOVA with Tukey's post hoc test (**k**) and is depicted as *, $p$ < 0.05; **, $p$ < 0.01; ***, $p$ < 0.001; ****, $p$ < 0.0001; ns, not significant. Scale bars denote 10 μm (**b**), 20 μm (**j**), 100 μm (**a**) or 500 μm (**f, n**).

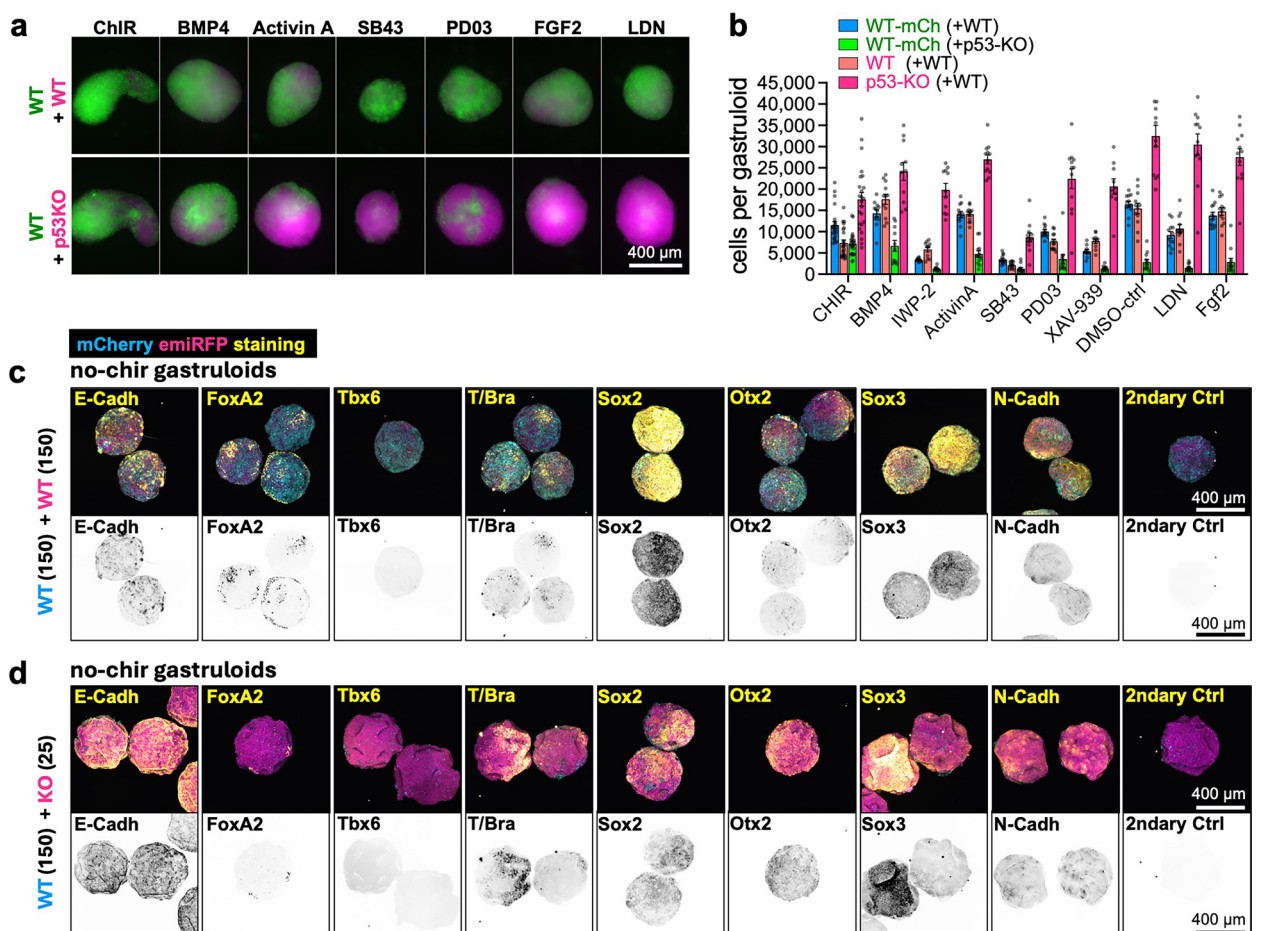

**Extended Data Fig. 8 | Effects of signaling perturbations on cell competition and fate specification. a**, **b**, Representative micrographs (**a**) and cell counts (**b**) of 120 h mosaic WT + WT and WT+p53KO gastruloids treated with developmental signal modulators. Gastruloids were treated with 3 µM ChIR (48-72 h), 1 ng/mL BMP4 (48-120 h), 1 µM IWP-2 (48-120 h), 100 ng/mL Activin A (48-72 h), 10 µM SB431542 (48-120 h), 1 µM PDO325901 (48-120 h), 1 µM XAV-939 (48-120 h), 12.5 ng/mL FGF2 (48-120 h), 100 nM LDN193189 (48-120 h), or 0.1% DMSO carrier control. n = 12 gastruloids each from 3 indep. experiments, except 'ChIR' with

n = 24 gastruloids from 6 indep. experiments. **c**, **d**, Fate marker immunostaining of mosaic WT + WT (**c**) and WT+p53KO (**d**) gastruloids at 120 h grown without ChIR treatment (DMSO control). Maximum intensity projections of spinning disk confocal imaged gastruloids. Micrographs depicted as composite (top panel) of mCherry (blue), emiRFP (magenta), and marker staining (yellow), or inverted grey scale marker staining alone (lower panel). All values depicted as mean ± SEM. All scale bars denote 400 µm.

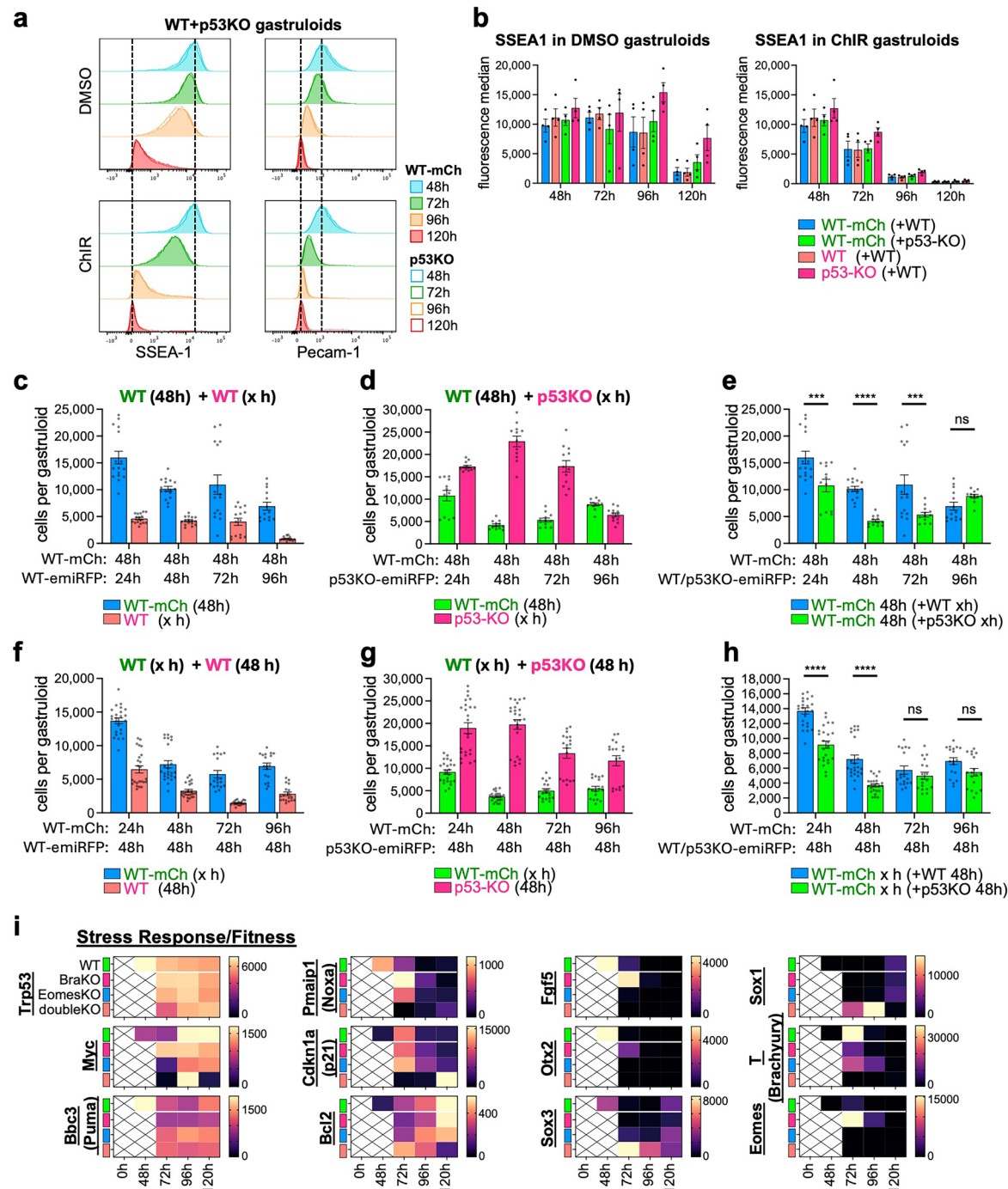

**Extended Data Fig. 9 | Temporal analysis of gene expression dynamics and heterochronic gastruloids. a**, Flow cytometric analysis of SSEA1 and Pecam-1 surface expression in WT-mCherry (filled histograms) and WT-emiRFP (open histograms) cells from mosaic gastruloids treated with ChIR (lower panel) or DMSO control (upper panel) analyzed at 48-120 h. Representative result from 4 indep. experiments. **b**, Median SSEA-1 surface expression in cells from mosaic gastruloids treated with ChIR (48-72 h) or DMSO control. n = 4 indep. experiments. **c–e**, Heterochronic mosaic gastruloids created by the reaggregation of cells from 48 h WT-mCherry gastruloids with cells from WT-emiRFP or p53KO-emiRFP gastruloid of varying timepoints. Counts in mosaic

WT + WT (**c**) or WT+p53KO (**d**) gastruloids at 120 h, 72 h after reaggregation, and comparison of WT-mCherry cell counts from WT + WT vs WT+p53KO gastruloids (**e**). From left to right: $p$adj = $4×10^{-4}$, $p$adj < $1×10^{-4}$, $p$adj = $2×10^{-4}$. **f–h**, Heterochronic experiments as in panels c-e, but with shifted developmental time of WT-mCherry cells. **** $p$adj < $1×10^{-4}$. (**c–h**) n = 12 gastruloids from 3 indep. experiments. Conducted with home-made N2B27. **i**, Gene expression dynamics (compare with Fig. 6a, b, n) in WT, Brachyury-KO, Eomesodermin-KO, or double-KO cells. Reanalyzed after Tosic et al., 2019. All values depicted as mean ± SEM. Statistical analysis by two-way ANOVA with Sidak's post hoc test (**e**, **h**), depicted as *** $p$ < 0.001; **** $p$ < 0.0001; ns, not significant.

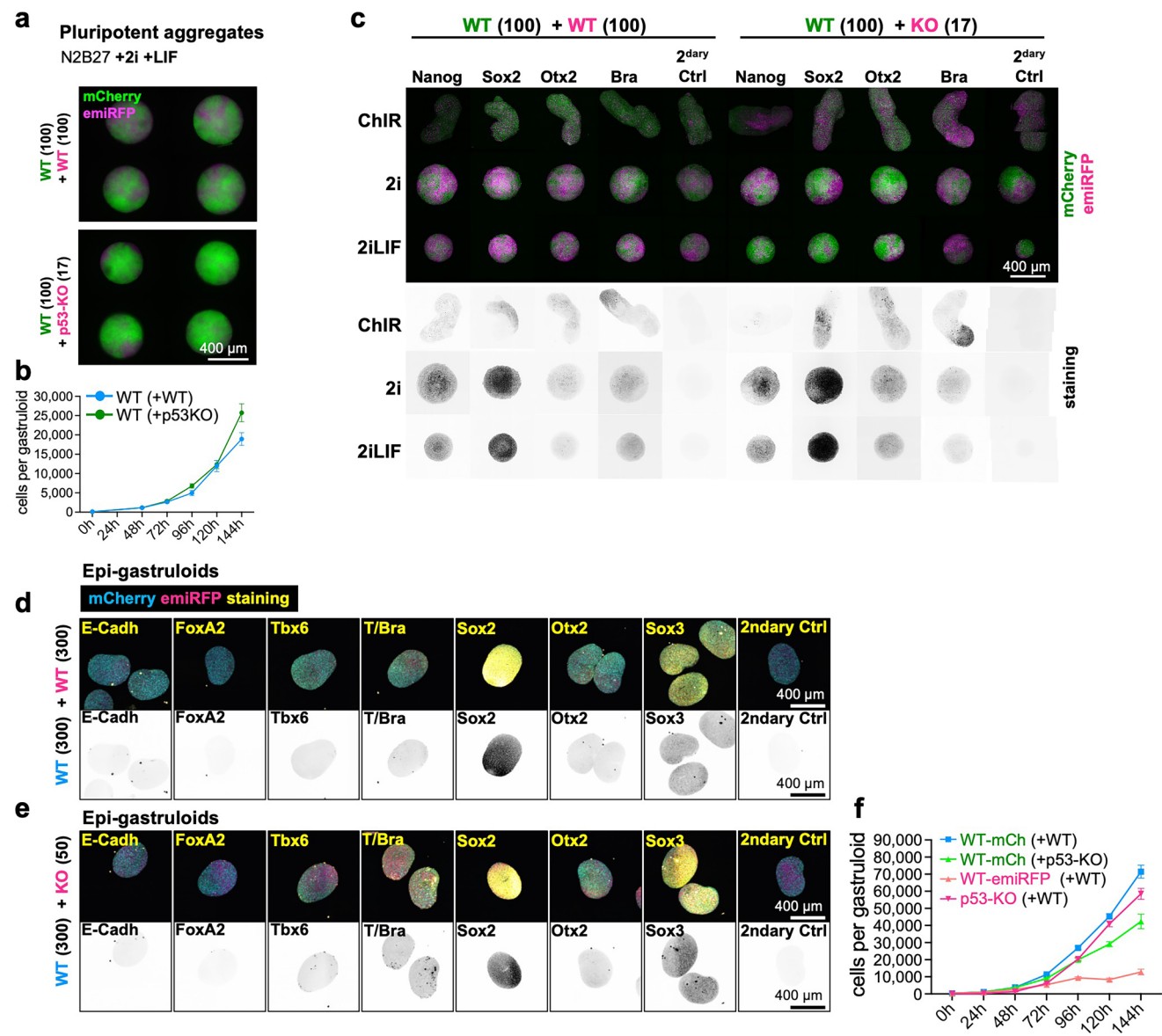

**Extended Data Fig. 10 | Fate and competition analysis of pluripotent aggregates and EpiGastruloids. a**, Representative micrographs of 120 h pluripotent mosaic 3D aggregates grown in N2B27 + 2i+LIF. (corresponds to condition without LIF in Fig. 7a). **b**, WT-mCherry growth curves in mosaic pluripotent aggregates seeded together with WT-emiRFP (blue) or p53KO-emiRFP (dark green) cells. Mean ± SEM of n = 12 gastruloids per datapoint from 3 indep. experiments. **c**, Fate marker immunostaining of 120 h mosaic WT + WT (left) and WT+p53KO (right) aggregates grown either as standard gastruloids (ChIR) or remaining as pluripotent aggregates in 2i or 2i+LIF. Maximum intensity projections of confocal laser scanning Z-stacks. Micrographs depicted as composite of mCherry

(green) and emiRFP (magenta), or inverted immunostaining alone (gray scale). **d**, **e**, Fate marker immunostaining of mosaic WT + WT (**d**) and WT+p53KO (**e**) EpiGastruloids at 72 h. Maximum intensity projections of spinning disk confocal Z-stacks. Micrographs depicted as composite (top panel) of mCherry (blue), emiRFP (magenta), and marker staining (yellow), or inverted grey scale marker staining alone (lower panel). **f**, Growth curves of cell populations in mosaic EpiGastruloids depicted as mean ± SEM. n = 17 gastruloids per datapoint from 3 indep. experiments. Scale bars denote 400 μm.

# Reporting Summary

## Statistics

For all statistical analyses, confirm that the following items are present in the figure legend, table legend, main text, or Methods section.

| n/a | Confirmed | |
|---|---|---|
| ☐ | ☒ | The exact sample size ($n$) for each experimental group/condition, given as a discrete number and unit of measurement |
| ☐ | ☒ | A statement on whether measurements were taken from distinct samples or whether the same sample was measured repeatedly |
| ☐ | ☒ | The statistical test(s) used AND whether they are one- or two-sided<br>*Only common tests should be described solely by name; describe more complex techniques in the Methods section.* |
| ☐ | ☒ | A description of all covariates tested |
| ☐ | ☒ | A description of any assumptions or corrections, such as tests of normality and adjustment for multiple comparisons |
| ☐ | ☒ | A full description of the statistical parameters including central tendency (e.g. means) or other basic estimates (e.g. regression coefficient) AND variation (e.g. standard deviation) or associated estimates of uncertainty (e.g. confidence intervals) |
| ☐ | ☒ | For null hypothesis testing, the test statistic (e.g. $F$, $t$, $r$) with confidence intervals, effect sizes, degrees of freedom and $P$ value noted<br>*Give P values as exact values whenever suitable.* |
| ☒ | ☐ | For Bayesian analysis, information on the choice of priors and Markov chain Monte Carlo settings |
| ☐ | ☒ | For hierarchical and complex designs, identification of the appropriate level for tests and full reporting of outcomes |
| ☒ | ☐ | Estimates of effect sizes (e.g. Cohen's $d$, Pearson's $r$), indicating how they were calculated |

*Our web collection on statistics for biologists contains articles on many of the points above.*

## Software and code

Policy information about availability of computer code

| Data collection | FlowJo (version 10.10.0), BD FACS Diva (version 6.2), Zeiss ZEN (version 3.11), Zeiss ZEN Blue (version 3.2), Harmony PhenoLOGIC (version 5.2), EPview (version v2.9.22), FastQC (version 0.11.9), TrimGalore (version 0.6.7), STAR aligner (version 2.7.11b) |
|---|---|
| Data analysis | ImageJ2 (version 2.14.0/1.54f), GraphPad Prism (version 10.4.1(532)); RNAseq analysis: Python (version 3.12.8), PyDESeq2 (version 0.4.0), scikit-learn library (version 1.2.2), seaborn (version 0.13.2), matplotlib (version 3.9.2); Image analysis: Python (version 3.9.13), qlivecell (version 0.7 available at https://github.com/dsb-lab/qlivecell), GSEA (version 4.3.3)<br><br>Custom code available under: https://github.com/stembryo-lab/cell_competition_gastruloids |

For manuscripts utilizing custom algorithms or software that are central to the research but not yet described in published literature, software must be made available to editors and reviewers. We strongly encourage code deposition in a community repository (e.g. GitHub). See the Nature Portfolio guidelines for submitting code & software for further information.

## Data

Policy information about availability of data

 All manuscripts must include a data availability statement. This statement should provide the following information, where applicable:
- Accession codes, unique identifiers, or web links for publicly available datasets
- A description of any restrictions on data availability
- For clinical datasets or third party data, please ensure that the statement adheres to our policy

> All RNA sequencing datasets generated in this study have been deposited and are publicly available in the Gene Expression Omnibus under accession no. GSE294530.

## Research involving human participants, their data, or biological material

Policy information about studies with human participants or human data. See also policy information about sex, gender (identity/presentation), and sexual orientation and race, ethnicity and racism.

| | |
|---|---|
| Reporting on sex and gender | N/A |
| Reporting on race, ethnicity, or other socially relevant groupings | N/A |
| Population characteristics | N/A |
| Recruitment | N/A |
| Ethics oversight | N/A |

Note that full information on the approval of the study protocol must also be provided in the manuscript.

# Field-specific reporting

Please select the one below that is the best fit for your research. If you are not sure, read the appropriate sections before making your selection.

☒ Life sciences ☐ Behavioural & social sciences ☐ Ecological, evolutionary & environmental sciences

For a reference copy of the document with all sections, see nature.com/documents/nr-reporting-summary-flat.pdf

# Life sciences study design

All studies must disclose on these points even when the disclosure is negative.

| | |
|---|---|
| Sample size | No statistical method were used to predetermine the samples sizes, but each result was repeated in at least 3 independent experiments, containing several technical repeats per biological repeat. Numbers of independent repeats and number of gastruloids per repeat are annotated in each figure legend. |
| Data exclusions | Individual gastruloids were only excluded from quantitative analysis of cell number counts if their counts were unreliable due to technical problems like loss of cells during the handling procedure of dissociation. No other data was excluded. |
| Replication | All experiments were replicated in at least 3 independent experiments. RNAseq samples were collected from 3 temporally separate independent experiments. Within each independent experiment multiple gastruloids were analyzed. |
| Randomization | Gastruloids were not specifically chosen for any experiment and individual gastruloids for quantification were picked randomly from 96 well plates without prior analysis or preference to specific gastruloids. Sequencing data was analyzed without human intervention in an unbiased fashion. |
| Blinding | Data collection and analysis were not performed blind to the conditions of the experiments. Blinding was not required for this study since it is not an intervention study and does not contain subjective or qualitative assessment. All samples were analyzed using the same uniform parameters in an unbiased manner. |

# Reporting for specific materials, systems and methods

We require information from authors about some types of materials, experimental systems and methods used in many studies. Here, indicate whether each material, system or method listed is relevant to your study. If you are not sure if a list item applies to your research, read the appropriate section before selecting a response.

## Materials & experimental systems

| n/a | Involved in the study |
|-----|----------------------|
| ☐ | ☒ Antibodies |
| ☐ | ☒ Eukaryotic cell lines |
| ☒ | ☐ Palaeontology and archaeology |
| ☒ | ☐ Animals and other organisms |
| ☒ | ☐ Clinical data |
| ☒ | ☐ Dual use research of concern |
| ☒ | ☐ Plants |

## Methods

| n/a | Involved in the study |
|-----|----------------------|
| ☒ | ☐ ChIP-seq |
| ☐ | ☒ Flow cytometry |
| ☒ | ☐ MRI-based neuroimaging |

# Antibodies

| | |
|---|---|
| Antibodies used | Primary antibodies:<br>p53 (WB) (Leica, Cat# NCL-L-p53-CM5p, 1:1000), p53(IF, Flow)(Cell Signaling, Cat# 2524S, clone 1C12, 1:4,000), beta-actin (Santa Cruz, Cat# sc-47778, 1:2000), E-Cadherin (R&D, Cat# AF648, 1:1000), N-Cadherin (Abcam, Cat# ab18203, 1:200), FoxA2 (Cell Signaling, Cat# 8186, 1:400), Tbx6 (R&D, Cat# AF4744, 1:200), T/Brachyury (R&D, Cat# AF2085, 1:100), Sox2 (Abcam, Cat# ab92494, 1:200), Otx2 (R&D, Cat# AF1979, 1:200), Sox3 (ThermoFisher, Cat# PA5-35983, 1:300), Nanog (Invitrogen, Cat# 14-5761-80, 1:250), phospho-Histone3 (Cell Signaling, Cat# 3377, 1:1600), cleaved Caspase3 (Cell Signalling, Cat# 9664, 1:500), YAP (Cell Signaling, Cat# 14074, clone D8H1X, 1:200), CD31/PECAM-1 AlexaFluor 488 conjugated (R&D, Cat# FAB3628G-025, 1:400), SSEA-1 AlexaFluor 405 conjugated (R&D, Cat# FAB2155V-100UG, 1:400).<br><br>Secondary antibodies: HRP-conjugated mouse anti-rabbit (Santa Cruz, Cat# sc2357, 1:2000), HRP-conjugated m-IgGk BP anti-mouse (Santa Cruz, Cat# sc-516102, 1:2000), Alexa Fluor 488 Donkey anti-Goat (Invitrogen, Cat# A-11055, 1:500), Alexa Fluor 488 Goat anti-Rabbit (Invitrogen, Cat# A-11034, 1:500), Alexa Fluor 488 Goat anti-Rat (Invitrogen, Cat# A-11006, 1:500). |
| Validation | Antibodies were validated by their manufacturers:<br>p53 antibody for Western blot: https://shop.leicabiosystems.com/ihc-ish/ihc-primary-antibodies/pid-p53-protein-cm5<br>p53 antibody for Immunofluorescence and Flow cytometry: https://www.cellsignal.com/products/primary-antibodies/p53-1c12-mouse-monoclonal-antibody/2524<br>beta-actin atibody: https://www.scbt.com/p/beta-actin-antibody-c4<br>E-Cadherin antibody: https://www.rndsystems.com/products/human-mouse-e-cadherin-antibody_af648<br>N-Cadherin antibody: https://www.abcam.com/en-us/products/primary-antibodies/n-cadherin-antibody-intercellular-junction-marker-ab18203<br>FoxA2 antibody: https://www.cellsignal.com/products/primary-antibodies/foxa2-hnf3b-d56d6-xp-rabbit-mab/8186<br>Tbx6 antibody: https://www.rndsystems.com/products/human-tbx6-antibody_af4744<br>Brachyury abntibody: https://www.rndsystems.com/products/human-mouse-brachyury-antibody_af2085<br>Sox2 antibody: https://www.abcam.com/en-us/products/primary-antibodies/sox2-antibody-epr3131-ab92494<br>Otx2 antibody: https://www.rndsystems.com/products/human-otx2-antibody_af1979<br>Sox3 antibody: https://www.thermofisher.com/antibody/product/SOX3-Antibody-Polyclonal/PA5-35983<br>Nanog antibody: https://www.thermofisher.com/antibody/product/Nanog-Antibody-clone-eBioMLC-51-Monoclonal/14-5761-80<br>Phospho-Histone3 antibody: https://www.cellsignal.com/products/primary-antibodies/phospho-histone-h3-ser10-d2c8-xp-rabbit-mab/3377<br>Cleaved Caspase 3 antibody: https://www.cellsignal.com/products/primary-antibodies/cleaved-caspase-3-asp175-5a1e-rabbit-mab/9664<br>YAP: https://www.cellsignal.com/products/primary-antibodies/yap-d8h1x-rabbit-monoclonal-antibody/14074<br>PECAM-1 Alexa Fluor 488-conjugated antibody: https://www.rndsystems.com/products/mouse-rat-cd31-pecam-1-alexa-fluor-488-conjugated-antibody_fab3628g<br>SSEA-1 Alexa Fluor 405-conjugated antibody: https://www.rndsystems.com/products/human-mouse-ssea-1-alexa-fluor-405-conjugated-antibody-mc-480_fab2155v<br>Donkey anti-Goat IgG Alexa Fluor 488 conjugated antibody: https://www.thermofisher.com/antibody/product/Donkey-anti-Goat-IgG-H-L-Cross-Adsorbed-Secondary-Antibody-Polyclonal/A-11055<br>Goat anti-Rabbit IgG Alexa Fluor 488 conjugated antibody: https://www.thermofisher.com/antibody/product/Goat-anti-Rabbit-IgG-H-L-Highly-Cross-Adsorbed-Secondary-Antibody-Polyclonal/A-11034<br>Goat anti-Rat IgG Alexa Fluor 488 conjugated antibody: https://www.thermofisher.com/antibody/product/Goat-anti-Rat-IgG-H-L-Cross-Adsorbed-Secondary-Antibody-Polyclonal/A-11006<br>HRP-conjugated m-IgGk BP anti-mouse: https://www.scbt.com/p/m-igg-kappa-bp-hrp<br>HRP-conjugated mouse anti-rabbit : https://www.scbt.com/p/mouse-anti-rabbit-igg-hrp |

# Eukaryotic cell lines

Policy information about cell lines and Sex and Gender in Research

| | |
|---|---|
| Cell line source(s) | All experiments of this manuscript were conducted in the E14TG2a mouse embryonic stem cell line (https://www.atcc.org/products/crl-1821) derived by T Doetschman from 129/Ola mice in 1987. All mutant clones were generated from this parental line in house, with the exception of Brachyury-knockout, Eomesodermin-knockout, and Bra/Eomes-double-knockout cells, which were derived in the laboratory of Dr. Sebastian Arnold, University of Freiburg. |
| Authentication | None of the cell lines were authenticated. |

| | |
|---|---|
| Mycoplasma contamination | All cell lines used in this study were regularly tested negative for mycoplasma. |
| Commonly misidentified lines<br>(See ICLAC register) | No commonly misidentified cell lines were used in this study. |

## Plants

| | |
|---|---|
| Seed stocks | N/A |
| Novel plant genotypes | N/A |
| Authentication | N/A |

## Flow Cytometry

### Plots

Confirm that:

☒ The axis labels state the marker and fluorochrome used (e.g. CD4-FITC).

☒ The axis scales are clearly visible. Include numbers along axes only for bottom left plot of group (a 'group' is an analysis of identical markers).

☒ All plots are contour plots with outliers or pseudocolor plots.

☒ A numerical value for number of cells or percentage (with statistics) is provided.

### Methodology

| | |
|---|---|
| Sample preparation | Gastruloids were collected at indicated timepoints, washed in PBS-/-, and enzymatically dissociated using Accutase (Lab Clinics Capricorn Scientific, Cat# ACC-1B). Optimal dissociation was achieved by warming tubes containing gastruloids and Accutase to 37C in a water bath while shaking and eventually flicking the tubes during a time course of 5 minutes. Gastruloids were further dissociated mechanically using a p1000 pipette while adding flow buffer (PBS-/-, 2% BSA, 2mM EDTA). Resulting single cell suspensions were washed in flow buffer and pelleted by centrifugation at 200 xg for 3 minutes. Single gastruloid flow as depicted in Figure 2 of this manuscript was conducted by analyzing the dissociated gastruloids live at this step of the protocol.<br>For antibody stained flowcytometry multiple gastruloids of the same time point and batch were pooled and stained together. Single cells were washed one more time in PBS-/- followed by fixation in 4% PFA for 15 minutes. Fixed cells were washed 3 times in flow buffer, followed by blocking in 10% BSA in PBS-/- containing 0.1% Triton X-100 for 30 minutes at room temperature. After setting aside a fraction of the sample for use as negative or secondary antibody control, primary antibodies were diluted in 2% BSA in PBS-/- with 0.1%Triton X-100 (staining buffer). Cells were incubated with primary antibodies for 1 hour shaking at room temperature, followed by 3 washing steps in staining buffer. Fluorophore-conjugated secondary antibodies were diluted 1:500 in staining buffer unless otherwise stated and used to stain cells for 30 minutes at room temperature shaking. Cells were washed 3 times in staining buffer again and transferred to flow cytometry tubes for analysis. |
| Instrument | All samples were analyzed using a BD Bioscience LSRFortessa system. |
| Software | Analysis was conducted using: FlowJo (version 10.10.0) and BD FACS Diva (version 6.2) |
| Cell population abundance | Flow cytometry was used primarily as final readout and not to generate input of experimental starting material, with the exception of RNAseq samples isolated from mosaic gastruloids. Due to the temporal contraints of analysing sorted cells quickly before loss or change of RNAquality, purity of these samples was not validated after FACS isolation. |
| Gating strategy | All flow cytometry analysis started with pregating of cells vs debris using FSC-A vs SSC-A channels. This was followed by an exclusion of doublets using FSC-A vs FSC-H and SSC-A vs SSC-H. For population percentages, mCherry-positive and emiRFP670-positive events were then quantified. For phosho-Histone 3, SSEA-1, Pecam-1 staining, cells were pre-gated for mCherry or emiRFP670 positivity, and then analyzed separately in their respective stained channels. |

☒ Tick this box to confirm that a figure exemplifying the gating strategy is provided in the Supplementary Information.

