## [Peer Review File · Nature Cell Biology]

Mosaic gastruloids reveal a temporal restriction for developmental cell competition

Corresponding Author: Dr Joshua Frenster

Version 0:

Decision Letter:

*Please delete the link to your author homepage if you wish to forward this email to co-authors.

Dear Dr Frenster,

Your manuscript, "Temporal constraints for developmental cell competition in mosaic gastruloids", has now been seen by 3 referees, who are experts in stem cells (referee 1); embryonic development, cell competition (referee 2); and p53, development (referee 3). As you will see from their comments (attached below) they find this work of potential interest, but have raised substantial concerns, which in our view would need to be addressed with considerable revisions before we can consider publication in Nature Cell Biology.

Nature Cell Biology editors discuss the referee reports in detail within the editorial team, including the chief editor, to identify key referee points that should be addressed with priority, and requests that are overruled as being beyond the scope of the current study. To guide the scope of the revisions, I have listed these points below. We are committed to providing a fair and constructive peer-review process, so please feel free to contact me if you would like to discuss any of the referee comments further.

In particular, it would be essential to:

A) Address the novelty concern as raised by Reviewer 1;

B) All other referee concerns pertaining to strengthening existing data, providing controls, methodological details, clarifications and textual changes, should also be addressed.

C) Finally please pay close attention to our guidelines on statistical and methodological reporting (listed below) as failure to do so may delay the reconsideration of the revised manuscript. In particular please provide:

We would be happy to consider a revised manuscript that would satisfactorily address these points, unless a similar paper is published elsewhere, or is accepted for publication in Nature Cell Biology in the meantime.

- ensure that it conforms to our format instructions and publication policies (see below and <https://www.nature.com/nature/for-authors>).

- provide a point-by-point rebuttal to the full referee reports verbatim, as provided at the end of this letter.

- provide the completed Reporting Summary (found here <https://www.nature.com/documents/nr-reporting-summary.pdf>). This is essential for reconsideration of the manuscript will be available to editors and referees in the event of peer review. For more information see a

<http://www.nature.com/authors/policies/availability.html> or contact me.

Nature Cell Biology is committed to improving transparency in authorship. As part of our efforts in this direction, we are now requesting that all authors identified as 'corresponding author' on published papers create and link their Open Researcher and Contributor Identifier (ORCID) with their account on the Manuscript Tracking System (MTS), prior to acceptance. ORCID helps the scientific community achieve unambiguous attribution of all scholarly contributions. You can create and link your ORCID from the home page of the MTS by clicking on 'Modify my Springer Nature account'. For more information please visit www.springernature.com/orcid.

This journal strongly supports public availability of data. Please place the data used in your paper into a public data repository, or alternatively, present the data as Supplementary Information. If data can only be shared on request, please explain why in your Data Availability Statement, and also in the correspondence with your editor. Please note that for some data types, deposition in a public repository is mandatory - more information on our data deposition policies and available repositories appears below.

Link Redacted

We would like to receive a revised submission within six months.

We hope that you will find our referees' comments, and editorial guidance helpful. Please do not hesitate to contact me if there is anything you would like to discuss.

Best wishes,

Zhe Wang

Zhe Wang, PhD
Senior Editor
Nature Cell Biology

Tel: +44 (0) 207 843 4924
email: zhe.wang@nature.com

Reviewers' Comments:

Reviewer #1 (Remarks to the Author):

In this study, Frenster et al. investigated cell competition using a mosaic 3D mouse gastruloid model, which is a stem cell based model of gastrulation. The authors generated gastruloids composed of WT and p53KO cells and found that even a very low number (as few as two) of p53KO cells could act as the "winner" during cell competition. They also discovered that cell competition is temporally restricted to a window of 48–96 hours after aggregation during gastruloid development, corresponding to embryonic days E5.5–E7.5 in mice. Moreover, the models of more posterior stages of development did not exhibit any competition, supporting that developmental cell competition is specific to the onset of gastrulation. These findings provide new insights into cell fitness evaluation during mammalian embryogenesis. Regarding the mechanism, the authors demonstrated that cell competition in gastruloids is driven by intrinsic apoptosis, independent of proliferation rates, nutrient availability, reactive oxygen species, and not influenced by Nodal or ERK signaling, but can be moderately protected by Wnt and BMP signaling. Overall, the data are comprehensive and clearly presented, offering valuable insights into the use of the gastruloid 3D model for investigating developmental stage-specific cell competition. However, there are

several major concerns that prevent me from recommending this paper in its current form for publication in Nature Cell Biology. These are detailed in the comments below.

Major Comments:

1. The findings, while technically solid, do not substantially advance our understanding of cell competition. Reproducing known phenomena in a 3D gastruloid model, without uncovering new mechanisms or generating novel insights, falls short of what I would consider a significant contribution. The main strength of the gastruloid system lies in its ability to combine the scalability of 2D culture with the spatial organization and developmental complexity of the *in vivo* embryo. However, this potential is underutilized here. The authors do not use the model to explore new mechanistic hypotheses or address unanswered questions in the field. It is already well established that p53 loss enhances competitive behavior, and that cell competition occurs in the gastrulating epiblast. Previous studies have implicated factors such as cMyc, BMP signaling, autophagy, mitochondrial function, and aneuploidy. In my view, the pressing question is no longer whether competition occurs, but how it occurs—what molecular signals define a “winner” or “loser,” and how cells communicate and act on this information. These are the deeper mechanistic questions the field is now poised to answer, and the current study does not move us closer to that goal.

2. While the authors present a thorough analysis of potential mechanisms, most of the experimental results are negative or descriptive in nature. As a result, the key factors driving the competitive interactions between WT and p53KO cells remain unclear. In addition, the specific cell types involved in the competition have not been clearly defined. Interestingly, this work is reminiscent of findings from a recent study (PMID: 33508854), which showed that cell competition between human and mouse pluripotent stem cells occurs specifically in the primed, but not naïve, state. Although the underlying mechanisms are likely different, it would be valuable to explore whether the competition observed in this study is also limited to primed PSCs. For example, co-culturing WT and p53KO EpiSCs under conditions that support the primed state (e.g., with FGF2 and a Wnt antagonist such as IWR1 or XAV939) could help determine whether primed PSCs mediate the observed competition. If competition is observed in this context, EpiSC co-cultures may serve as a more defined system to dissect the molecular mechanisms involved.

3. The authors attempt to probe the role of signaling pathways in cell competition by treating cell aggregates with various agonists and antagonists. However, this section has several major issues that limit the interpretability of the results:

a) These treatments lead to different lineage outcomes and cell compositions. Given that cell competition is highly context- and cell type-dependent, it's unclear whether any meaningful conclusions can be drawn from comparisons across such divergent samples.

b) Both WT and p53KO cells are simultaneously exposed to these pathway modulators, making it impossible to determine whether the observed effects are due to changes in the “winner,” the “loser,” or both. Without cell-type-specific perturbation, mechanistic insights remain speculative.

c) All treatment comparisons are made relative to DMSO controls, not CHIR. This is problematic because gastruloid formation, unlike simple embryoid body differentiation, requires CHIR as a baseline condition. Any chemical perturbations should be performed in the presence of CHIR and compared to a CHIR-only control. That said, layering additional agonists or antagonists onto CHIR may introduce confounding effects similar to those raised in point (a).

Given these limitations, I recommend removing this section unless the authors can design more targeted and interpretable experiments.

• The authors claim that glucose level manipulation did not impact cell competition as seen in figure 4H, but strikingly they seem to leave out that on almost complete glucose deprivation to 0.5mM that cell competition is almost completely abolished. This is a fascinating result that is not explored. Admittedly when comparing WT+WT to WT+p53KO at different levels it would appear that there little effect of glucose on the cell competition relative to WT+WT. However I believe that this is the wrong scientific comparison being made. Instead of WT+WT vs WT+p53KO in 21mM, what should be compared is WT+p53KO 12mM vs. WT+p53KO 21mM. If you compare the WT+p53KO condition across glucose levels, it seems to me that clearly you have an effect where lowering the amount of glucose drives lower cell competition, even so much as to almost completely abolish it in the 0.5mM condition.

Minor Comments:

1. Lines 59-65, it is relevant to cite the study by Zheng et al (PMID: 33508854) and briefly mention interspecies primed PSC competition mediated by innate immune response in loser cells.

2. Lines 66-68, it is relevant to cite the study by Dejosez et al (PMID: 24030493).

3. Lines 99-114, is the cell competition observed in confluent naïve cultures contact-dependent. To address this, the authors should consider using transwell co-cultures.

4. The scale bar is missing in Figure 1g.

5. It is puzzling that WT-mCherry cells show a longer doubling time compared to WT-emiRFP cells (Figure 1b), yet they form larger gastruloids (Figure 1i). This apparent contradiction raises questions about the factors driving gastruloid growth in this system. The authors should clarify how a slower-proliferating cell line produces larger structures and provide possible explanations—such as differences in cell survival, differentiation dynamics, or organization—that might account for this outcome.

6. In Figure 2h, the WT-WT co-culture data show that WT-mCherry and WT-emiRFP cells maintain a stable ratio up to 72 hours. However, after that point, WT-mCherry cells begin to dominate, indicating a potential competitive advantage even between two WT lines. This raises concerns about underlying differences between these cell lines that the authors do not address. The fact that one WT line consistently outcompetes the other suggests there may be alternative mechanisms at play that remain unexplored. The authors should investigate this further, for instance, by staining for markers of cell proliferation and apoptosis to determine whether actual competition is occurring.

7. Another issue is the authors' claim that “the same WT-mCherry population can co-develop with other WT cells by equally contributing to the final gastruloid.” This is directly contradicted by their own data: both immunofluorescence (Figure 2A) and flow cytometry (Figure 2H) show that WT-mCherry cells become the dominant population in 3D gastruloids. Furthermore,

- this dominance pattern conflicts with the results shown in Extended Data Figure 1, where WT-emiRFP appears to gain the upper hand during co-culture (e.g., ED Fig. 1e). This discrepancy raises an important question: if WT-mCherry ESCs proliferate more slowly in naive culture, how do they come to dominate in 3D gastruloids? Do they proliferate or differentiate more efficiently during gastruloid formation? The authors should clarify this point and reconcile the inconsistent results.
8. The authors repeatedly refer to Figure 2K, but no such figure exists in the manuscript. It's likely they intended to reference Figure 2I.
 9. In Figure 3a, it appears the authors are showing a chimeric gastruloid composed of WT-mCherry and WT-emiRFP cells. If that is the case, the individual mCherry and emiRFP fluorescence channels should be shown alongside the pH3 staining. This is important to assess how proliferation (as marked by pH3) is distributed between the two cell populations.
 10. Figures 3f–3g show that Bcl2 overexpression markedly increases the number of WT-emiRFP cells—particularly at 144h—in chimeric gastruloids formed with WT-mCherry cells. This suggests that, under normal conditions, WT-mCherry cells outcompete WT-emiRFP cells even in the absence of p53 knockout.
 11. ED Fig. 5a-5b, the authors should provide quantification of the percentages of Active Caspase 3 positive cells in different cell line in the chimeric gastruloids.
 12. In Figures 4g–4h, the authors state that glucose level manipulation does not impact cell competition. However, in the 0.5 mM glucose condition shown in Figure 4h, competition appears to be nearly abolished—an intriguing observation that is not further explored. This result suggests that low-glucose conditions may alter the competitive dynamics, potentially by affecting proliferation or survival of p53KO cells. The authors should investigate this more closely. For example, is reduced competition due to impaired proliferation, increased apoptosis, or altered metabolism in p53KO cells under glucose restriction?
 13. Lines 305-306, it is relevant to cite the study by Price et al. (PMID: 34407428)
 14. Lines 352-354, Figure 6c and ED Figure 8b, from the presented data, it is not clear to me “non-autonomously delaying enighboring WT-mCherry cells in mosaic gastruloids” is correct.
 15. Please clarify the statistical methods used in the Methods section or figure legends. For example, if a t-test or one-way analysis of variance (ANOVA) was used, please specify.

Reviewer #2 (Remarks to the Author):

Gastruloids generated from mouse embryonic stem cells provide a useful system with which to investigate mechanisms of gastrulation. In this manuscript, Frenster and co-workers generated modified clones of E14Tg2A ESCs, initially to include fluorescent labels and knockout of P53 to use as tools to investigate aspects of transitions from naïve to primed pluripotency during pre- and post-gastrulation stages. As preliminary validation of the tools generated, the authors performed cell competition assays in 2D and confirmed previous studies showing that, in the presence of P53KO cells, WT clones were dramatically out-competed within a specific time scale, corresponding to pre- and peri-gastrulation in mouse embryos (~E5.5-E7.5). Using 3D gastruloids, they revealed that a surprisingly small number of P53KO cells were required to overcome more than half of the WT cells within a short time window, within 120 hours of culture. The authors note that gastruloids produced from P53KO or WT clones separately exhibit different morphology; those from P53KO cells tended to be bigger and more irregular than WTs, but the effect of P53KO cells on WTs in mosaic gastruloids from 72 hours after aggregation was much more dramatic. This can be blocked via inhibition of apoptosis in ‘loser’ cells via Bcl2 overexpression, as has been shown previously in 2D culture. This system allowed the authors to investigate the window of opportunity for rescue through Bcl2. Inducible overexpression of Bcl2 in ‘loser’ cells from 0-48 hours of aggregation or 48-72 hours appears to be protective from cell competition. Given the long half-life of BCL2 protein, this probably suggests a critical window between 48-96 hours during which cell competition induces apoptosis. Accordingly in WT-KO mosaics, a peak of Noxa expression is detected 48 hours after aggregation, followed by a peak of p21 at 72 hours and Bcl2 at 96 hours. The pulse of Chiron, typically administered from 48-72 hours, is somewhat protective, with increased levels of competition when not provided, probably because Chiron increases the developmental pace of cells in this context, driving them out of the developmental window in which cell competition occurs.

The authors generate ‘heterochronic’ gastruloids by mixing and reaggregating cells from time mis-matched gastruloids. They find that 96 hours after aggregation P53KO cells lose the capacity to outcompete 48 hour wild-type cells, while late (72-96 hours) WT cells are no longer outcompeted by 48 hour P53KO cells, whereas early cells (24 hours) are both able to influence and be influenced by their 48 hour neighbors. As expected, cell competition was not observed if the cells were in the EpiSC state; the resulting structures comprised mostly spinal cord and brain tissue, confirming their temporal position outside the developmental window of cell competition. Interestingly, P53KO cells tended to differentiate preferentially to mesoderm rather than endoderm. Transcriptomic analysis revealed bias at 120h for P53KO cells to form paraxial mesoderm and caudal epiblast rather than blood lineage or PGC-like cells. Both WT and P53KO clones exhibited phagocytosis, implying that this mechanism was unlikely to be responsible for driving cell competition. From 48 hours, WT cells appeared to stagnate and undergo apoptosis, as has been reported for ‘loser’ cells in chimeric mouse embryos at the onset of gastrulation. P53KO cells exhibit impaired growth in response to blockade of apoptosis in WT cells or when Bcl2 is expressed. Apoptotic cells tended to cluster in the central region of the gastruloid. Since metabolic processes have been shown to play a role in cell competition, the effect of glucose concentration was tested, but found to be of little significance, as was ROS and cytoskeletal tension. Absence of BMP signalling or Chiron appeared to delay gastruloid cell competition. Also, P53KO cells were delayed in exit from naïve pluripotency, implicating some disruption of the networks in P53KO cells. By 72h WT cells were found to lose responsiveness to cell competition. However, the period of permissive cell competition can be varied by modulating Wnt signalling. This work is largely well planned and executed, resulting in novel information likely to have an important impact on the developmental and stem cell biology fields, particularly in the context of modelling early human postimplantation

development. However, there are several issues that the authors should address.

Presentation of methods:

1. The code relating to the RNA-sequencing analysis is largely clear and well documented, using standard techniques. However, it is unclear where the reference embryo data used for the spatial analysis (anterior-posterior identity) comes from. It appears that the data is annotated with spatial information that does not seem to be present in the data referenced.
2. The code, and methods, relating to most of the image analysis is insufficient to reproduce this work. The authors use a tool (QLIVECELL) which does not appear to have been validated in this work or in other publications. The code and documentation provided are not sufficient to run the validation tests, and the methods section does not detail which functions were called or what parameters were used. It is unclear which steps were performed using this tool and which were performed as downstream analysis.
3. Throughout most of this manuscript, statistical analyses are reported without indicating what statistical test has been performed.
4. There does not appear to be any explanation of the generation of the Bcl2, Fas, FasL or Fadd cell lines in the methods.

Points for clarification:

1. In Figure 5c, it is found that Chiron stimulation from 48-72 hours post-aggregation strongly weakens the effect of cell competition compared to DMSO, while BMP4 treatment provides a milder protective effect. In Figure 4k, the authors see a similarly mild effect of cell competition in Chiron-only treated mosaic gastruloids. This is contrasted by the very strong and consistent effect seen in Figures 1h, 2i, 3n, 4c, and 4p. This is unexpected since each experiment is performed with technical and biological replicates which appear to show relatively low variability. Are the authors able to identify any reason why the cell competition effect is so much weaker in just these two experiments?

Minor points and typos:

2. In Fig.1g, why are the gastruloids green only? Should they not be (equal) mixtures of pink and green?
3. The legend for Fig. 2 states that cell competition is temporally, but not spatially restricted, whereas the main text says that cell competition occurs predominantly in the central region of the gastruloid
4. Please show the variability in Figure 2e
5. Figure 2j is referred to as Figure 2k in the text
6. The treatment times in Figure 5c differ between the figure legend/methods and the main body text
7. Could the authors please elaborate on the blebby appearance of P53KO gastruloids in Fig. 5d?
8. Line 438: 'Blc2' instead of 'Bcl2'
9. In Figure 6k it currently appears as if there are 0 mCherry+ cells when 48 hour WT cells are mixed with 24 hour P53KO cells or with 72 hour WT cells. These conditions are apparently also tested in Extended Figure 8. Please could the authors clarify this confusion?

Reviewer #3 (Remarks to the Author):

In this manuscript, Frenster et al. develop a mouse 3D gastruloid model for studying cell competition during gastrulation and investigate the role of the tumor suppressor p53 in this process. Gastruloids were generated from ES cells grown in specific culture conditions for 5 days to make elongated and polarized structures representing embryos at E8.5. The authors first examine the effects of co-culture of WT-mCherry (labeled green) ES cells with either control WT-emiRFP ES cells or isogenic p53KO-emiRFP ES cells. They find that by coculturing the WT-mCherry ES cells with the p53KO-emiRFP ES cells, the p53KO cells are supercompetitors, with as few as 2 p53KO cells impairing growth/survival of the remaining wild-type cells in the gastruloid. They show that this phenomenon is not due to spatial positioning of cells, but due to the cells' existence in a specific time window of development centered around 48h-96h post aggregation, which corresponds to E5.5-E7.5. The authors show that expression of Bcl2 in the WT-mCherry cells blocks the ability of p53KO cells to supercompete, suggesting that p53 KO cells promote apoptosis in neighboring WT cells. They show further that intrinsic, not extrinsic, apoptosis in the WT-mCherry cells is the cause of cell competition, and they rule out several causes of this phenomenon, including nutrient availability, ROS, and mechanical compression. Finally, the authors show that increased Wnt signaling results in decreased cell competition.

Overall, this study provides important insight into the process of gastrulation and the factors that influence cell fitness during gastrulation. This work provides important new perspectives and would be of great interest to the readership of Nature Cell Biology. However, before publication can be recommended, a number of points and critical controls must be addressed in a revised manuscript to improve the story. Importantly, there are a lot of places where clarity could be enhanced in the figures and text.

Specific points:

1. Why were two RFPs used (mCherry and emiRFP) instead of two fluorescent proteins with more different excitation spectra?
2. Were the WT-emiRFP cells subjected to the same process as the p53KO-emiRFP cells? That is an important control for the experiment given the selection process that is needed to make the p53KO cells. Even better would be a rescue experiment where p53 is restored in the p53KO-emiRFP cells. This would rule out any secondary mutations that might influence the behavior of these p53KO-emiRFP cells.

3. In discussing the p63/p73/p53 triple knockout phenotypes, it is important to also consider and cite the study by Van Nostrand et al. (Cell Death and Differentiation 2017).
4. Is there some way to quantitate E-cadherin and FoxA2 in Ext. Fig. 2b/c? Also, why are endoderm markers relatively unchanged between p53KO and WT cells in Ext. Fig. 2e, which contrasts with what is mentioned about Ext. Fig. 2 b/c?
5. To solidify the proliferation data, it is important to examine other proliferation markers than p-H3, such as Ki67 or BrdU.
6. Can the authors do assays to show definitively that Bcl2 is acting in the ES cells to inhibit apoptosis? That is an important control for the mechanism they propose.
7. The argument that the wild-type cells are affected by intrinsic apoptosis could be stronger. For example, inhibiting Caspase 3 or 9 should have an effect on competition but inhibiting Caspase 8 should not. The FADD deletion experiments are not sufficient to make this claim, especially since no western blot confirming ablation of the protein is shown.
8. Can the authors provide proof that ROS was inhibited in Figure 4/5?
9. The text describing Fig. 5c indicates that modulating ActivinA, Nodal, and TGFb signaling had a non-significant effect on cell competition, but the stats on these compared to DMSO are not clear. In general, the effects of the various factors are not shown in the clearest way. Inhibitors and activators of pathways should be clearly separated and explained. Also, where is the proof that these activators and inhibitors are inducing/inhibiting the expected pathways?
10. Fig. 6n is very important for showing specific characteristics of each time point of gastruloid development, as it describes essential genes for apoptosis, cell cycle arrest, etc. However, it is not clear how this analysis was done. Was this bulk RNA-seq or something else? If bulk RNA-seq was done on cells collected from each time point, pathway analysis should be done to define pathways at different timepoints. Also, why would p21 be induced after Puma and Noxa, which are lower affinity p53 target genes? Perhaps p21 is induced in a p53-independent manner?
11. In the discussion the authors state that “loser cells respond to fitter cells by stabilizing p53 protein”. The authors should show that p53 is being stabilized in loser cells. Also, does knockout of p53 target genes improve the fitness of the loser cells?

Additional minor points:

1. Extended Fig 1g schematic is confusing, as it is not clear why one big cell rather than a cluster of cells is indicated in the right side of the image.
2. “ChIR” treatment is first mentioned in Ext. Fig. 2b/c, but in the text is not described until the last couple figures. This should be explained when it first mentioned.
3. Fig. 2k cited in text, but there is no Fig.2k.
4. In Ext. Fig 3c, right panel, the p53KO labels appear to be mixed up.
5. In general, the labeling of panels in figures and the figure legends could be improved to make the paper less confusing to read. For example, in Fig. 1D, each clone type could be labeled with the actual name (like WT-mCherry).

Methods should be written concisely, but should contain all elements necessary to allow interpretation and replication of the results. As a guideline, Methods sections typically do not exceed 3,000 words. The Methods should be divided into subsections listing reagents and techniques. When citing previous methods, accurate references should be provided and any alterations should be noted. Information must be provided about: antibody dilutions, company names, catalogue numbers and clone numbers for monoclonal antibodies; sequences of RNAi and cDNA probes/primers or company names and catalogue numbers if reagents are commercial; cell line names, sources and information on cell line identity and authentication. Animal studies and experiments involving human subjects must be reported in detail, identifying the committees approving the protocols. For studies involving human subjects/samples, a statement must be included confirming that informed consent was obtained. Statistical analyses and information on the reproducibility of experimental results should be provided in a section titled "Statistics and Reproducibility".

All Nature Cell Biology manuscripts submitted on or after March 21 2016 must include a Data availability statement as a separate section after Methods but before references, under the heading "Data Availability". For Springer Nature policies on data availability see <http://www.nature.com/authors/policies/availability.html>; for more information on this particular policy see <http://www.nature.com/authors/policies/data/data-availability-statements-data-citations.pdf>. The Data availability statement should include:

- Accession codes for primary datasets (generated during the study under consideration and designated as "primary accessions") and secondary datasets (published datasets reanalysed during the study under consideration, designated as "referenced accessions"). For primary accessions data should be made public to coincide with publication of the manuscript. A list of data types for which submission to community-endorsed public repositories is mandated (including sequence, structure, microarray, deep sequencing data) can be found here <http://www.nature.com/authors/policies/availability.html#data>.
- Unique identifiers (accession codes, DOIs or other unique persistent identifier) and hyperlinks for datasets deposited in an approved repository, but for which data deposition is not mandated (see here for details <http://www.nature.com/sdata/data-policies/repositories>).
- At a minimum, please include a statement confirming that all relevant data are available from the authors, and/or are included with the manuscript (e.g. as source data or supplementary information), listing which data are included (e.g. by figure panels and data types) and mentioning any restrictions on availability.
- If a dataset has a Digital Object Identifier (DOI) as its unique identifier, we strongly encourage including this in the Reference list and citing the dataset in the Methods.

We recommend that you upload the step-by-step protocols used in this manuscript to protocols.io. More details can be found at <https://www.protocols.io/help/publish-articles>.

All imaging data should be accompanied by scale bars, which should be defined in the legend.

Cropped images of gels/blots are acceptable, but need to be accompanied by size markers, and to retain visible background signal within the linear range (i.e. should not be saturated). The boundaries of panels with low background have to be demarked with black lines. Splicing of panels should only be considered if unavoidable, and must be clearly marked on the figure, and noted in the legend with a statement on whether the samples were obtained and processed simultaneously. Quantitative comparisons between samples on different gels/blots are discouraged; if this is unavoidable, it should only be performed for samples derived from the same experiment with gels/blots were processed in parallel, which needs to be stated in the legend.

EXTENDED DATA FIGURES - When re-submitting your manuscript, please ensure that any supplementary figures and tables that are crucial to the manuscript's conclusions are converted into Extended Data figures and tables to increase visibility of these data. Extended Data figures and tables are online-only (present in the online PDF and full-text HTML versions of the paper), peer-reviewed display items that provide essential background to the article but are not included in the main article due to space constraints. A maximum of ten Extended Data display items (figures and tables) is permitted.

The total number of Supplementary Figures (not including the “unprocessed scans” Supplementary Figure) should not exceed the number of main display items (figures and/or tables (see our Guide to Authors and March 2012 editorial <http://www.nature.com/ncb/authors/submit/index.html#suppinfo>; <http://www.nature.com/ncb/journal/v14/n3/index.html#ed>). No restrictions apply to Supplementary Tables or Videos, but we advise authors to be selective in including supplemental data.

GUIDELINES FOR EXPERIMENTAL AND STATISTICAL REPORTING

REPORTING REQUIREMENTS – We are trying to improve the quality of methods and statistics reporting in our papers. To that end, we are now asking authors to complete a reporting summary that collects information on experimental design and reagents. The Reporting Summary can be found here <https://www.nature.com/documents/nr-reporting-summary.pdf> If you would like to reference the guidance text as you complete the template, please access these flattened versions at <http://www.nature.com/authors/policies/availability.html>.

We strongly recommend the presentation of source data for graphical and statistical analyses as a separate Supplementary Table, and request that source data for all independent repeats are provided when representative experiments of multiple independent repeats, or averages of two independent experiments are presented. This supplementary table should be in Excel format, with data for different figures provided as different sheets within a single Excel file. It should be labelled and numbered as one of the supplementary tables, titled “Statistics Source Data”, and mentioned in all relevant figure legends.

Version 1:

Decision Letter:

Our ref: NCB-A57719A

21st January 2026

Dear Dr. Frenster,

Thank you for submitting your revised manuscript "Mosaic gastruloids reveal a temporal restriction for developmental cell competition" (NCB-A57719A). It has now been seen by the original referees and their comments are below. The reviewers find that the paper has improved in revision, and therefore we'll be happy in principle to publish it in Nature Cell Biology, pending minor revisions to satisfy the referees' final requests and to comply with our editorial and formatting guidelines.

Meanwhile, you may modify the manuscript files and figures according to editorial guidelines below:

1. As we allow up to 10 extended data figures and for better accessibility, please move key data from supplementary figures to extended data figures;
2. Please move and supply unprocessed gels in supplementary figures separately as source data files;
3. Please provide all figures separately from main text, one figure per file;
4. Please separate figure legends from the figures;
5. Please provide a code availability statement and ensure to make qlivecell more broadly accessible
6. For better legibility, please ensure that all figures adhere to a maximum page size of roughly 180mm wide x 200mm high to fit standard page format and use a font size of no smaller than 7pt Arial or Helvetica throughout the figures.

Thank you again for your interest in Nature Cell Biology Please do not hesitate to contact me if you have any questions.

Sincerely,

Zhe Wang, PhD
Senior Editor
Nature Cell Biology

Tel: +44 (0) 207 843 4924
email: zhe.wang@nature.com

Reviewer #1 (Remarks to the Author):

In this revised manuscript, the authors have comprehensively addressed all previous concerns. The study successfully demonstrates that utilizing gastruloids—and stem cell-based embryo models more broadly—effectively bridges the gap between traditional 2D cultures and in vivo embryogenesis. By using the 3D gastruloid model, the authors provide a more physiologically relevant platform for studying cell competition, avoiding the "incomplete or artificial readouts" often associated with 2D systems. Key Strengths of the Revision. The data highlighting the temporal nature of cell competition and the differences between 2D vs. 3D gastruloids are particularly compelling. The inclusion of new results regarding p53-degron and glucose manipulation significantly strengthens the mechanistic depth of the paper. All remaining technical points have been adequately clarified. Given the high quality of the revisions and the importance of the findings, I recommend the manuscript for publication.

Reviewer #2 (Remarks to the Author):

We are very satisfied with the in depth and thoughtful response of the authors to the criticism and suggestions from all reviewers. Our points have all been addressed, so happy to recommend publication of the work in its revised state.

Reviewer #2 (Remarks on code availability):

The code was reviewed and found to work for the information required to support the data presented. However, access to qlivecell is apparently not accessible more broadly at present.

Reviewer #3 (Remarks to the Author):

The authors have thoroughly addressed my comments. This paper will be a great contribution to the field!

Version 2:

Decision Letter:

Dear Dr Frenster,

I am pleased to inform you that your manuscript, "Mosaic gastruloids reveal a temporal restriction for developmental cell competition", has now been accepted for publication in *Nature Cell Biology*.

Over the next few weeks, your paper will be copyedited to ensure that it conforms to *Nature Cell Biology* style. Once your paper is typeset, you will receive an email with a link to choose the appropriate publishing options for your paper and our Author Services team will be in touch regarding any additional information that may be required.

Publication is conditional on the manuscript not being published elsewhere and on there being no announcement of this work to any media outlet until the online publication date in *Nature Cell Biology*.

Please note that *Nature Cell Biology* is a Transformative Journal (TJ). Authors may publish their research with us through the traditional subscription access route or make their paper immediately open access through payment of an article-processing charge (APC). Authors will not be required to make a final decision about access to their article until it has been accepted. [Find out more about Transformative Journals](https://www.springernature.com/gp/open-research/transformative-journals)

Authors may need to take specific actions to achieve compliance with funder and institutional open access mandates. If your research is supported by a funder that requires immediate open access (e.g. according to [Plan S principles](https://www.springernature.com/gp/open-science/plan-s-compliance) or the [NIH public access policy](https://www.springernature.com/gp/open-science/us-federal-agency-compliance)) then you should select the gold OA route, and we will direct you to the compliant route where possible. Because authors warrant under our subscription licensing terms that they haven't committed to licensing any version of their article under a licence inconsistent with the terms of our agreement – including the applicable embargo period – publication under the subscription model isn't suitable for authors whose funders require no embargo.

If your paper includes color figures, please be aware that in order to help cover some of the additional cost of four-color reproduction, Nature Portfolio charges our authors a fee for the printing of their color figures. Please contact our offices for

exact pricing and details.

If you have not already done so, we strongly recommend that you upload the step-by-step protocols used in this manuscript to protocols.io (<https://protocols.io>), an open online resource that allows researchers to share their detailed experimental know-how. All uploaded protocols are made freely available and are assigned DOIs for ease of citation. Protocols and Nature Portfolio journal papers in which they are used can be linked to one another, and this link is clearly and prominently visible in the online versions of both. Authors who performed the specific experiments can act as primary authors for the Protocol as they will be best placed to share the methodology details, but the Corresponding Author of the present research paper should be included as one of the authors. By uploading your Protocols onto protocols.io, you are enabling researchers to more readily reproduce or adapt the methodology you use, as well as increasing the visibility of your protocols and papers. You can also establish a dedicated workspace to collect your lab Protocols. Further information can be found at <https://www.protocols.io/help/publish-articles>.

Nature Cell Biology encourages authors presenting evidence for cell, biological, molecular, and genetic interactions to consider communicating these findings using Biofactoid (<https://biofactoid.org/>). This tool helps users share a searchable representation of interactions (e.g. binding, gene expression, post-translational modification) between genes, gene products, or chemicals. Information added to Biofactoid, with author attribution, is shared on social media and public databases, such as Pathway Commons, where it can be discovered and analyzed in the context of a large and growing corpus of knowledge.

With kind regards,

Zhe Wang, PhD
Senior Editor
Nature Cell Biology

Tel: +44 (0) 207 843 4924
email: zhe.wang@nature.com

** Visit the Springer Nature Editorial and Publishing website at http://editorial-jobs.springernature.com?utm_source=ejp_NCB_email&utm_medium=ejp_NCB_email&utm_campaign=ejp_NCB for more information about our career opportunities. If you have any questions please click [here](mailto:editorial.publishing.jobs@springernature.com).

Reviewers' comments and point-by-point response:

Reviewer #1 (Remarks to the Author):

In this study, Frenster et al. investigated cell competition using a mosaic 3D mouse gastruloid model, which is a stem cell based model of gastrulation. The authors generated gastruloids composed of WT and p53KO cells and found that even a very low number (as few as two) of p53KO cells could act as the "winner" during cell competition. They also discovered that cell competition is temporally restricted to a window of 48–96 hours after aggregation during gastruloid development, corresponding to embryonic days E5.5–E7.5 in mice. Moreover, the models of more posterior stages of development did not exhibit any competition, supporting that developmental cell competition is specific to the onset of gastrulation. **These findings provide new insights into cell fitness evaluation during mammalian embryogenesis.** Regarding the mechanism, the authors demonstrated that cell competition in gastruloids is driven by intrinsic apoptosis, independent of proliferation rates, nutrient availability, reactive oxygen species, and not influenced by Nodal or ERK signaling, but can be moderately protected by Wnt and BMP signaling. **Overall, the data are comprehensive and clearly presented, offering valuable insights into the use of the gastruloid 3D model for investigating developmental stage-specific cell competition.** However, there are several major concerns that prevent me from recommending this paper in its current form for publication in Nature Cell Biology. These are detailed in the comments below.

Authors (in blue): We thank the reviewer for recognizing the full potential of gastruloids as a system. We agree that the use of gastruloids allows us to study cell competition and development from angles otherwise inaccessible. In our current study we specifically make use of the modularity of gastruloids to advance our understanding of cell competition in ways that we could not in mouse embryos or 2D systems.

Major Comments:

1. The findings, while technically solid, do not substantially advance our understanding of cell competition. Reproducing known phenomena in a 3D gastruloid model, without uncovering new mechanisms or generating novel insights, falls short of what I would consider a significant contribution. The main strength of the gastruloid system lies in its ability to combine the scalability of 2D culture with the spatial organization and developmental complexity of the in vivo embryo. However, this potential is underutilized here. The authors do not use the model to explore new mechanistic hypotheses or address unanswered questions in the field. It is already well established that p53 loss enhances competitive behavior, and that cell competition occurs in the gastrulating epiblast. Previous studies have implicated factors such as cMyc, BMP signaling, autophagy, mitochondrial function, and aneuploidy. In my view, the pressing question is no longer whether competition occurs, but how it occurs—what molecular signals define a “winner” or “loser,” and how cells communicate and act on this information. These are the deeper mechanistic questions the field is now poised to answer, and the current study does not move us closer to that goal.

We appreciate this comment as it has helped us look at our study again in comparison with others and highlight its value. Sometimes, the field may want to see a new molecular player but when dealing with such complex processes as ‘cell competition’, revealing systemic and physiological variables is just as important. Here we highlight what we believe are novel findings, including our new additional findings in the revised version of this manuscript:

A central focus of this manuscript is the **precise and causal characterization of a developmental window permissive for cell competition.** We address this aspect not only by descriptive means and correlation as has been done before, but by **experimentally dissecting it through use of the modular nature of the gastruloid system.**

To start, we used time-resolved Bcl2 rescue to demonstrate that no cell competition-related apoptosis occurs until after cells progress past the stage of primed pluripotency (Fig. 3r; Extended Data Fig. 3p). At the other end, our introduction of Epi-Gastruloids, a new developmental 3D structure, demonstrates the

inability to compete once 3D developmental structures have moved past the onset of gastrulation (Fig. 7). The ability to control developmental progression speed in gastruloids (Fig. 6c-d; Extended Data Fig. 6a-b), uniquely allowed us to link cell competition effect size to the duration spent within this stage permissive for cell competition. Furthermore, the modularity of gastruloids uniquely allowed us to generate heterochronic 3D developmental structures (Fig. 6g-m; Extended Data Fig. 6c-h) to reveal for the first time that both winner and loser cells must reside within the same cell competition permissive window of time, an unthinkable experiment in mouse embryos. This observation not only demonstrates at new temporal resolution the inability of cells to compete outside of our proposed developmental window but also highlights something important regarding the nature of cell-cell communication during cell competition, namely, that both sides, the signal sending and receiving cells, only communicate at this stage, and that mismatch of one side is sufficient to abolish cell competition.

Mechanistically, we reveal a temporally opportune expression pattern of the apoptosis and stress response machinery concurrent with the downregulation of epigenetic repressors of p53 transcriptional targets at the onset of gastrulation (Fig. 6n), followed by an upregulation of anti-apoptotic genes at mid-gastrulation. The progression from primed pluripotency into gastrulation during which cell competition occurs depends on the transcriptional programs governed by Brachyury and Eomesodermin (PMID: 31792383; PMID: 37633271). We now demonstrate that Brachyury, and particularly Brachyury/Eomes-double knockout cells are almost entirely unaffected by cell competition and neighboring supercompetitors (Fig. 6o). Fittingly, double knockout cells lack the Noxa and p21 expression observed in WT cells at 72h of gastruloid formation (Extended Data Fig. 6 i-j). This sheds new light on a possible involvement of the Brachyury transcriptional program during cell competition, while strengthening our claims toward the temporal dynamics of cell competition.

Secondly, as the reviewer points out, it is already known that long-term p53 knockout cells act as supercompetitors in 2D cultures, however, in addition to the caveat on 2D systems we express below, it previously remained unclear whether cell competition was a direct effect of acute p53 protein levels, or an indirect effect caused by long-term loss of *Trp53*. To answer this question, we generated a new p53-degron based on an optimized version of the plant OstIR system (OstIR1(F74G)-mAID2). With this, we demonstrate for the first time that short-term and reversible, partial reductions of p53 protein levels, in otherwise WT cells, are fully sufficient to generate supercompetitor states, thereby providing the most direct experimental validation that acute relative differences in p53 protein levels specifically at the onset of gastrulation determine competitive outcomes (Fig. 8). While this may have been assumed before, we believe that this manuscript will be the reference for the first clear experimental demonstration thereof.

Thirdly, we benchmark our new 3D developmental cell competition system by comparing it to previous 2D co-culture setups and not only find many mechanistic differences but also flag important issues with previous (intra-species) 2D systems. We show that cell competition in differentiating 2D layers is dependent on cell density and does not occur when mixed colonies are passaged instead of grown to over-confluence (Extended Data Fig. 2g-n). In contrast, in 3D gastruloids, total gastruloid size is irrelevant and competition instead depends on developmental progression stages. Additionally, inhibition of cytoskeletal machineries has no effects on competition in 3D gastruloids, and YAP nuclear-to-cytosolic ratios are unaltered in winner or loser settings, which contrasts previous 2D studies. This is important as many of the conclusions in the field have been drawn from 2D studies, while their conclusions should be carefully considered, as they do not represent competition in a steady state situation as in the embryo or, as we show here, in gastruloids. Of note, our remark excludes recent studies on inter-species chimerism, which reported that sub-confluent species-mixed colonies of mouse and human cells undergo cell competition. Inter-species competition may however reflect different biology from intra-species competition. For the reasons above, we believe that even those aspects of our study that rigorously confirm previous observations from confluent 2D cultures in a 3D developmental context, add valuable and important confidence to the field.

Additionally, our study demonstrates that cell competition is independent of winner cell growth rates or neighbor mortality. This has not been demonstrated before and we believe it to be an important contribution to a field which is highly sensitive to artifacts and to the confusion between active elimination and passive overgrowth of individual clones. Through modulation of neighbor cell numbers, and their Bcl2-mediated apoptosis resistance, we demonstrate that WT cells in gastruloids do not respond to having either 400 mortal neighbors, or 17,000 apoptosis-resistant neighbors (Fig. 3o). This not only holds true in

WT+WT cocultures, but also in p53KO supercompetitors, where glucose limitation results in altered growth rates without affecting the relative cell competition effect on neighboring WT cells (Fig 4m-n; Extended Data Fig. 4m-n). These results support that cell competition in developmental 3D structures is not defined by the growth rate of supercompetitor neighbors and sets it apart from previous 2D co-culture systems.

The exact (FACS single cell sorted) control over cell numbers in gastruloids allowed for a precise stoichiometry analysis during supercompetition in a 3D developmental system, which as the reviewer highlights above demonstrated for the first time that as few as two supercompetitor cells per developmental structure suffice to impair growth of their much more numerous WT neighbors (Fig. 1h). We believe that such precise stoichiometry has neither been demonstrated before, nor would it be feasible in mouse embryos at any reasonable throughput.

Despite this drastic effect size in gastruloids, we demonstrate that supercompetition, is phenotypically silent in the surviving WT population, without preferentially affecting any specific cell fates/lineages represented in gastruloids (Fig. 1j), which could not have been demonstrated in simple 2D co-cultures.

The easy genetic perturbation and mixing of cells with different genotypes into mosaic gastruloids allowed us to demonstrate that knockout of Fas, FasL, and Fadd (key members of the extrinsic apoptosis pathway) does not interrupt cell competition, which together with our Bcl2 rescue strongly suggests that it is not mediated by extrinsic apoptosis. This direct exclusion of extrinsic apoptosis pathways (Fas-FasL-Fadd) through knockout of all its members is novel.

Lastly, we thank the reviewer for highlighting the scalable nature of gastruloids and the high interest in finding the cell-cell communication pathways during competition (as has recently been discovered elegantly for inter-species competition (PMID: 41289993)). We absolutely agree that this is a great opportunity, and that the system as characterized here is ideal for such a screen, which we are currently gearing up to do.

2. While the authors present a thorough analysis of potential mechanisms, most of the experimental results are negative or descriptive in nature. As a result, the key factors driving the competitive interactions between WT and p53KO cells remain unclear. In addition, the specific cell types involved in the competition have not been clearly defined. Interestingly, this work is reminiscent of findings from a recent study (PMID: 33508854), which showed that cell competition between human and mouse pluripotent stem cells occurs specifically in the primed, but not naïve, state. Although the underlying mechanisms are likely different, it would be valuable to explore whether the competition observed in this study is also limited to primed PSCs. For example, co-culturing WT and p53KO EpiSCs under conditions that support the primed state (e.g., with FGF2 and a Wnt antagonist such as IWR1 or XAV939) could help determine whether primed PSCs mediate the observed competition. If competition is observed in this context, EpiSC co-cultures may serve as a more defined system to dissect the molecular mechanisms involved.

The above-mentioned study (PMID: 33508854) as well as another elegant follow-up study by Dr. Jun Wu's group (PMID: 41289993) have indeed demonstrated that inter-species competition occurs in primed pluripotency state but not naïve pluripotency. However, we also agree with the reviewer that the mechanisms governing inter-species competition (e.g. foreign RNA sensing) may likely differ from intra-species fitness sensing competition.

In respect to our study, we thank the reviewer for the emphasis on the pluripotency state of competing cells, which we agree is a highly relevant and key aspect of this manuscript. Our gastruloid time course data does indeed strongly suggest that it is exactly the time between primed pluripotency and gastrulation, during which cells undergo competition. We observe that no competition takes place until 48h of gastruloid formation, when cells still demonstrate peak expression of primed pluripotency/epiblast markers FGF5, Otx2, Pou3f1, Dnmt3b (Fig. 6b). Only after 48h, when these primed pluripotency markers are downregulated and cells transition into gastrulation (peak Brachyury expression at 72h, Fig. 6b; E-to-N-Cadherin switch reminiscent of EMT and mesoderm specification at 96h, Fig. 6n), do we observe the full effects of cell competition. Our new degron-based short-term reductions of p53 protein levels demonstrate that lowering p53 between 48-72h results in strong but incomplete cell competition, compared to when p53 protein levels are kept low until 96h. We therefore believe that cell competition starts at primed pluripotency (48h) but that it peaks at a transition stage into gastrulation (72h). Additionally, and to most directly answer the reviewer's question, we did test and demonstrate that in 2D co-cultures no competition takes place in naïve pluripotency, but that cell competition does take place

between primed pluripotent EpiSC co-cultures (grown in N2B27 + Activin A + FGF)(Extended Data Fig. 1g-m). However, we also observe that cell competition in 2D EpiSC cultures only takes place when cells are grown to over-confluence (Extended Data Fig. 1i), and not if cells are passaged as sub-confluent yet tightly packed mixed EpiSC colonies (Extended Data Fig. 1m). This strongly suggests that our observed intraspecies competition in 2D EpiSCs might thus be caused by mechanical compression and/or nutrient scarcity, both of which we think not to play large roles in 3D gastruloid cell competition. We therefore believe that gastruloids as a model allow us to study physiological 3D cell competition in a manner that we could not achieve with 2D cultures.

3. The authors attempt to probe the role of signaling pathways in cell competition by treating cell aggregates with various agonists and antagonists. However, this section has several major issues that limit the interpretability of the results:

a) These treatments lead to different lineage outcomes and cell compositions. Given that cell competition is highly context- and cell type-dependent, it's unclear whether any meaningful conclusions can be drawn from comparisons across such divergent samples.

We agree with the reviewer that cell competition is context and cell type dependent. It is for this reason that we use signaling perturbations to alter the context, and ask whether the cell populations with the same relative fitness difference compete more or less in these new contexts. We do not believe that these developmental signals directly play a role in communicating fitness, but instead believe that cells in different signaling environments during development might demonstrate heightened or lowered sensitivity to cell competition stress. We have now clarified this in the text.

b) Both WT and p53KO cells are simultaneously exposed to these pathway modulators, making it impossible to determine whether the observed effects are due to changes in the "winner," the "loser," or both. Without cell-type-specific perturbation, mechanistic insights remain speculative.

We indeed treated whole mosaic gastruloids with these signaling perturbations, to mimic different parts of the developing epiblast, where neighboring cells would also experience the same signaling environments. While cell type specific perturbation of developmental signaling could result in interesting observations, this was not the scope or question that the experiments aimed to address.

c) All treatment comparisons are made relative to DMSO controls, not CHIR. This is problematic because gastruloid formation, unlike simple embryoid body differentiation, requires CHIR as a baseline condition. Any chemical perturbations should be performed in the presence of CHIR and compared to a CHIR-only control. That said, layering additional agonists or antagonists onto CHIR may introduce confounding effects similar to those raised in point (a). Given these limitations, I recommend removing this section unless the authors can design more targeted and interpretable experiments.

We thank the reviewer for raising the issue of comparing all conditions to DMSO controls. We have now restructured this section and draw all comparisons between agonists and their respective antagonists of the same pathway (Fig. 5). We do believe that the treatment strategy is appropriate given the goal of this experiment and the biology of gastruloids, as we explain in more detail below, and believe therefore that deleting this section entirely would not improve the manuscript.

Both the first and second manuscript originally describing gastruloids demonstrated that gastruloids can elongate without CHIR treatment albeit to a lesser extent (PMID: 28951435; PMID: 25371360). Additionally, our group has recently conducted an in-depth study specifically on the role of CHIR and Wnt signaling during gastruloid formation, and demonstrated that CHIR is not required for gastruloid formation per se. Instead, CHIR acts to coordinate and accelerate developmental progression and entry into gastrulation (<https://www.biorxiv.org/content/10.1101/2025.01.11.632562v2>). Accordingly, gastruloids can still elongate without CHIR treatment due to intrinsic activation of Wnt signaling, however, this occurs with a delay, as developmental progression is slowed, and in a less coordinated manner, possibly due to the heterogeneity of intrinsically driven Wnt activation compared to the externally synchronizing CHIR pulse. This is also the reason why we believe that the DMSO vs CHIR treated gastruloids are most different in their timing and developmental progression speed (Fig. 6c, Extended Data Fig. 6a-b). Lastly, we agree, that layering signaling perturbations on top of CHIR treatment would result in confounding effects difficult to interpret. Our group has observed very reproducibly that CHIR treatment overpowers other developmental signaling stimuli. We therefore believe that comparing the agonists vs antagonists of BMP, Nodal, and ERK signaling in the absence of CHIR treatment is the best way to address the posed question, and have restructured and re-analyzed this section now thanks to the reviewer.

- The authors claim that glucose level manipulation did not impact cell competition as seen in figure 4H, but strikingly they seem to leave out that on almost complete glucose deprivation to 0.5mM that cell competition is almost completely abolished. This is a fascinating result that is not explored. Admittedly when comparing WT+WT to WT+p53KO at different levels it would appear that there little effect of glucose on the cell competition relative to WT+WT. However I believe that this is the wrong scientific comparison being made. Instead of WT+WT vs WT+p53KO in 21mM, what should be compared is WT+p53KO 12mM vs. WT+p53KO 21mM. If you compare the WT+p53KO condition across glucose levels, it seems to me that clearly you have an effect where lowering the amount of glucose drives lower cell competition, even so much as to almost completely abolish it in the 0.5mM condition.

We thank the reviewer for the interest regarding our glucose limitation studies. Going into these glucose limitations, our goal was to test whether cell competition in 3D gastruloids was caused by competition for nutrients, which would predict exacerbated competition at low glucose. We too were excited when seeing that the opposite was true. Our analysis viewpoint asks the question whether the same number of WT-mCherry cells introduced to a gastruloid with supercompetitor cells or with other WT cells, would grow into the same or a lower number of mCherry-survivors. This is what our comparison of blue to green bar graphs throughout the manuscript measures and what we call the “effect size of competition” (e.g. Fig. 4n the glucose normalized graph). We have now included the comparison of WT-mCherry vs p53KO-emiRFP cells across glucose levels that the reviewer suggested (Extended Data Fig. 4o), which supports that lower glucose levels diminish the difference in cell numbers between these two populations. However, when we look at the absolute counts of cells per gastruloid for all populations (Fig. 4m), we notice that the p53KO cells are the only population following a clear trend in growth behavior relative to glucose levels, while the other populations are unaffected in the range of 2.5 mM to 21 mM. This suggests that p53KO cells are likely to be more affected by glucose limitation. At the 0.5 mM glucose levels all populations are impaired in their growth behavior, which is why we found this lowest glucose data point less trustworthy to interpret. In any event, we have conducted and added new experiments to test whether inhibition of glycolysis or oxidative phosphorylation around the time of primed pluripotency would impair cell competition (Fig. 4o,p) and observe that both perturbations similarly affected growth of p53KO cells and the cell competition effect size, but only to a lesser degree the growth of all other populations. This suggested that the lower cell competition effect size in extreme low glucose levels may not have been specifically due to glycolytic vulnerabilities, but possibly an overall reliance on higher metabolic rates in the p53KO cells. Lastly, sparked by the reviewer’s interest, we were reminded of recent studies linking glycolysis at the onset of gastrulation to the expression of Brachyury in gastruloids (PMID: 40245870; PMID: 38636516). New data now included in our manuscript demonstrates that full loss of Brachyury desensitizes cells to competition. We thus asked whether our 0.5mM glucose medium impaired Brachyury expression but unfortunately found that this was not the case (Reviewer Figure 1).

Taken together, following the reviewer’s suggestion, we have investigated the glucose limitation effect on cell competition further, and now believe that the lowered competition effect size is surprisingly due to the higher sensitivity of the p53KO cells to metabolic perturbation compared to other populations. One additional very valuable insight came from the comparison the reviewer suggested, namely, that the growth behavior of the supercompetitors does not influence the cell competition outcome (Extended Data Fig. 4o), and that as long as there are at least as many supercompetitors as loser cells within a gastruloid, their faster or slower growth does not add more competition pressure (Compare WT-mCh and p53KO bars between 2.5 mM and 21 mM glucose). This is another novel aspect previously unknown.

Reviewer Figure 1. Quantified Brachyury immunofluorescence intensity of stained gastruloids treated with low glucose of 2-DG. n=11-80 gastruloids from 3 independent experiments per condition were fixed,

stained and imaged as confocal stacks using a spinning disk microscope. Sum-projections were generated before quantification of fluorescent intensities.

Minor Comments:

1. Lines 59-65, it is relevant to cite the study by Zheng et al (PMID: 33508854) and briefly mention interspecies primed PSC competition mediated by innate immune response in loser cells.

Thank you for this suggestion, we agree that this is an important and relevant part of the literature and have now included it.

2. Lines 66-68, it is relevant to cite the study by Dejosez et al (PMID: 24030493).

Thank you for the suggestion, we have now included this study in our manuscript.

3. Lines 99-114, is the cell competition observed in confluent naïve cultures contact-dependent. To address this, the authors should consider using transwell co-cultures.

We thank the reviewer for this suggestion and have now conducted transwell assays with naïve and primed pluripotent cells (Extended Data Fig. 1o). We do not observe any growth impairments on WT-mCherry cells co-cultured with larger numbers of WT-emiRFP, or p53KO-emiRFP cells. This supports our hypothesis that 2D competition is likely compression based. To follow up with the reviewer's suggestion even further, we have also performed co-cultures of 3D gastruloids in adjacent hydrogel microcavities within the same medium continuum (Fig. 4a-b). In these co-cultures, WT-mCherry 3D gastruloids are surrounded by six WT-emiRFP or p53KO-emiRFP gastruloids within the same "Gri3D" 96-well. Final cell numbers of WT-mCherry cells in the central gastruloids were assessed at 120h. We do not observe any cell competition effects if competing cells are not within the same gastruloid. Furthermore, to account for the possibility that secreted "winner" signals might be context specific and only sent upon encountering loser cells, we surrounded WT-mCherry gastruloids by mosaic WT+p53KO gastruloids. However, this still elicited no competition pressure on the central gastruloids. Together, these data indicate that cell competition signals in this context are either contact dependent, or very short range. We thank the reviewer for inspiring this valuable addition.

4. The scale bar is missing in Figure 1g.

Thank you, we have now ensured that all panels contain scale bars.

5. It is puzzling that WT-mCherry cells show a longer doubling time compared to WT-emiRFP cells (Figure 1b), yet they form larger gastruloids (Figure 1i). This apparent contradiction raises questions about the factors driving gastruloid growth in this system. The authors should clarify how a slower-proliferating cell line produces larger structures and provide possible explanations—such as differences in cell survival, differentiation dynamics, or organization—that might account for this outcome.

We agree that at first glance it seems counterintuitive that the clone with shorter doubling times generates smaller gastruloids, as the reviewer has keenly recognized. We believe that this is primarily explained by the fact that proliferation rates change during differentiation (slowing down with developmental progression) and that proliferation rates in one developmental stage may not be a good indicator for proliferation in other stages. In naïve pluripotency, we do observe that the WT-emiRFP cells have a shorter doubling time than the WT-mCherry cells (Fig. 1b) and outgrow co-cultured naïve WT-mCherry cells (Extended Data Fig. 1e). However, the proliferation rates of all clones slow down through developmental time as cells differentiate (Fig. 3b-d). We had previously demonstrated this by phospho-Histone3 quantification and have now added full EdU-based cell cycle analysis of all clones through all developmental timepoints (Fig. 3c-d; Extended Data Fig. 3a-d). In this analysis we observe that the WT-emiRFP clone has the same percentage of cells in G1/0 as the WT-mCherry clone during naïve pluripotency (depicted as "0 h"), but a slightly elevated percentage of cells in G1/0 in all timepoints from 48h onward (Extended Data Fig. 3b). This might indicate clonal differences where WT-emiRFP cells downregulate cell cycle progression slightly more during differentiation than WT-mCherry clones. Given that the WT-emiRFP clone is the direct parental clone of the p53KO-emiRFP supercompetitors, we still believe that this clonal growth difference during differentiation only makes the competition phenotype we observe even more

impressive, as the previously faster growing WT-mCherry clone turns into the loser clone upon p53 loss in the emiRFP cells.

6. In Figure 2h, the WT-WT co-culture data show that WT-mCherry and WT-emiRFP cells maintain a stable ratio up to 72 hours. However, after that point, WT-mCherry cells begin to dominate, indicating a potential competitive advantage even between two WT lines. This raises concerns about underlying differences between these cell lines that the authors do not address. The fact that one WT line consistently outcompetes the other suggests there may be alternative mechanisms at play that remain unexplored. The authors should investigate this further, for instance, by staining for markers of cell proliferation and apoptosis to determine whether actual competition is occurring.

We thank the reviewer for this observation, which we believe connects to the point raised above. As explained, even the pure WT-emiRFP clone gives rise to slightly smaller gastruloids than the pure WT-mCherry clone, likely due to clonal differences. Clonal differences during early development have recently been demonstrated to play an important role (<https://doi.org/10.1101/2025.05.23.655664>), which is why we put such emphasis that the p53KO-emiRFP clone is a direct derivation of the WT-emiRFP clone with only 3 passages difference.

Given that cell competition is believed to be a delicate quality control to detect even small clonal differences beyond the otherwise cell-autonomous error detection mechanisms, we believe it to be likely that cell competition might to some degree also occur between the WT clones. Even in WT+WT gastruloids, cells seem to take up fluorescent debris from dying neighbors (Fig. 2d), and as the reviewer observed, the WT-mCherry cells of interest seem to dominate WT+WT gastruloids slightly after 72h (Fig. 2h).

To more directly answer the question whether these growth differences in mosaic WT+WT gastruloids are caused by inherent growth differences of the clones as observed in Extended Data Fig. 2c, or by cell competition between the WT clones, we have now conducted more detailed analysis of stress and apoptosis in WT+WT mosaic gastruloids at 72h of gastruloid formation. While WT-emiRFP cells in mosaic WT+WT gastruloids demonstrate slightly elevated percentages of late apoptotic cells (AnnexinV-DAPI double-positives by flow cytometry; Extended Data Fig. 3e-f), WT-emiRFP clones do not have higher proportions of cleaved caspase 3 positive cells than their WT-mCherry neighbors (Extended Data Fig. 3g), and do not display elevated p53 protein levels compared to their WT-mCherry neighbors (Extended Data Fig. 3h-i). We thus believe that the dominance of WT-mCherry cells in WT+WT mosaic gastruloids is mostly due to the cell-autonomous growth differences during differentiation discussed in the remark above. However, we do believe that a slight clonal difference in this direction, with the WT-mCherry cells being “dominant”, is advantageous for our study. Given that we demonstrate the WT-mCherry clone to be an inherently fit clone among WT clones, we have higher confidence in the data demonstrating that same clone turning into an outcompeted loser upon loss of p53 in the neighboring emiRFP cells.

Lastly, to overcome any notion of clonal differences, we have now generated a p53-degron-emiRFP cell line. With this, we generate mosaics between the same two clones (WT-mCherry + p53degron-emiRFP) in presence or absence of degron induction (Fig. 8) and observe that cell competition is truly based on transient p53 levels of competing clones. Even short term, reversible, and incomplete reduction of p53 levels in the emiRFP neighbor clone is sufficient to cause outcompetition of otherwise fit WT-mCherry cells.

7. Another issue is the authors' claim that “the same WT-mCherry population can co-develop with other WT cells by equally contributing to the final gastruloid.” This is directly contradicted by their own data: both immunofluorescence (Figure 2A) and flow cytometry (Figure 2H) show that WT-mCherry cells become the dominant population in 3D gastruloids. Furthermore, this dominance pattern conflicts with the results shown in Extended Data Figure 1, where WT-emiRFP appears to gain the upper hand during co-culture (e.g., ED Fig. 1e). This discrepancy raises an important question: if WT-mCherry ESCs proliferate more slowly in naive culture, how do they come to dominate in 3D gastruloids? Do they proliferate or differentiate more efficiently during gastruloid formation? The authors should clarify this point and reconcile the inconsistent results.

We agree with the reviewer that the phrasing of “equal” contribution might have been poorly chosen and might even lead to underappreciation of the drastic behavior switch that the WT-mCherry cells display in presence of WT vs p53KO neighbors. We have changed our phrasing in the main text accordingly to: “Mixing 150+150 WT-mCherry and WT-emiRFP cells (WT+WT) during aggregation resulted in mosaic gastruloids composed of both clones with slight overrepresentation of WT-mCherry cells after 120h”.

We believe that the explanation and the new experimental data discussed in the two remarks above (5. and 6.) overlap with this remark (7.) and provide more detailed answers.

8. The authors repeatedly refer to Figure 2K, but no such figure exists in the manuscript. It's likely they intended to reference Figure 2I.

We thank the reviewer for finding this labeling mistake. The figure previously referred to is now Figure 2n, and the main text has been adjusted accordingly.

9. In Figure 3a, it appears the authors are showing a chimeric gastruloid composed of WT-mCherry and WT-emiRFP cells. If that is the case, the individual mCherry and emiRFP fluorescence channels should be shown alongside the pH3 staining. This is important to assess how proliferation (as marked by pH3) is distributed between the two cell populations.

We have now replaced the previous 120h WT gastruloid image with newly generated and more detailed microscopy panels displaying mosaic WT+WT and WT+p53KO gastruloids along with their fluorescent markers. The quantification of pH3 positive cells among each population at each timepoint has been conducted by flow cytometry and is not affected by this replacement.

10. Figures 3f–3g show that Bcl2 overexpression markedly increases the number of WT-emiRFP cells—particularly at 144h—in chimeric gastruloids formed with WT-mCherry cells. This suggests that, under normal conditions, WT-mCherry cells outcompete WT-emiRFP cells even in the absence of p53 knockout. Bcl2 overexpression in WT-emiRFP cells does indeed result in higher numbers of cells at 120h and 144h. While this could suggest that there is competition between the WT clones, it is very likely that blocking apoptosis also prevents competition-unrelated cell death in these cells. Even pure clonal gastruloids display some degree of apoptosis and shedding of cells, and inhibition thereof would increase cell numbers particularly toward the end of the life cycle of gastruloids. To assess whether there are signs of cell competition between the two WT clones, we have now conducted additional experiments to investigate proliferation and apoptosis rates between the clones and detect a very slightly elevated percentage of late apoptotic cells at 72h in the WT-emiRFP population (not significant, see **Reviewer Figure 2**), but no increased p53 levels as sign of competition stress response. Bcl2 overexpression decreases the cleaved-caspase 3 positive percentage to almost zero even in WT+WT-Bcl2 co-cultures (Fig. 3s). We discuss our observations on WT+WT competition in more detail in the points above.

Reviewer Figure 2. Analysis of early apoptotic, late apoptotic, and necrotic percentages in 72h gastruloids as assessed by AnnexinV+DAPI stained flow cytometry. Related to Manuscript Extended Data Figure 3e-f. Apoptosis in WT-emiRFP cells is only slightly but not significantly elevated compared to co-cultured WT-mCherry cells.

11. ED Fig. 5a-5b, the authors should provide quantification of the percentages of Active Caspase 3 positive cells in different cell line in the chimeric gastruloids.

We thank the reviewer for this helpful suggestion. To get higher representation of large numbers of gastruloids, we have now analyzed the percentage of active/cleaved caspase 3 positive cells by flow cytometry in pooled independent replicate batches of gastruloids. In agreement with our overall observations of cell competition, we observe a significant increase in the number of active/cleaved-caspase 3 positive WT-mCherry cells in mosaics with p53KO cells, when compared to mosaics with other

WT cells (Fig. 3f, paired two-tailed t-test; and Extended Data Fig. 3g, one-way ANOVA with Tukey's posthoc testing adjusted for multiple comparisons).

12. In Figures 4g–4h, the authors state that glucose level manipulation does not impact cell competition. However, in the 0.5 mM glucose condition shown in Figure 4h, competition appears to be nearly abolished—an intriguing observation that is not further explored. This result suggests that low-glucose conditions may alter the competitive dynamics, potentially by affecting proliferation or survival of p53KO cells. The authors should investigate this more closely. For example, is reduced competition due to impaired proliferation, increased apoptosis, or altered metabolism in p53KO cells under glucose restriction?

We thank the reviewer again for highlighting this aspect of the study. As discussed in detail in our reply to the overlapping Major Comment above, we have followed up with this observation by investigating the effects of glycolysis and OxPhos inhibition, as well as an analysis of whether the low glucose effects were related to impaired brachyury expression, but have ultimately concluded that the lack of cell competition is most likely due to a heightened sensitivity to metabolic perturbation or nutrient limitation in the p53KO cells. This aligns with the observed growth behavior in which the p53KO cells show a clear correlation of overall growth to glucose levels across all tested concentrations, while the other clones are unaffected by glucose limitation until the lowest concentration is reached. We believe that the lack of cell competition is likely due to overall impairment of growth in all cells at the lowest glucose level, in combination with a heightened response to glucose limitation in the p53KO cells.

13. Lines 305-306, it is relevant to cite the study by Price et al. (PMID: 34407428)

We absolutely agree! We have now added the citation to the statement. Inspired by the reviewer's comment and the mentioned study, we have additionally conducted staining and image segmentation analysis for nuclear-to-cytosolic YAP ratios in competition gastruloids. We observe no differences in YAP localization between WT-mCherry cells in WT+WT gastruloids vs WT+p53KO gastruloids at 72h (Fig. 4j; Extended Data Fig. 4j-k). This agrees with our observation that mechanical inhibitors do not prevent cell competition, resulting in our statement that cell competition in 3D gastruloids is not likely governed by mechanical cues and thereby different from 2D cell competition.

14. Lines 352-354, Figure 6c and ED Figure 8b, from the presented data, it is not clear to me “non-autonomously delaying neighboring WT-mCherry cells in mosaic gastruloids” is correct.

We have now deleted this part of the sentence.

15. Please clarify the statistical methods used in the Methods section or figure legends. For example, if a t-test or one-way analysis of variance (ANOVA) was used, please specify.

We apologize for not having clarified this sufficiently before and have now listed the specific statistical methods used. Unless otherwise stated, all statistical comparisons of only 2 samples were conducted by two-sided t-test, and all comparisons containing more than 2 samples we conducted by one-way ANOVA with Tukey's post hoc multiple comparisons test, or two-way ANOVA with Tukey's post hoc multiple comparisons test, depending on the dimensionality of the dataset. We have now also added the previously missing section on Statistical Analysis in the manuscript.

Reviewer #2 (Remarks to the Author):

Gastruloids generated from mouse embryonic stem cells provide a useful system with which to investigate mechanisms of gastrulation. In this manuscript, Frenster and co-workers generated modified clones of E14Tg2A ESCs, initially to include fluorescent labels and knockout of P53 to use as tools to investigate aspects of transitions from naïve to primed pluripotency during pre- and post-gastrulation stages. As preliminary validation of the tools generated, the authors performed cell competition assays in 2D and confirmed previous studies showing that, in the presence of P53KO cells, WT clones were dramatically out-competed within a specific time scale, corresponding to pre- and peri-gastrulation in mouse embryos (~E5.5-E7.5). Using 3D gastruloids, they revealed that a surprisingly small number of P53KO cells were required to overcome more than half of the WT cells within a short time window, within 120 hours of culture.

The authors note that gastruloids produced from P53KO or WT clones separately exhibit different morphology; those from P53KO cells tended to be bigger and more irregular than WTs, but the effect of P53KO cells on WTs in mosaic gastruloids from 72 hours after aggregation was much more dramatic. This can be blocked via inhibition of apoptosis in 'loser' cells via Bcl2 overexpression, as has been shown previously in 2D culture. This system allowed the authors to investigate the window of opportunity for rescue through Bcl2. Inducible overexpression of Bcl2 in 'loser' cells from 0-48 hours of aggregation or 48-72 hours appears to be protective from cell competition. Given the long half-life of BCL2 protein, this probably suggests a critical window between 48-96 hours during which cell competition induces apoptosis. Accordingly in WT-KO mosaics, a peak of Noxa expression is detected 48 hours after aggregation, followed by a peak of p21 at 72 hours and Bcl2 at 96 hours. The pulse of Chiron, typically administered from 48-72 hours, is somewhat protective, with increased levels of competition when not provided, probably because Chiron increases the developmental pace of cells in this context, driving them out of the developmental window in which cell competition occurs.

The authors generate 'heterochronic' gastruloids by mixing and reaggregating cells from time mis-matched gastruloids. They find that 96 hours after aggregation P53KO cells lose the capacity to outcompete 48 hour wild-type cells, while late (72-96 hours) WT cells are no longer outcompeted by 48 hour P53KO cells, whereas early cells (24 hours) are both able to influence and be influenced by their 48 hour neighbors. As expected, cell competition was not observed if the cells were in the EpiSC state; the resulting structures comprised mostly spinal cord and brain tissue, confirming their temporal position outside the developmental window of cell competition. Interestingly, P53KO cells tended to differentiate preferentially to mesoderm rather than endoderm. Transcriptomic analysis revealed bias at 120h for P53KO cells to form paraxial mesoderm and caudal epiblast rather than blood lineage or PGC-like cells. Both WT and P53KO clones exhibited phagocytosis, implying that this mechanism was unlikely to be responsible for driving cell competition. From 48 hours, WT cells appeared to stagnate and undergo apoptosis, as has been reported for 'loser' cells in chimeric mouse embryos at the onset of gastrulation. P53KO cells exhibit impaired growth in response to blockade of apoptosis in WT cells or when Bcl2 is expressed. Apoptotic cells tended to cluster in the central region of the gastruloid. Since metabolic processes have been shown to play a role in cell competition, the effect of glucose concentration was tested, but found to be of little significance, as was ROS and cytoskeletal tension. Absence of BMP signalling or Chiron appeared to delay gastruloid cell competition. Also, P53KO cells were delayed in exit from naïve pluripotency, implicating some disruption of the networks in P53KO cells. By 72h WT cells were found to lose responsiveness to cell competition. However, the period of permissive cell competition can be varied by modulating Wnt signalling. This work is largely well planned and executed, resulting in novel information likely to have an important impact on the developmental and stem cell biology fields, particularly in the context of modelling early human postimplantation development. However, there are several issues that the authors should address.

Presentation of methods:

1. The code relating to the RNA-sequencing analysis is largely clear and well documented, using standard techniques. However, it is unclear where the reference embryo data used for the spatial analysis (anterior-posterior identity) comes from. It appears that the data is annotated with spatial information that does not seem to be present in the data referenced.

We first want to thank the reviewer for their kind words and positive assessment of this study. Regarding the spatial analysis, data from a published embryo atlas (PMID: 37982461) containing dissected anterior and posterior portions of E8.5 embryos were used as anterior and posterior references, and were mixed at different proportions to generate a pseudobulked anterior to posterior signature gradient. Gastruloid samples were then embedded into this gradient via PCA to assess spatial correspondence. We have now clarified this in the Methods section.

2. The code, and methods, relating to most of the image analysis is insufficient to reproduce this work. The authors use a tool (QLIVECELL) which does not appear to have been validated in this work or in other publications. The code and documentation provided are not sufficient to run the validation tests, and the methods section does not detail which functions were called or what parameters were used. It is unclear which steps were performed using this tool and which were performed as downstream analysis.

Thank you for pointing out this oversight. We have now added detailed documentation on all image analysis pipelines to our linked github repository, as well as the code developer's repository, including descriptions, all code, as well as an example dataset for testing.

https://github.com/stembryo-lab/cell_competition_gastruloids

<https://github.com/dsb-lab/qlivecell>

3. Throughout most of this manuscript, statistical analyses are reported without indicating what statistical test has been performed.

We now specifically write which statistical test has been used in each figure legend.

4. There does not appear to be any explanation of the generation of the Bcl2, Fas, FasL or Fadd cell lines in the methods.

We thank the reviewer for pointing this out. We have now detailed the generation of these knockout clones by CRISPR Cas9 mediated excision of exonic regions, as well as the sequencing confirmation of excision and resulting frame shifts in the methods section.

Points for clarification:

1. In Figure 5c, it is found that Chiron stimulation from 48-72 hours post-aggregation strongly weakens the effect of cell competition compared to DMSO, while BMP4 treatment provides a milder protective effect. In Figure 4k, the authors see a similarly mild effect of cell competition in Chiron-only treated mosaic gastruloids. This is contrasted by the very strong and consistent effect seen in Figures 1h, 2i, 3n, 4c, and 4p. This is unexpected since each experiment is performed with technical and biological replicates which appear to show relatively low variability. Are the authors able to identify any reason why the cell competition effect is so much weaker in just these two experiments?

We thank the reviewer for raising awareness to this very important aspect. The mentioned experiments with less drastic cell competition effect size were conducted on the background of homemade N2B27 medium, whereas all remaining experiments of this study were conducted in commercially available N2B27 (Takara, NDiff227). While both of these media generate gastruloids faithfully and clearly display cell competition, gastruloids in homemade medium experience an overall shifted (lower) effect size of cell competition. Our group has in fact very recently published a study comparing these two N2B27 media in detail during gastruloid formation (PMID: 41287934) and observed that homemade N2B27 gastruloids in general contain lower percentages of apoptotic cells, and possibly lowered sensitivity to stress. Due to the proprietary nature of commercial N2B27, we are unfortunately unable to deduce where these differences originate from. In this current study, only 3 experiments were conducted with homemade N2B27, while all other experiments were conducted in the same commercial N2B27. The ROS inhibitor and signaling perturbation studies were conducted with homemade N2B27 during a time when the commercial N2B27 was unavailable for a longer time, and the glucose limitation studies have been conducted with an altered formulation of the homemade N2B27 to generate the necessary glucose-free and pyruvate-poor background. The ROS inhibition experiments have now been removed from the manuscript and the glucose limitation and signaling studies now clearly mention the difference in medium background. Throughout this manuscript, comparisons have only been made between experiments conducted side by side in the exact same medium conditions, and no comparisons across different media formulations were made. We thank the reviewer for noticing this difference and we are now mentioning this more clearly in the manuscript as exemplified below:

“To test the influence of glucose availability on cell competition, we generated gastruloids in a pyruvate-poor N2B27 medium with varying glucose concentrations (Fig. 4m-n; Extended Data Fig. 4n-o). While the effect size of competition in this homemade medium was less pronounced than in commercial medium, lowering the amount of glucose eightfold from 20 mM to 2.5 mM did not alter cell competition”

Minor points and typos:

2. In Fig.1g, why are the gastruloids green only? Should they not be (equal) mixtures of pink and green?

Figure 1g displays 120h gastruloids in which a fixed number of WT-mCherry cells (150) were mixed with a FACS-sorted titration of emiRFP cell numbers. In the composite image of the WT+WT condition that looks green only, 150 cells mCherry were mixed with e.g. 1, or 2, or 4 etc emiRFP cells. Even at the maximum number of WT-emiRFP cells, they still are mixed with 10-fold more WT-mCherry cells in the starting population. Because of the nature of this titration experiment, we are not observing an equal mix of

mCherry and emiRFP WT cells at 120h and are thus so impressed by the strong competition phenotype in the WT+KO condition. We are now displaying all single channel images and a purposely overexposed version of the emiRFP channel to visualize the few titrated WT-emiRFP cells that are otherwise not becoming apparent (Supplementary Figure 2a-b).

3. The legend for Fig. 2 states that cell competition is temporally, but not spatially restricted, whereas the main text says that cell competition occurs predominantly in the central region of the gastruloid

We agree with the reviewer that the title of Figure 2 was not well chosen. We originally meant to express with the figure title that there is no spatial bias or preference within gastruloids where each clone is localized, indicating that cell competition does not for example eradicate loser cells from the core or edge of gastruloids drastically. However, this statement was based on the 3D image segmentation analysis and location of clones (Figure 2a-e) without taking apoptosis into account. The 3D analysis of cleaved caspase positive apoptotic cells displayed in Extended Data Figure 5 detects that the density of early apoptotic cells is higher toward the core of the gastruloid than toward the edge. These two statements may seem contradictory, but given the incredibly motile nature of cells in gastruloids, it is possible that the slight bias of apoptosis toward the center is insufficient to shift the overall population distributions measurably. In agreement with the reviewer, we have now changed the title of Figure 2 to "*Cell competition results in stagnation of loser cell numbers after 48h*".

4. Please show the variability in Figure 2e

We do now display the variability in Figure 2e as error ribbons surrounding the mean values.

5. Figure 2j is referred to as Figure 2k in the text

We thank the reviewer for noticing this mislabeling and apologize for the confusion this might have caused. This figure has now become Figure 2n in the revised manuscript

6. The treatment times in Figure 5c differ between the figure legend/methods and the main body text

Again, we thank the reviewer for finding this mistake. We have now adjusted the text to the correct treatment timings. All treatments were conducted between 48-120h, except Activin A and ChIR which were treated from 48-72h.

7. Could the authors please elaborate on the blebby appearance of P53KO gastruloids in Fig. 5d?

The "blebby" gastruloids in Figure 5d (now Figure 5f) are mosaic gastruloids containing p53KO cells in the absence of treatment with the Wnt agonist ChIR. We believe that this appearance is an indication that p53KO gastruloids tend towards neuroectodermal lineages in absence of ChIR. Previous literature has demonstrated that p53 is upstream of Wnt expression and mesendodermal differentiation (PMID: 27889317). We believe that in the absence of p53 and absence of ChIR treatment, this tendency toward neuroectoderm instead of mesendoderm becomes apparent and results in the folded surface appearance mentioned by the reviewer. To follow-up with this lead, we have purposely generated WT+KO aggregates that we directed toward neuroectodermal fates (not shown), but this resulted in less competition than when regular gastruloids grew in the absence of ChIR, which lead to our conclusion that this neuroectodermal shift is not what caused the increased competition we observed in absence of ChIR.

8. Line 438: 'Blc2' instead of 'Bcl2'

Thank you, we corrected this typo.

9. In Figure 6k it currently appears as if there are 0 mCherry+ cells when 48 hour WT cells are mixed with 24 hour P53KO cells or with 72 hour WT cells. These conditions are apparently also tested in Extended Data Figure 8. Please could the authors clarify this confusion?

We thank the reviewer for raising our attention to the confusion this panel caused. Previous Figure 6k had depicted a combination of WT-mCherry counts from panels 6h and 6j to allow for a more direct comparison. Due to the nature of the experiments in Figures 6h and 6j, the empty slots did not contain zero-counts but were instead conditions not included in the experiment. To avoid this confusion for future readers, we are now omitting this comparison panel, as it does not add new data to the manuscript. There is indeed an overlap in the conditions tested in Figures 6h-j and what is now Extended Data Figure 6c-h. The reason for this is that the experiment in Figure 6h-j was specifically focused on the creation of a 24h pluripotency shift between two WT populations and the abolishment of the observed pluripotency shift of

p53KO cells. A full heterochronic shift matrix was later created as depicted in Figures 6k-m and Extended Data Figure 6c-h, however, these were independent experiments, and we decided to depict both to keep the focus on the narrower hypothesis testing in Figure 6h-j.

Reviewer #3 (Remarks to the Author):

In this manuscript, Frenster et al. develop a mouse 3D gastruloid model for studying cell competition during gastrulation and investigate the role of the tumor suppressor p53 in this process. Gastruloids were generated from ES cells grown in specific culture conditions for 5 days to make elongated and polarized structures representing embryos at E8.5. The authors first examine the effects of co-culture of WT-mCherry (labeled green) ES cells with either control WT-emiRFP ES cells or isogenic p53KO-emiRFP ES cells. They find that by coculturing the WT-mCherry ES cells with the p53KO-emiRFP ES cells, the p53KO cells are supercompetitors, with as few as 2 p53KO cells impairing growth/survival of the remaining wild-type cells in the gastruloid. They show that this phenomenon is not due to spatial positioning of cells, but due to the cells' existence in a specific time window of development centered around 48h-96h post aggregation, which corresponds to E5.5-E7.5. The authors show that expression of Bcl2 in the WT-mCherry cells blocks the ability of p53KO cells to supercompete, suggesting that p53 KO cells promote apoptosis in neighboring WT cells. They show further that intrinsic, not extrinsic, apoptosis in the WT-mCherry cells is the cause of cell competition, and they rule out several causes of this phenomenon, including nutrient availability, ROS, and mechanical compression. Finally, the authors show that increased Wnt signaling results in decreased cell competition.

Overall, this study provides important insight into the process of gastrulation and the factors that influence cell fitness during gastrulation. This work provides important new perspectives and would be of great interest to the readership of Nature Cell Biology. However, before publication can be recommended, a number of points and critical controls must be addressed in a revised manuscript to improve the story. Importantly, there are a lot of places where clarity could be enhanced in the figures and text. We thank the reviewer for the kind words about our study and for the suggested improvements.

Specific points:

1. Why were two RFPs used (mCherry and emiRFP) instead of two fluorescent proteins with more different excitation spectra?

The choice of fluorescent proteins for the H2B-tags was made to avoid overlap with the green background fluorescence of cell culture medium, which allowed us to image live cells and gastruloids in cell culture medium with high signal to noise ratio. The mCherry (red) and emiRFP670 (far red) fluorophores are comparable to popular fluorophores Alexa568 and Alexa647 respectively and have distinct non-overlapping spectra. Neither by flow cytometry, nor by microscopy did we observe any bleed-through between channels, as can be seen by the negative and single-color control flow cytometry plots in Supplementary Figure 5a.

2. Were the WT-emiRFP cells subjected to the same process as the p53KO-emiRFP cells? That is an important control for the experiment given the selection process that is needed to make the p53KO cells. Even better would be a rescue experiment where p53 is restored in the p53KO-emiRFP cells. This would rule out any secondary mutations that might influence the behavior of these p53KO-emiRFP cells. We agree with the reviewer that it is very important to ensure that our p53KO-emiRFP cells are identical with the WT-emiRFP cells except for the loss of p53, in order to draw valid comparisons. For this reason, the p53KO clone is a direct derivative of the WT-emiRFP clone, with only 3 passages having occurred between freezing of the two cell stocks. The only process that the p53KO cells underwent that the WT-emiRFP cells did not, was one passage under Nutlin-3a selection, which is an MDM2 inhibitor that induces apoptosis in p53-intact cells and therefore could not be survived by WT cells. Additionally, because long-term loss of p53 can lead to karyotypical abnormalities which would influence our results, we have now also conducted and included karyotyping with G-banding for all clones and validated that none of them are karyotypically abnormal.

Inspired by the reviewer's comment, and to fully ensure that the cell competition effects we see are truly due to the loss of p53 and not secondary mutations or long-term downstream effects of p53 loss, we have now generated and included a p53-degron system (Figure 8). This is based on an optimized version of the plant OsTIR system (OsTIR1(F74G) – miniAID2)(PMID: 33177522; PMID: 32941625) and allows inducible degradation of p53 protein within half an hour, with full reversibility and quick restoration of p53 levels. The induction of this degron system is controlled by 5-Ph-IAA, a derivative of the plant hormone auxin, which we validated has no measurable effects on cell competition on its own even at 10x its effective dose (Figure 8c-d). This has allowed us to compare co-cultures of the same two clones (WT-mCherry + OsTIR-p53-emiRFP) in the presence or absence of 5-Ph-IAA. We observed that induction of the p53 degradation resulted in strong competition pressure on neighboring WT-mCherry cells compared to the same clone in absence of p53 degradation (Figure 8c-d). We were positively surprised to see that partial reduction of p53 protein levels (Figure 8a-b) was sufficient to induce strong cell competition, indicating that supercompetition is truly caused by relative differences in p53 protein levels, and does not require a full gene loss. We believe this to be the first ever experimental demonstration thereof.

Additionally, the use of this p53-degron system enabled temporal control over p53 protein degradation, which we use to strengthen the central aspect of this manuscript to demonstrate that relative p53 protein level differences specifically during 48-96h determine cell competition (Figure 8g-k). In these degron pulse induction experiments, we also demonstrate that p53 protein levels are reestablished after the pulse window (Figure 8h-i). Following the reviewer's suggestion of rescuing the p53 levels, we demonstrate that when ending the degron pulse at 72h, we observe significantly less cell competition than when continuing the degron induction until 96h, thereby demonstrating that restoration of p53 protein levels in supercompetitors reverses the supercompetition effect (Figure 8j-k). To our knowledge, this is the most granular experimental dissection of the effect of dynamic p53 protein level changes and the cell non-autonomous effects on WT neighbors that have been conducted to date. We thank the reviewer for guiding us in this direction, and we believe that this makes our manuscript more impactful.

3. In discussing the p63/p73/p53 triple knockout phenotypes, it is important to also consider and cite the study by Van Nostrand et al. (Cell Death and Differentiation 2017).

We thank the reviewer for reminding us of this study. We have now mentioned and cited the study in our manuscript as follows:

"Triple knockout of *Trp53*, and its homologues *Trp63* and *Trp73* has been reported to impair mesendoderm formation in mouse embryos (Wang et al, 2017). However, one grossly normal triple-knockout embryo has been described at E11 by others (Van Nostrand et al., 2017), and p53-deficient extraembryonic cells have been shown to contribute to embryonic gut formation (Batki et al., 2024). We thus set out to test how germ layer specification was influenced by p53 deletion in our mosaic competitive conditions..."

4. Is there some way to quantitate E-cadherin and FoxA2 in Ext. Fig. 2b/c? Also, why are endoderm markers relatively unchanged between p53KO and WT cells in Ext. Fig. 2e, which contrasts with what is mentioned about Ext. Fig. 2 b/c?

Following the reviewer's suggestion, we have added multiple new independent repeats of the experiment and quantified the staining intensities of n=78-125 gastruloids per condition and marker. This new data is now depicted in Extended Data Figures 2i-j. To address the question about the endoderm marker transcript levels, we now generated separate plots for E- and N-Cadherin, Sox17, and FoxA2 transcript levels from sorted RNAseq populations of 120h gastruloids (Extended Data Figures 2k-n). This data more rigorously confirmed that the total FoxA2 protein levels in WT+p53KO mosaic gastruloids are reduced compared to WT+WT gastruloids (Ext. Data Fig. 2j), although the surviving WT-mCherry cells in WT+p53KO gastruloids seem to display increased FoxA2 transcript levels (Ext. Data Fig. 2n). Fittingly, we also observe a reduction in Sox17 transcript levels in p53KO cells together with increased Sox17 transcript levels in neighboring WT-mCherry cells (Ext. Data Fig. 2m). Together, this confirms our previous observation that p53KO cells might intrinsically be biased against the formation of endodermal lineages, which is interesting in the context of a recent study demonstrating that p53KO cells from extraembryonic lineages are nonetheless able to participate in the formation of the embryonic gut (PMID: 38849542).

When quantifying the protein levels of E-cadherin in this larger number of repeats, we observed only a very slight but non-significant reduction in WT+KO gastruloids (Ext. Data Fig. 2i), although on a transcript level p53KO cells displayed a significant reduction in *Cdh1* (E-Cadherin) and increase in *Cdh2* (N-Cadherin) (Ext. Data Fig. 2k-l). This could support the notion that p53KO cells are biased toward ectodermal lineages rather than endodermal lineages, a point we discuss with Reviewer 2 under point 7 as well.

5. To solidify the proliferation data, it is important to examine other proliferation markers than p-H3, such as Ki67 or BrdU.

We again thank the reviewer for suggesting strengthening the confidence in our data. We have now conducted full cell cycle analysis using combined EdU and Hoechst staining during all timepoints of mosaic gastruloid formation. This is now depicted in Figure 3c-d and Extended Data Figures 3b-d. This new analysis confirmed our previous pH3 data in that WT-mCherry cells do not have any alterations to their cell cycle phase distribution independent of whether they are in mosaic gastruloids with p53KO or WT control cells. This new data did however reveal a higher percentage of WT-emiRFP cells and a lower percentage of p53KO cells in G1/0 phase, with a corresponding increase of p53KO cells in S-phase (Extended Data Figure 3b,c). We additionally conducted Ki67 staining analyzed by flow cytometry and immunofluorescence in 3D mosaic gastruloids and observed that almost all cells were Ki67 positive (data not shown), which prohibited quantification. This is not surprising given that Ki67 only marks cells as negative in G0 phase, which pluripotent cells at this early developmental stage do not usually reside in. We thank the reviewer for suggesting the more detailed exploration of the proliferation profiles through EdU, which has strengthened our manuscript.

6. Can the authors do assays to show definitively that Bcl2 is acting in the ES cells to inhibit apoptosis? That is an important control for the mechanism they propose.

It is true that regulation of stress and apoptosis machineries differ between pluripotency/development and adult tissues. This is exemplified by the observation that naïve ES cells contain high transcript and protein levels of p53 (Fig. 1a; Fig. 6n) without detectable levels of its downstream target p21 (Fig. 6n, "0h"). We thus followed the reviewer's suggestion to test whether Bcl2 overexpression truly inhibits apoptosis in our system, as tested by cleaved Caspase 3 staining and flow cytometry. Indeed, we observe that our inducible Bcl2-overexpression clone can undergo apoptosis in gastruloids in the absence of doxycycline but entirely lacks cleaved-caspase 3 positive cells at 72h even in mosaic gastruloids containing p53KO supercompetitors, when the Bcl2 expression is induced (Extended Data Fig. 3m). This is an important control, and we are happy to now have included it in the manuscript.

7. The argument that the wild-type cells are affected by intrinsic apoptosis could be stronger. For example, inhibiting Caspase 3 or 9 should have an effect on competition but inhibiting Caspase 8 should not. The FADD deletion experiments are not sufficient to make this claim, especially since no western blot confirming ablation of the protein is shown.

Following the reviewer's advice we have attempted to test the effect of various Caspase inhibitors on cell competition. Counterintuitively, we observed in multiple repeats problems of toxicity from the treatments during gastruloid formation. We unfortunately do not have a good explanation for this, but we have now toned down the emphasis on intrinsic apoptosis throughout the manuscript and instead speak of "mitochondrial apoptosis", which we confirm solidly by the complete rescue through Bcl2 overexpression.

8. Can the authors provide proof that ROS was inhibited in Figure 4/5?

We thank the reviewer for raising this point. We have attempted multiple times to measure ROS levels in gastruloids using ROS sensitive fluorescent dyes (CellROX Green reagent, Thermo Fisher) but observed that diffusion of the dye into gastruloids was much slower than the reaction speed of the assay, such that the gradient of staining ranging from negative centers to saturated edges made the analysis unreliable. N-Acetyl-L-Cysteine (NAC) is a commonly used ROS scavenger/quencher, and we observe a concentration dependent growth impairment of cells at higher concentrations of NAC, indicating that the compound was present during our treatments. Nonetheless, because our observations regarding this compound's effect on cell competition were negative observations, and in lack of a confident assessment of ROS levels, we decided to exclude this data from the manuscript.

9. The text describing Fig. 5c indicates that modulating ActivinA, Nodal, and TGF β signaling had a non-significant effect on cell competition, but the stats on these compared to DMSO are not clear. In general, the effects of the various factors are not shown in the clearest way. Inhibitors and activators of pathways should be clearly separated and explained. Also, where is the proof that these activators and inhibitors are inducing/inhibiting the expected pathways?

We apologize if the data had previously not been clearly presented. The statistical analysis in the previous Figure 5c was a Two-way ANOVA with post hoc Tukey's test corrected for multiple comparisons. In this, all

signaling perturbations were tested against each other within a WT+p53KO setting, which indicated that cell competition in ChIR and BMP4 treated gastruloids was significantly less pronounced than in most other conditions. To improve clarity of the presentation of this data, we have now restructured Figure 5. Within each signaling pathway we now compare the cell competition effect size between agonists and antagonists of that same pathway, to ask whether modulation toward both extremes has any influence on the amount of cell competition in the system. Toward this, we have also conducted additional experiments and included new data on Wnt signaling antagonists IWP2 and XAV-939 as comparison to the Wnt agonist ChIR. This new analysis confirmed that Wnt and BMP modulation influenced cell competition effect size, while Nodal and ERK signaling modulation does not (Fig. 5a-d). We still included a multiple comparison analysis across all perturbations as shown before (Fig. 5e), which depicts that ChIR and BMP4 treatment are the only conditions with significantly lower competition than the annotated other pathway perturbations. All agonists and antagonists used in this experiment are commercially available and widely used standards in the field. None of the compounds were in-house produced or modified, and all compounds were used according to literature and the manufacturer's instructions.

10. Fig. 6n is very important for showing specific characteristics of each time point of gastruloid development, as it describes essential genes for apoptosis, cell cycle arrest, etc. However, it is not clear how this analysis was done. Was this bulk RNA-seq or something else? If bulk RNA-seq was done on cells collected from each time point, pathway analysis should be done to define pathways at different timepoints. Also, why would p21 be induced after Puma and Noxa, which are lower affinity p53 target genes? Perhaps p21 is induced in a p53-independent manner?

We thank the reviewer for recognizing the importance of this Figure. The time-resolved transcriptomic heatmaps in Figure 6 are based on the same bulk sequencing data as displayed in Figure 1j (120h). As the reviewer guessed correctly, this data was generated by harvesting pooled mosaic gastruloids every 24h from 0-144h of gastruloid formation, separating the two fluorescent populations by FACS, followed by bulk mRNA sequencing of each group. We generated a dataset of 69 samples to have 3 independent repeats of the 3 starting populations (0h), as well as 3 independent repeats of the sorted 4 populations from WT+WT or WT+p53KO mosaic gastruloids for all timepoints 24h-140h. This dataset will be available with GEO accession number GSE294530 upon publication of this manuscript and can already be viewed by the reviewers using the following link and login token:

<https://www.ncbi.nlm.nih.gov/geo/query/acc.cgi?acc=GSE294530>

Enter token **upmzwusirdctbmr** into the box.

We have compared WT-mCherry cells from either WT+WT or WT+p53KO mosaic gastruloids to each other to test how their transcriptome responds to the supercompetitor neighbors. Interestingly, we observed almost no differentially expressed genes between these conditions until 96h (Extended Data Fig. 2p), after cell competition had already peaked. This suggested that cell competition on the loser cell side might not be transcriptionally regulated but likely results in cell death before transcriptional changes can occur. However, following the reviewer's suggestion we now have additionally conducted GSEA pathway analysis to pick up on small transcriptional changes converging in the same pathways (Extended Data Fig. 2q). Surprisingly, here we found that WT-mCherry cells in mosaic gastruloids with p53KO cells downregulated growth related pathways including MTORC1 signaling, MYC targets, and E2F targets from 48h onward, even before cell competition related apoptosis is observed. Additionally, from 96h onward we observe an upregulation in "p53-pathway", which might fit an epigenetic release of p53 targets after a sharp downregulation of Trim24 at 72h, a previously described epigenetic repressor of p53 transcriptional targets (PMID: 37386214), which we now included in Figure 6n and the main text. The downregulation of growth and cell division related gene sets was surprising, given that we had not observed any significant differences in proliferation or cell cycle distribution.

Regarding the temporal regulation of p21, Puma, and Noxa expression, it is important to note that Noxa is surprisingly also upregulated in p53KO cells (Fig. 6n), which indicates that its transcription is regulated independently of p53 and thus not constrained by p53 binding affinity. Additionally, in pluripotent stem cells, p21 expression is repressed even when high levels of p53 protein are present. In human ESCs, the p21 promoter is epigenetically repressed by high H3K27me3 occupancy but kept in a bivalent poised state through concomitantly H3K4me3 occupancy such that p53 protein can bind, but no p21 transcription takes place (PMID: 27346849). Additionally, as mentioned above, we observe high expression of Trim24 in all clones during pluripotency, which is sharply downregulated at 72h. Trim24 has been shown to bind to poised promoters of p53 transcriptional targets and blocking their transcription (PMID: 37386214), which might be the reason why p21 transcription only occurs after 72h of gastruloid formation. We thus believe

that the expression of these three genes might be regulated by different machineries, rather than directly competing for p53 affinity.

We are grateful to the reviewer for asking us to explore this data further, as it now adds valuable new insights into the mechanistic regulation of the stress response.

11. In the discussion the authors state that “loser cells respond to fitter cells by stabilizing p53 protein”. The authors should show that p53 is being stabilized in loser cells. Also, does knockout of p53 target genes improve the fitness of the loser cells?

We thank the reviewer for the detailed focus on p53 regulation and its downstream targets. Regarding the question whether knockout of p53 target genes would improve fitness of loser cells, work by others has demonstrated that knockout of p21 and Noxa does not improve fitness over other WT cells (PMID: 38131530) (PMID: 38489392), but that knockout of Puma does create a supercompetitor phenotype in WT cells (PMID: 38489392). The list of p53 transcriptional targets is unfortunately very long, but we agree that our setup would be a good screening platform to test the relevance of a large number p53 downstream targets toward cell competition and might set up such screen in the future.

Regarding the stabilization of p53 protein in loser cells, the original statement referred to prior literature in 2D monolayer cultures of canine MDCK cells, which demonstrated that compression-sensing via Rho-associated Kinase led to an increase in p53 levels and ultimately apoptosis (PMID: 27109213). Following the reviewer’s suggestion we have now conducted p53 quantifications in our own cell competition setting and demonstrate directly that WT-mCherry cells acutely upregulate p53 protein levels when in the presence of p53KO cells, compared to when being in control mosaic gastruloids with other WT cells (Fig. 3g-k; Extended Data Fig. 3h-i). For higher confidence, we have quantified p53 protein levels at 72h in mosaic gastruloids using flow cytometry, as well as immunofluorescence and image analysis of 3D confocal stacks. When analyzing the mean population levels, we observed an increase of p53 in WT-mCherry cells with p53KO neighbors compared to with WT-emiRFP neighbors. However, more excitingly, we observed a 4-fold increase in cells with extremely high p53 levels (>4.5x of interquartile range) when WT-mCherry cells were together with p53KO neighbors (Fig. 3j). While we had not realized this during the original submission, this had never been directly demonstrated in a mouse developmental model. Instead, previous literature described that cells with pre-established high p53 levels turned into loser cells, but not that encountering supercompetitors dynamically raised p53 protein levels in their neighbors. We thank the reviewer for suggesting these experiments and adding additional novelty to our manuscript.

Additional minor points:

1. Extended Fig 1g schematic is confusing, as it is not clear why one big cell rather than a cluster of cells is indicated in the right side of the image.

The bigger cell was meant to indicate growth in a figurative manner, but we realize how this depiction is inaccurate and have replaced it by a cluster of cells to indicate increase in numbers rather than increase in size. We thank the reviewer for helping us increase the visual clarity in this.

2. “ChIR” treatment is first mentioned in Ext. Fig. 2b/c, but in the text is not described until the last couple figures. This should be explained when it first mentioned.

We thank the reviewer for pointing this out. We have now explained the pulse of Wnt agonism through ChIR treatment early when first introducing gastruloids in the results section as part of Figure 1.

3. Fig. 2k cited in text, but there is no Fig. 2k.

We thank the reviewer for finding this labeling mistake. The figure previously referred to is now Figure 2n. The main text has been adjusted accordingly.

4. In Ext. Fig 3c, right panel, the p53KO labels appear to be mixed up.

Very well spotted. We thank the reviewer and have adjusted this label in what is now Supplementary Figure 4c.

5. In general, the labeling of panels in figures and the figure legends could be improved to make the paper less confusing to read. For example, in Fig. 1D, each clone type could be labeled with the actual name (like WT-mCherry).

We again thank the reviewer for helping us improve clarity of the manuscript. We have adjusted the

mentioned label in what is now Extended Data Figure 2a as well as in other places throughout the manuscript toward a more uniform naming of clones.

Panels of new experimental data and analyses conducted during the revision process:

Figure 3 a,c,d,e,f,g,h,i,j,k,s

Figure 4 a,b,j,o,p

Figure 5 a

Figure 6 o

Figure 8 a,b,c,d,e,f,g,h,i,j,k

Extended Data Figure 1 b, o

Extended Data Figure 2 i,j,k,l,m,n, q

Extended Data Figure 3 b,c,d,e,f,g,h, i, m

Extended Data Figure 4 j,k,l,m,

Extended Data Figure 6 i

Supplementary Figure 3 a

Supplementary Figure 5 b,c

Supplementary Figure 6 a,b,c,d

Supplementary Figure 7 a,b,c,d

Supplementary Figure 8 a,b,c,d,e,f,g

Reviewers' comments and point-by-point response:

Reviewer #1:

Remarks to the Author:

In this revised manuscript, the authors have comprehensively addressed all previous concerns. The study successfully demonstrates that utilizing gastruloids—and stem cell-based embryo models more broadly—effectively bridges the gap between traditional 2D cultures and in vivo embryogenesis. By using the 3D gastruloid model, the authors provide a more physiologically relevant platform for studying cell competition, avoiding the "incomplete or artificial readouts" often associated with 2D systems. Key Strengths of the Revision. The data highlighting the temporal nature of cell competition and the differences between 2D vs. 3D gastruloids are particularly compelling. The inclusion of new results regarding p53-degron and glucose manipulation significantly strengthens the mechanistic depth of the paper. All remaining technical points have been adequately clarified. Given the high quality of the revisions and the importance of the findings, I recommend the manuscript for publication.

Response: We thank the reviewer kindly for their support and guidance during the revision process.

Reviewer #2:

Remarks to the Author:

We are very satisfied with the in depth and thoughtful response of the authors to the criticism and suggestions from all reviewers. Our points have all been addressed, so happy to recommend publication of the work in its revised state.

Remarks on code availability:

The code was reviewed and found to work for the information required to support the data presented. However, access to qlivecell is apparently not accessible more broadly at present.

Response: We thank the reviewer for their kind words and for being part of the engaging and stimulating discussion during the revisions. We have now made qlivecell fully available to the public and we appologize for the delay in this.

Reviewer #3:

Remarks to the Author:

The authors have thoroughly addressed my comments. This paper will be a great contribution to the field!

Response: We thank the reviewer for their positive words and for helping to improve this manuscript to its current state.